# An Interpretable Contrastive GAN Approach for Identifying Heterogeneous Pathological Imaging Patterns

## Abstract

Despite the rapid development of representation learning applied to neuroimaging, accurately disentangling the heterogeneity of neurological diseases remains a significant challenge. Typically, unsupervised approaches may capture disease heterogeneity that is dominated by confounding factors rather than pathological changes in brain structure or function. Existing semi-supervised methods can reveal disease-specific subtypes or dimensions by contrasting with a background population, but they usually rely on the assumption that non-pathological variations are identically distributed between background and target datasets—a condition often unmet in real-world data. To address this, we introduce **InfoSepGAN**, a contrastive generative framework designed to separate context (non-pathological) and attribute (pathological) factors between background and target datasets, reducing biases in learned disease-related representations when the assumption is violated. Furthermore, we regularize the learned imaging patterns for continuity, sparsity, and monotonicity, ensuring distinct and interpretable disease-related patterns along each dimension. Finally, InfoSepGAN employs a "synthetic twin" mechanism to perform subject-level counterfactual reconstruction, generating non-pathological counterparts for each patient and providing visualizations of disease-related regions. Experiments on both synthetic and real-world Alzheimer's disease datasets demonstrate that InfoSepGAN effectively extracts pathological imaging patterns while adjusting for potential confounders, outperforming recent baseline methods in both accuracy and interpretability.

## 1 Introduction

Neuroimaging has become an indispensable tool for investigating the human brain, allowing direct assessment of structural and functional changes in vivo and across large populations. These advances have substantially deepened our understanding of how diseases influence the brain's structure and function. Nevertheless, accurately capturing the clinical and neurobiological heterogeneity inherent in neurological disorders remains a formidable challenge for precise diagnosis and targeted therapy (Young et al., 2018; Vogel et al., 2021; Wen et al., 2025). Recent advances in deep learning offer an opportunity to disentangle this heterogeneity and quantify individualized disease characteristics (Leonenko et al., 2021; Wen et al., 2024).

To uncover data-driven, neurobiologically plausible and disease-specific subtypes or dimensions, semi-supervised learning methods have been widely applied to parse heterogeneity in a target group (TG, denoted by $Y$) of patients by contrasting with a background group (BG, denoted by $X$) of healthy controls (Filipovych et al., 2012; Varol et al., 2015; Eavani et al., 2016; Wen et al., 2020). They implicitly rely on the assumption that BG and TG share the same non-pathological (context) distribution. However, this assumption does not necessarily hold in practice, which may result in confounding variations, such as demographics, spuriously incorporated into the learned disease-related representations. This is common in clinical studies where sampling bias may lead to an age imbalance between control and patient groups.

With a distinct perspective, Contrastive Analysis (CA) approaches aim to uncover two underlying generative factors that ($i$) only present in target samples and not in background samples—termed **attribute factors** $\mathbf{z}_a$, and that ($ii$) are shared between the two groups (BG and TG)—termed **context factors** $\mathbf{z}_c$ (Abid & Zou, 2019; Weinberger et al., 2022; Zou et al., 2022; Louiset et al., 2023).

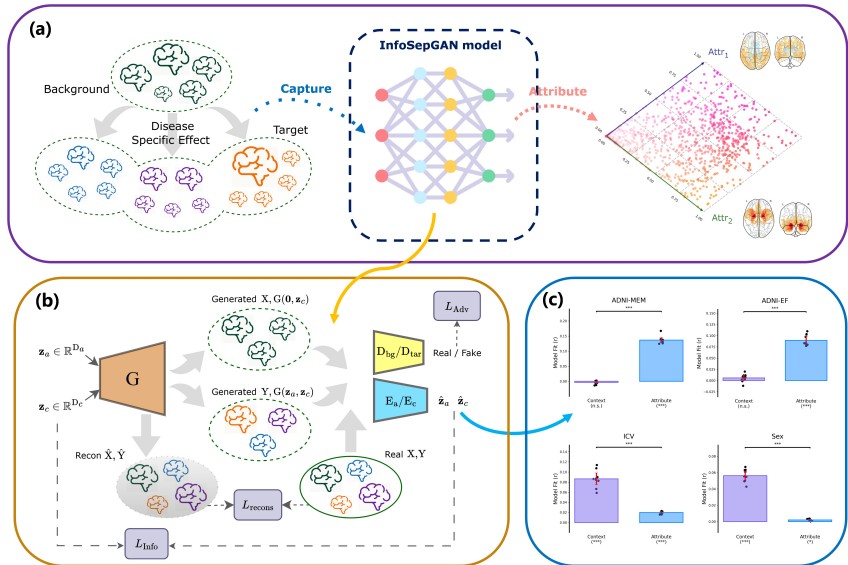

Figure 1: Overview of InfoSepGAN. (a) Disease heterogeneity leads to distinct brain structural changes (color variations represent pathology-related patterns captured by $\mathbf{z}_a$, different sizes reflect non-pathological variations captured by $\mathbf{z}_c$). (b) General architecture of InfoSepGAN. (c) $\mathbf{z}_c$ capture variability shared by BG and TG, while $\mathbf{z}_a$ capture target-related patterns.

However, most existing methods rarely discuss the potential of CA in mitigating biases caused by distributional shifts in the context space—situations where non-pathological brain variations (context factors) are unequally distributed between BG and TG. Moreover, neuroimaging applications of these methods often lack explicit regularization of the learned disease-specific representations (attribute factors), resulting in these features that are less interpretable and less biologically meaningful. To address these challenges, we introduce **InfoSepGAN**, a contrastive generative framework that aims to robustly and interpretably decipher disease-specific neuroanatomical heterogeneity from cross-sectional neuroimaging data by leveraging CA principles. The overview of InfoSepGAN is shown in Fig. 1.

We validate InfoSepGAN through extensive experiments on synthetic, semi-synthetic, and real-world Alzheimer's Disease (AD) data. Our results show that InfoSepGAN robustly extracts pathological imaging patterns and outperforms current state-of-the-art methods. In particular, our key contributions are: ($i$) We propose a novel framework that disentangles neuroimaging data into context and attribute factors, while mitigating biases introduced by differences in non-pathological variation between BG and TG; ($ii$) We regularize the attribute space to obtain continuous, sparse, and monotonic representations aligned with disease severity. In this way, each dimension captures a distinct disease pattern, providing an interpretable way for patient-level severity assessment; and ($iii$) We conduct a subject-level counterfactual reconstruction, which produces non-pathological counterparts of patient data to highlight disease-related variability in brain structure and provide intuitive visualizations of affected regions.

## 2 RELATED WORK

Neuroimaging research is moving from traditional case-control comparisons to methods that capture the full spectrum of disease heterogeneity. Early case-control approaches relied on standard statistical tests to identify group-level differences, but cannot capture patient-level variability (Habeck et al., 2008; Hampel et al., 2008; Ewers et al., 2011). This limitation motivated the adoption of machine learning and deep learning techniques for heterogeneity analysis.

Unsupervised learning methods attempt to discover latent subtypes or dimensions directly in the TG domain (Fig. 2a). Examples include SuStaIn (Young et al., 2018), Non-negative Matrix Factorization (Chen et al., 2023b), and Bayesian Latent Dirichlet Allocation (Zhang et al., 2016). While powerful for heterogeneity analysis, these approaches remain limited in avoiding potential non-pathological

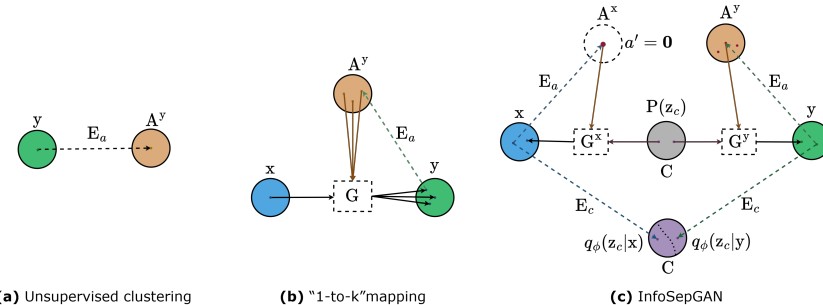

(a) Unsupervised clustering     (b) "1-to-$k$"mapping     (c) InfoSepGAN

Figure 2: Overview of three heterogeneity analysis paradigms. Solid lines denote the generative assumptions each method makes, whereas dashed lines indicate inference paths (e.g., mappings from observed data to latent factors). (a) Unsupervised clustering performed directly in the TG domain. (b) Semi-supervised "1-to-$k$" mapping from BG to TG. (c) Our method models data using a shared context space $C$ and separated attribute spaces $A^x$ (e.g., fixed to a constant $\mathbf{a}'$) and $A^y$. It uncovers two underlying generative factors via approximate posteriors over $\mathbf{z}_a$ and $\mathbf{z}_c$. E and G represent the learned inference and generative mappings respectively, operating between the two domains.

brain variations.

To this end, several semi-supervised learning methods try to overcome this challenge by introducing BG to guide heterogeneity analysis, typically by modeling a "1-to-k" mapping from BG to TG (Fig. 2b). Representative approaches such as Dong et al. (2015), Varol et al. (2017), and Yang et al. (2021) identify categorical subtypes by clustering patients through the patterns or transformations between BG and TG. In contrast, SurrealGAN (Yang et al., 2024) models disease heterogeneity along continuous dimensions by learning multiple BG–TG mappings with an inverse function for parsing heterogeneity, reflecting the fact that many chronic brain disorders develop along a spectrum rather than in discrete clusters. Nevertheless, these methods implicitly assume that BG and TG share comparable non-pathological variations (context), i.e., $P(\mathbf{z}_c|\mathbf{x}) = P(\mathbf{z}_c|\mathbf{y})$, due to without modeling of latent context factors. This assumption is often violated in clinical neuroimaging due to scanner/site differences or sampling bias. In this case, these approaches fail to rule out non-pathological variations (context), which are then spuriously captured within disease-specific (attribute) representations. While known confounders can be corrected using linear adjustments, it is impractical to account for the various potential confounding factors, and their effects may be non-linear. These challenges highlight the need for adaptive methods that remain robust when the distribution assumption is violated.

Recently, Contrastive Analysis (CA) has emerged as a principled framework, with diverse applications and extensions. For example, contrastive VAEs (Abid & Zou, 2019; Aglinskas et al., 2022) and GAN-based model (Carton et al., 2024) were developed to capture higher-level semantics by distinguishing target-specific from shared variation. In parallel, information-theoretic method (Louiset et al., 2024) leverage contrastive learning principles to extract semantically relevant representations for CA tasks. Another related approach is NestedVAE (Vowels et al., 2020), which explicitly models common factors between visual domains to reduce bias, but requires paired data. Collectively, these methods emphasize separating shared context from target-specific attributes, thereby enabling more semantically structured representation learning. Motivated by these insights, we propose InfoSepGAN, a framework that robustly and interpretably disentangles pathological neuroanatomical heterogeneity from neuroimaging data by leveraging CA principles. Our model is applied to regional volume (ROIs) data (volumes of different graymatter (GM) and whitematter (WM) regions) derived from structural MRI (Fig. 9).

## 3 METHODS

We apply to a standard Contrastive Analysis setup. Let $X = \{\mathbf{x}_i\}_{i=1}^{N_X}$ and $Y = \{\mathbf{y}_j\}_{j=1}^{N_Y}$ denote BG and TG datasets. Both were assumed to be sampled i.i.d. from a conditional distribution $p_\theta(\cdot|\mathbf{z}_a, \mathbf{z}_c)$ with unknown parameter $\theta$, where $\mathbf{z}_c \in \mathbb{R}^{D_c}$ represents **context factors** shared between $X$ and $Y$, and $\mathbf{z}_a \in \mathbb{R}^{D_a}$ represents **attribute factors** specific to $Y$. Accordingly, the

generative process is: $\mathbf{x}_i \sim p_\theta(\mathbf{x}_i|\mathbf{z}_a = \mathbf{a}', \mathbf{z}_c)$, $\mathbf{y}_j \sim p_\theta(\mathbf{y}_j|\mathbf{z}_a, \mathbf{z}_c)$, where the attribute factors of $X$ are fixed to a constant $\mathbf{a}'$ (e.g., $\mathbf{0}$), so that $X$ is explained solely by $\mathbf{z}_c$. We define the latent factors priors as $P_Y(\mathbf{z}_a) \sim \mathcal{U}[0,1]^{D_a}$, $P_X(\mathbf{z}_a) \sim \delta(\mathbf{z}_a = \mathbf{0})$, and $P_X(\mathbf{z}_c) = P_Y(\mathbf{z}_c) = \mathcal{N}(\mathbf{0}, \boldsymbol{I}_{D_c})$. Latent factors are typically assumed to be independent and to follow factorized priors: (i.e., $P(\mathbf{z}_a, \mathbf{z}_c) = P(\mathbf{z}_a)P(\mathbf{z}_c) = \prod_{i=1}^{D_a} P((\mathbf{z}_a)_i) \prod_{j=1}^{D_c} P((\mathbf{z}_c)_j)$. Our objective is to *separate* and *infer* latent variables $\mathbf{z}_a$ and $\mathbf{z}_c$ given the input data $X$ and $Y$. To achieve that, we introduce an encoder $E$ parameterized by $\phi$. This encoder is decomposed into two modules, $E_a$ and $E_c$, for inferring the latent factors $\mathbf{z}_a$ and $\mathbf{z}_c$ respectively. The conditional posteriors are approximated as $q_\phi(\mathbf{z}_a, \mathbf{z}_c|\cdot) \approx E(\cdot)$, and are typically assumed to be conditionally independent: $q_\phi(\mathbf{z}_a, \mathbf{z}_c|\cdot) = q_\phi(\mathbf{z}_a|\cdot)q_\phi(\mathbf{z}_c|\cdot)$. Unlike prior "1-to-k" mapping–based approaches (Fig. 2b) only modeling factor $\mathbf{z}_a$, our framework employs distinct encoders $E_a$ and $E_c$ to model the posteriors of $\mathbf{z}_a$ and $\mathbf{z}_c$ simultaneously, with both encoders shared between $X$ and $Y$ domains. This design is advantageous when the assumption $P(\mathbf{z}_c|\mathbf{x}) = P(\mathbf{z}_c|\mathbf{y})$ is violated in practice: by jointly modeling context factors between both domains, InfoSepGAN can recognize and mitigate biases induced by distributional shifts in context space, thereby preventing non-pathological variability from leaking into the attribute space and ensuring separation between $\mathbf{z}_a$ and $\mathbf{z}_c$ (Fig. 2c). The general architecture of InfoSepGAN is illustrated in Fig. 1b. Our framework integrates adversarial learning, information-theoretic constraints, reconstruction, and attribute space regularization to achieve robust disentanglement. Each loss serves a distinct purpose, summarized below:

## 3.1 Adversarial GAN Loss

Following the standard GAN framework (Goodfellow et al., 2020), we employ a generator $G$ and a discriminator $D$ in a min–max game. The generator produces synthetic BG and TG samples as $\mathbf{x}' = G(\mathbf{0}, \mathbf{z}_c)$ and $\mathbf{y}' = G(\mathbf{z}_a, \mathbf{z}_c)$, while $D(\cdot)$ outputs the probability that the input is real data rather than synthetic data. Therefore, the discriminator $D$ attempts to maximize the adversarial objective while the generation function $G$ attempts to minimize against it. To explicitly model class-specific structures, we decompose the generator into $G_{\text{bg}} : \mathbb{Z}_c \to \mathbb{X}$ and $G_{\text{diff}} : \mathbb{Z}_a \times \mathbb{X} \to \mathbb{Y}$. Specifically, $G_{\text{bg}}$ synthesizes BG samples, while $G_{\text{diff}}$ models the disease-related differences, yielding $\mathbf{y}' = G(\mathbf{z}_a, \mathbf{z}_c) = G_{\text{bg}}(\mathbf{z}_c) + G_{\text{diff}}(\mathbf{z}_a, G_{\text{bg}}(\mathbf{z}_c)) = \mathbf{x}' + G_{\text{diff}}(\mathbf{z}_a, \mathbf{x}')$, subject to the constraint $G_{\text{diff}}(\mathbf{0}, \mathbf{x}') = \mathbf{0}$ which means that $\mathbf{z}_a$ contributes only to disease-specific brain variations. Similarly, the discriminator is decomposed into $D_{\text{bg}}$ and $D_{\text{tar}}$, each specialized for background and target domains. The adversarial loss is denoted as $\mathcal{L}_{\text{Adv}}(D, G)$ (see Eq. 4), and the full details and complete loss formulation are provided in Appendix A.1.

## 3.2 Information Regularization Loss

We introduce mutual information-based regularization terms to encourage informative latent representations, inspired by InfoGAN (Chen et al., 2016). Unlike the original InfoGAN which separates informative from nuisance factors, our objective is to explicitly disentangle context and attribute factors. Because the context factors $\mathbf{z}_c$ are intended to capture variability shared across both domains, we maximize their mutual information with the generated background samples $\mathbf{x}'$ as well as with the generated target samples $\mathbf{y}'$. Likewise, we maximize the mutual information between the attribute factors $\mathbf{z}_a$ and **only** the target samples $\mathbf{y}'$, ensuring that $\mathbf{z}_a$ encodes target-specific structure. To guarantee that background variability is captured entirely by $\mathbf{z}_c$, we further enforce the attribute factors of the BG domain to be always equal to a constant value $\mathbf{a}'$ (i.e., contain no information). This is achieved by minimizing the KL divergence between $P(\mathbf{z}_a|\mathbf{x}')$ and a Dirac delta distribution $\delta(\mathbf{a}')$ (e.g., $\mathbf{a}' = \mathbf{0}$). Our goal is to maximize the following cost function:

$$\arg\max_{E,G} \lambda_1\big(I(\mathbf{x}';\mathbf{z}_c) + I(\mathbf{y}';\mathbf{z}_c)\big) + \lambda_2 I(\mathbf{y}';\mathbf{z}_a) \quad \text{s.t.} \quad D_{\text{KL}}((\mathbf{z}_a)_{\mathbf{x}}\|\delta(\mathbf{a}')) = 0.$$

We approximate these mutual-information terms using variational lower bounds $\mathcal{L}_{\text{Info}}(E, G)$ (see Eq. 5). The detailed derivations and implementation are provided in Appendix A.2.

## 3.3 Reconstruction Loss

We use a variational objective function to both reconstruct observed data and obtain parametrized approximations to the intractable posteriors over $(\mathbf{z}_a, \mathbf{z}_c)$. The lower bound for the target domain

can be written as:

$$\log P(\mathbf{y}) \geq \mathbb{E}_{q_\phi(\mathbf{z}_a, \mathbf{z}_c \mid \mathbf{y})}\big[\log P(\mathbf{y} \mid \mathbf{z}_a, \mathbf{z}_c)\big] - D_{\mathrm{KL}}\big(q_\phi(\mathbf{z}_a|\mathbf{y}) \,\|\, P(\mathbf{z}_a)\big) - D_{\mathrm{KL}}\big(q_\phi(\mathbf{z}_c|\mathbf{y}) \,\|\, P(\mathbf{z}_c)\big),$$

while the background lower bound follows the same by setting $\mathbf{z}_a = \mathbf{0}$ (section A.5).

We model $P(\cdot \mid \mathbf{z}_a, \mathbf{z}_c)$ as a Laplace distribution $\mathbf{L}(\mu, 1)$, which leads to an $\ell_1$ reconstruction term $\mathcal{L}_{\mathrm{recons}}(E, G)$ (see Eq. 10). The KL terms act as regularizers constraining the posterior distributions $q_\phi(\mathbf{z}_a, \mathbf{z}_c|\cdot)$ to match their priors. Since $D_{\mathrm{KL}}(q_\phi(\mathbf{z}_a|\mathbf{y})\|\mathcal{U}[0,1]^{D_a})$ does not have a closed-form solution, we replace it with the Cramer–Wold (CW) distance (Knop et al., 2020), which yields a kernel-based metric to control the distributional discrepancy. The resulting prior regularization term is denoted as $\mathcal{L}_{\mathrm{prior}}(E)$ (Eq. 11, see Appendix A.6 for details).

### 3.4 Attribute Space Regularization

A continuous latent variable, by itself, does not guarantee interpretable or clinically meaningful patterns. To ensure that the attribute space reflects plausible pathological processes, we constrain the function class of the generator $G$ and the encoder $E$ with several regularization terms. We adapt and extend the terms supposed by Yang et al. (2024) to the contrastive generative framework. Specifically, these regularizations are integrated to encourage attribute factors $\mathbf{z}_a$ to correspond to distinct, localized, and progressively interpretable changes in neuroimaging data. (1) We impose a sparsity loss ($\mathcal{L}_{\mathrm{sparse}}$) on the generated pathological differences to reflect the clinical reality that pathology often affects localized brain regions. (2) We also introduce a pattern separation loss ($\mathcal{L}_{\mathrm{disen}}$) to encourage each dimension of the attribute vector $\mathbf{z}_a$ to encode a unique and orthogonal anatomical variation. (3) A monotonicity loss ($\mathcal{L}_{\mathrm{mono}}$) is used to ensure a positive correlation between the latent attribute value and the severity of the generated changes, reflecting disease progression. (4) Additionally, we employ a background consistency loss ($\mathcal{L}_{\mathrm{bg}}$) to ensure that the generator produces no pathological changes when the attribute factors are set to zero. (5) Furthermore, to mitigate mode collapse and encourage diverse, semantically meaningful representations, we constrain the model's parameters to ensure Lipschitz continuity and apply a decomposition loss ($\mathcal{L}_{\mathrm{decom}}$). The detailed mathematical formulations for all these regularization terms are provided in Appendix A.7.

**All Regularization.** All structural constraints, including sparsity, pattern separation, monotonicity, background consistency, and decomposition, are combined into a single regularization term:

$$\mathcal{L}_{\mathrm{regular}} = \lambda_{\mathrm{sparse}}\mathcal{L}_{\mathrm{sparse}} + \lambda_{\mathrm{disen}}\mathcal{L}_{\mathrm{disen}} + \lambda_{\mathrm{mono}}\mathcal{L}_{\mathrm{mono}} + \lambda_{\mathrm{bg}}\mathcal{L}_{\mathrm{bg}} + \lambda_{\mathrm{decom}}\mathcal{L}_{\mathrm{decom}}. \tag{1}$$

### 3.5 Total Loss

Our full objective combines adversarial training, information maximization, reconstruction, prior regularization, and the above structural constraints. Details of our training procedure and model architecture are provided in Appendix A.8. Specifically, the total loss is defined as

$$\mathcal{L}_{\mathrm{total}}(E, G, D) = \mathcal{L}_{\mathrm{Adv}} + \lambda_{\mathrm{Info}}\mathcal{L}_{\mathrm{Info}} + \lambda_{\mathrm{recons}}\mathcal{L}_{\mathrm{recons}} + \lambda_{\mathrm{prior}}\mathcal{L}_{\mathrm{prior}} + \mathcal{L}_{\mathrm{regular}}. \tag{2}$$

The final optimization follows a min–max game:

$$E^*, G^* = \arg\min_{E,G} \max_{D} \ \mathcal{L}_{\mathrm{total}}(E, G, D). \tag{3}$$

### 3.6 Model Application

Our framework offers several key applications of the learned representation (latent factors). First, we can use the generator $G$ to generate new background and target samples that are consistent with the underlying data-generating process, such that $G(\mathbf{0}, \mathbf{z}_c) \approx p_\theta(\mathbf{x}|\mathbf{0}, \mathbf{z}_c)$ and $G(\mathbf{z}_a, \mathbf{z}_c) \approx p_\theta(\mathbf{y}|\mathbf{z}_a, \mathbf{z}_c)$. Second, the encoders $E_a$ and $E_c$ enable direct inference of latent both attribute and context factors for any subject, yielding quantitative measures of both attributes and context: $(\hat{\mathbf{z}}_a)_i = E_a(\mathbf{y}_i)$ and $(\hat{\mathbf{z}}_c)_i = E_c(\mathbf{y}_i)$. Third, we can construct a *synthetic twin* for each subject by setting the attribute vector to zero: $\bar{\mathbf{y}}_i = G(\mathbf{0}, (\hat{\mathbf{z}}_c)_i)$. This twin serves as a matched counterpart that preserves background variations while removing target-specific features, providing an intuitive visualization tool for individualized analysis. Finally, extending this counterfactual idea to a study group $S$, we compute the mean relative change: $\frac{1}{N_S} \sum_{i \in S} ((\bar{\mathbf{y}}_i - \mathbf{y}_i)/\bar{\mathbf{y}}_i)$, which summarizes the characteristic imaging alterations of the group and highlights disease-specific patterns at a population level (shown in Appendix B.10).

## 4 EXPERIMENTAL SETUP

Our method is evaluated on a series of synthetic, semi-synthetic, and real-world brain MRI datasets to validate its ability to learn representations of underlying disease-related imaging patterns.

### 4.1 DATASETS

**Synthetic Data.** To test our model's ability to recover disease-related imaging patterns under unbalanced confounding effects, we designed synthetic experiments with the ground truth. We generated 100 baseline ROI features for 1,800 subjects (900 BG; 900 Pseudo-TG). The baseline values $v_{ij}$ for the $j$-th ROI of the $i$-th subject were sampled from $\mathcal{N}(1, 0.1)$. For Pseudo-TG subjects, we simulated three atrophy patterns with severity $s_{ik} \sim \mathcal{U}[0, 1]$, reducing affected ROIs as: $v_{ij} \leftarrow v_{ij} - v_{ij} * s_{ik} * \epsilon_{ijk} * 0.3$ (where $\epsilon_{ijk} \sim \mathcal{N}(1, 0.1)$ adds noise), with partial ROI overlap across patterns to mimic realistic disease processes. To introduce confounding, two additional patterns with severity $\mathbf{c}_i = (c_{i1}, c_{i2}) \sim \mathcal{U}[0, 1]^2$ were applied to subsets of both BG and Pseudo-TG: $v_{ij} \leftarrow v_{ij} - v_{ij} * c_{ik} * 0.3$. Confounding regions might overlap with atrophy regions (depending on the scenario). By varying (1) confounding severity, (2) the proportion of affected subjects, and (3) the degree of confounding–atrophy overlap, we constructed seven representative scenarios to evaluate model's ability to capture disease effect. Full details are in Appendix B.1.

**Semi-Synthetic Data.** To assess robustness under more realistic conditions, we further designed semi-synthetic experiments based on real brain MRI data from the UK Biobank (UKBB). After excluding individuals diagnosed with neurological or systemic diseases, we obtained regional volumetric measurements for 95 ROIs from 30,858 healthy controls (HCs). Following the synthetic setup, we generated Pseudo-TG subjects by sampling subsets of HCs and applying simulated atrophy patterns to their baseline ROI volumes $v_{ij}$ were modified using atrophy patterns parameterized by severity scores $s_{ik} \sim \mathcal{U}[0, 1]$. This procedure preserves the natural biological variation in real MRI data while explicitly preserving the ground-truth disease effects for evaluation. Unlike the fully synthetic case, no extra confounding effects were added because real MRI data already contain inherent biological and demographic confounding. We designed two sets of semi-synthetic experiments: (1) *Pattern-based experiments:* Similar to Yang et al. (2022), we varied the structure and severity of imposed atrophy patterns across five datasets (e.g., Semi Basic, Large Overlap, Large noise, Mild, Scarce) to test model robustness against different disease-pattern complexities. (2) *Age-confounding experiments:* We treated age as a confounder and varied BG–TG age overlap across four levels, simulating unbalanced confounding effect (No, Small, Middle, and Large Age Gap). For both experiment types, we also compared results with and without linear correction for age, sex, and intracranial volume (ICV) to test whether the model can adaptively mitigate real-world confounding in the absence of prior adjustment and accurately separate true atrophy signals. Detailed parameterizations and preprocessing procedures are provided in the Appendix B.2.

**Real Data.** We further validated our framework on structural MRI data from the Alzheimer's Disease Neuroimaging Initiative (ADNI), including 2,587 participants after quality control (988 cognitively normal (CN) as BG; 1,599 mild cognitive impairment (MCI) or Alzheimer's disease (AD) as TG). Baseline T1-weighted images underwent standardized preprocessing: intensity correction, skull-stripping, multi-atlas segmentation into 139 ROIs, and merging symmetric ROIs to yield 72 ROI volumes as features. Site-specific mean and variance were estimated from BG using a validated statistical harmonization approach (Pomponio et al., 2020) and applied to the full cohort (see Appendix B.6).

### 4.2 EVALUATION & BASELINES

**Evaluation Metrics.** Similar to Yang et al. (2022), we primarily used the Concordance Index (c-index) to evaluate model performance. For synthetic and semi-synthetic experiments, all hyperparameters were set to default values (Appendix A.8). In each setting, we independently ran the model 10 times. For each ground-truth pattern $\mathbf{s}_k$, we calculated a concordance index (c-index) between the inferred attribute factor $\mathbf{z}_{a,k}$ and $\mathbf{s}_k$. Averaging across all $D_a$ derived c-indices yielded the *pattern-c-index* (PCI), which we used as the primary evaluation metric. We also quantified the pairwise agreement between any two trained models by calculating a *pattern-agr-index* (ACI), which is the pattern-c-index between the attribute factors derived by the two different models. We further introduce two additional metrics, Pattern-Pearson-Correlations (PPC) and Pattern-Difference-

Correlations (PDC), to assess performance from different perspectives (Details in Appendix A.9). For real-world data, where no ground truth exists, we evaluated candidate hyperparameter configurations by varying $D_a = 2, 3, 4$ and $D_c = 3, 4, 5$, running the model 10 times for each combination. For each configuration, we measured the pairwise agreement of the inferred attribute factors across runs and selected the run with the highest average agreement as the optimal model and derived subject-level attribute factors by the optimal one, ensuring the stability and reliability of the results. **Ablation and Robustness.** We conducted comprehensive ablation and robustness experiments. In the ablation study, key regularization terms were individually removed to quantify their contributions, and the context encoder $E_c$ was ablated to evaluate its role in managing distributional shifts across context space. Building on this, the robustness study systematically varied the core hyperparameters to assess the model's sensitivity to perturbations. Detailed results for both ablation and robustness analyses are provided in Appendix B.5.

**Baseline Methods.** We benchmarked our framework against six representative methods for learning disease-related imaging patterns, which can be grouped into three categories. (1) Generative Models: We included SurrealGAN, a state-of-the-art (SOTA) generative model specifically designed for learning disease-related imaging patterns based on '1-to-k' mapping. (2) Classical Dimensionality Reduction: We compared our method to classical approaches including NMF and Factor Analysis (FA), which are widely applied in biomedical research to capture dominant latent structures. We also included Orthogonal Projective NMF (opNMF) (Sotiras et al., 2015), which extends standard NMF with an orthogonality constraint to improve interpretability, and LDA, which has been adapted to discover hidden subtypes in neuroimaging cohorts (Zhang et al., 2016). (3) Contrastive Analysis Methods: We considered Contrastive Principal Component Analysis (cPCA) (Abid et al., 2018), a linear method that identifies components enriched in the target group relative to BG, as well as the non-linear CA family methods, including CAVAE (Aglinskas et al., 2022), DoubleInfoGAN (Carton et al., 2024), and MMCAVAE (Weinberger et al., 2022). Detailed preprocessing pipelines and implementation settings for all baselines are provided in Appendix B.3.

# 5 RESULTS

## 5.1 RESULTS ON SYNTHETIC DATA

We first present results on synthetic data designed to evaluate the ability of the models to disentangle disease-related from non-disease signals under varying confounding effects (Fig. 3a). We primarily compare InfoSepGAN against SurrealGAN, while results of other baselines (FA, LDA, NMF, Op-NMF, cPCA) are reported in the Appendix B.4. In both Basic setting without confounding and the balanced confounding scenarios (Conf and Conf Ovlp), SurrealGAN performed slightly better, but both models maintained high accuracy. In contrast, the next four scenarios (Conf Sev Ovlp through Conf Extreme) introduce increasingly unbalanced confounding, thereby violating the assumption: $P(\mathbf{z}_c|\mathbf{x}) = P(\mathbf{z}_c|\mathbf{y})$. In these more challenging settings, InfoSepGAN consistently outperforms SurrealGAN, illustrating its capacity to handle heterogeneous populations. The most extreme case (Conf Extreme), where confounding affects all TG subjects but none of the BG subjects, both models experience substantial performance drops. This is expected, as confounding becomes indistinguishable from true disease signals when present only in TG. Without any additional information to correct this bias, it becomes impossible to disentangle confounding from pathology signal. Overall, the synthetic experiments show that while both models perform well under balanced conditions, InfoSepGAN maintains superior accuracy when confounding is unevenly distributed across BG and TG populations. This advantage underscores its applicability to real-world datasets, where confounding effects are typically imbalanced. Additional evidence of InfoSepGAN's generalization is provided in Appendix B.4.

## 5.2 RESULTS ON SEMI-SYNTHETIC DATA

We next evaluated InfoSepGAN on semi-synthetic datasets derived from UKBB brain MRI data, focusing on two experiment groups: pattern-based and age-confounding. Table 1 and Fig. 3b,c illustrate comparisons between InfoSepGAN and SurrealGAN, while additional baseline models and generalization results are provided in Appendix B.4. For the *Pattern-based experiments*, we tested five scenarios—Basic, Large Noise, Large Overlap, Mild, and Scarce—to examine whether models can capture ground-truth disease effects across varying levels of structural complexity. Performance

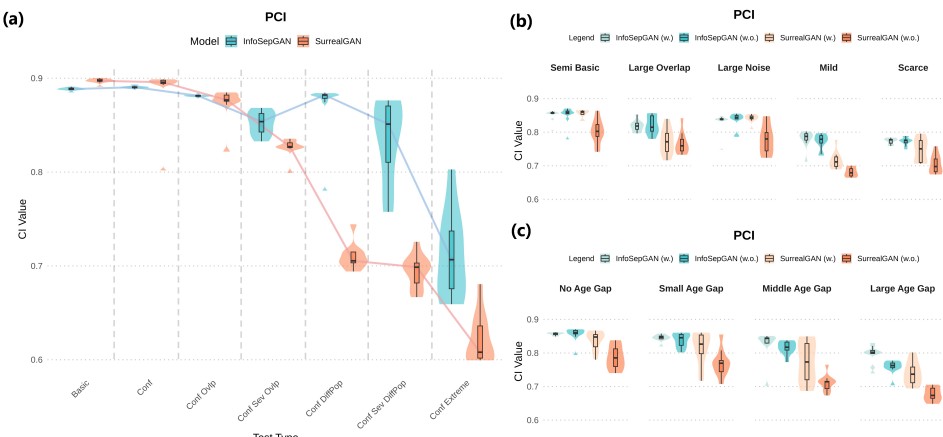

Figure 3: InfoSepGAN and SurrealGAN results on (Semi-) Synthetic data, displayed as violin–box plots. All accuracies are measured by the Pattern-c-index. (a) Synthetic data. The x-axis lists increasing confounding difficulty: Basic (no confounding), Conf and Conf Ovlp (balanced confounding), followed by four progressively unbalanced settings (Conf Sev Ovlp to Conf Extreme) (b,c) Semi-synthetic pattern-based and age-confounding experiments. "(w.)" vs "(w.o.)" indicates results with vs without linear correction for age, sex, and ICV.

Table 1: Semi-synthetic experiments (**without** linear adjustment; results reported as mean $\pm$ std). **Bold** marks the best mean performance. Statistical significance is indicated as $^{***}p < 0.0001$, $^{**}p < 0.001$, $^{*}p < 0.05$, and $^{\text{n.s.}}$ (not significant). "I" denotes InfoSepGAN, and "S" denotes SurrealGAN. The definitions of the metrics are detailed in Appendix A.9

| Experi | Methods | PCI | ACI | PPC | PDC |
|---|---|---|---|---|---|
| Semi Basic | I | $\mathbf{0.851^{*} \pm 0.027}$ | $\mathbf{0.901^{***} \pm 0.025}$ | $\mathbf{0.885^{*} \pm 0.043}$ | $\mathbf{0.880^{*} \pm 0.069}$ |
| | S | $0.802 \pm 0.036$ | $0.840 \pm 0.013$ | $0.795 \pm 0.065$ | $0.767 \pm 0.103$ |
| Large Overlap | I | $\mathbf{0.821^{*} \pm 0.027}$ | $\mathbf{0.859^{\text{n.s.}} \pm 0.011}$ | $\mathbf{0.836^{*} \pm 0.044}$ | $\mathbf{0.787^{*} \pm 0.076}$ |
| | S | $0.767 \pm 0.033$ | $0.856 \pm 0.013$ | $0.726 \pm 0.062$ | $0.586 \pm 0.115$ |
| Large Noise | I | $\mathbf{0.835^{*} \pm 0.025}$ | $\mathbf{0.893^{***} \pm 0.022}$ | $\mathbf{0.859^{*} \pm 0.042}$ | $\mathbf{0.844^{*} \pm 0.066}$ |
| | S | $0.779 \pm 0.0413$ | $0.814 \pm 0.016$ | $0.748 \pm 0.085$ | $0.674 \pm 0.138$ |
| Mild | I | $\mathbf{0.773^{***} \pm 0.023}$ | $\mathbf{0.844^{***} \pm 0.018}$ | $\mathbf{0.742^{***} \pm 0.048}$ | $\mathbf{0.694^{***} \pm 0.097}$ |
| | S | $0.681 \pm 0.013$ | $0.760 \pm 0.020$ | $0.523 \pm 0.034$ | $0.284 \pm 0.057$ |
| Scarce | I | $\mathbf{0.773^{***} \pm 0.012}$ | $\mathbf{0.854^{***} \pm 0.013}$ | $\mathbf{0.743^{***} \pm 0.025}$ | $\mathbf{0.715^{***} \pm 0.048}$ |
| | S | $0.705 \pm 0.028$ | $0.744 \pm 0.010$ | $0.580 \pm 0.065$ | $0.406 \pm 0.145$ |
| No Age Gap | I | $\mathbf{0.855^{**} \pm 0.023}$ | $\mathbf{0.904^{***} \pm 0.024}$ | $\mathbf{0.891^{**} \pm 0.035}$ | $\mathbf{0.873^{*} \pm 0.061}$ |
| | S | $0.787 \pm 0.035$ | $0.847 \pm 0.010$ | $0.767 \pm 0.067$ | $0.688 \pm 0.133$ |
| Small Age Gap | I | $\mathbf{0.838^{**} \pm 0.021}$ | $\mathbf{0.885^{***} \pm 0.020}$ | $\mathbf{0.867^{**} \pm 0.034}$ | $\mathbf{0.858^{*} \pm 0.054}$ |
| | S | $0.770 \pm 0.040$ | $0.827 \pm 0.015$ | $0.728 \pm 0.078$ | $0.706 \pm 0.119$ |
| Middle Age Gap | I | $\mathbf{0.816^{***} \pm 0.020}$ | $\mathbf{0.868^{**} \pm 0.026}$ | $\mathbf{0.812^{***} \pm 0.043}$ | $\mathbf{0.807^{***} \pm 0.075}$ |
| | S | $0.711 \pm 0.025$ | $0.819 \pm 0.010$ | $0.585 \pm 0.055$ | $0.525 \pm 0.119$ |
| Large Age Gap | I | $\mathbf{0.759^{***} \pm 0.020}$ | $0.820 \pm 0.030$ | $\mathbf{0.677^{***} \pm 0.044}$ | $\mathbf{0.671^{***} \pm 0.079}$ |
| | S | $0.677 \pm 0.020$ | $\mathbf{0.851^{*} \pm 0.010}$ | $0.488 \pm 0.048$ | $0.406 \pm 0.099$ |

decreased under the Mild and Scarce settings, where disease effects were either subtle relative to normal inter-individual variation or limited to a few ROIs. Nevertheless, InfoSepGAN consistently outperformed SurrealGAN, demonstrating superior sensitivity in identifying sparse or subtle pathological signals. For the *Age-confounding experiments*, we explicitly treated age as a confounder by varying the overlap between BG and pseudo-TG age distributions. Across all four overlap scenarios, InfoSepGAN achieved higher accuracy than SurrealGAN, indicating that the model effectively handles unbalanced confounding in realistic settings. Across both experiment groups, we compared performance with and without linear correction for age, sex, and ICV. InfoSepGAN consistently outperformed all baselines, and importantly, its performance remained stable regardless of whether

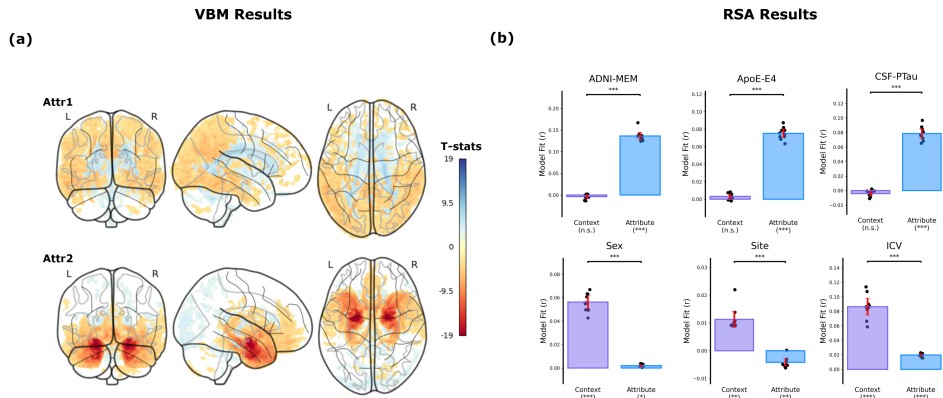

Figure 4: (a) The two attribute factors (Attr1 and Attr2) quantify individual differences along two dimensions of AD severity, each associated with distinct patterns of gray-matter atrophy. These patterns are shown via voxel-wise t-tests performed for each attribute factor while controlling for age, sex, intracranial volume (ICV), and the other attribute factor. Results are FDR-corrected at $p < 0.05$. (b) Representational Similarity Analysis (RSA) between learned attributes or context features and clinical or nonclinical variables on the ADNI dataset. Bars indicate similarity scores and red vertical lines indicate 95% confidence intervals. Significance is denoted by asterisks (* p < 0.05, ** p < 0.001, *** ,p < 0.0001)

prior covariate adjustment was applied. This demonstrates that our framework can adaptively mitigate confounding influences, reducing the need for researchers to rely on extensive prior knowledge of potential confounding factors. By contrast, other baselines showed greater sensitivity to preprocessing, reflecting their limitations when $p(\mathbf{z}_c|\mathbf{x}) \neq p(\mathbf{z}_c|\mathbf{y})$. Overall, these semi-synthetic experiments show that InfoSepGAN reliably recovers true disease-related patterns under diverse and challenging conditions while maintaining strong performance in the presence of real-world confounding effects.

## 5.3 RESULTS ON REAL DATA

We applied InfoSepGAN to ADNI cohort and selected $D_a = 2$, which yielded the highest model agreement. To interpret learned representations, we performed voxel-based morphometry (VBM) with Nilearn (Abraham et al., 2014), examining voxel-wise associations between each attribute component and GM tissue. The two learned attribute factors captured distinct gray matter atrophy patterns revealed by voxel-wise t-tests controlling for age, sex, ICV, and the remaining one attribute factor (Fig. 4a). Attr1 reflected diffuse cortical atrophies across parietal, occipital, and temporal regions (Du et al., 2007), whereas Attr2 highlighted focal medial temporal and subcortical atrophies, consistent with characteristic AD pathology (Scheltens et al., 1992; Yi et al., 2016). The reproducibility of the identified patterns was demonstrated through a random half-split of the ADNI dataset (Fig. 10), as well as through experiments with external UKBB reference groups (Fig. 11). Additionally, we applied the trained model to the UKBB general population to further explore the cognitive relevance of the derived attribute factors in a broad asymptomatic cohort (Appendix B.8). Correlation analysis with clinical variables (Table 15) revealed different associations between the two attributes. Both Attr1 and Attr2 were strongly associated with memory dysfunction (ADNI-MEM), overall cognitive impairment (MMSE), executive function (TMT-B), and CSF-Abeta, while Attr1 additionally showed stronger associations with WMH, CSF-Tau, and CSF-pTau. To assess the disentanglement achieved by InfoSepGAN, we performed RSA comparing attribute and context features with a broad set of clinical and nonclinical variables (Fig. 4b and Fig. 12). Attribute features showed significant associations with AD-specific properties, including cognitive scores (ADNI-MEM, ADNI-EF, ADNI-LAN), CSF biomarkers (Tau, pTau), and Apoe-E4 genotype, whereas context features showed no such associations. Site appeared only in the context space, consistent with its role in capturing variation unrelated to disease status. Attribute features were also found to be associated with age, sex and ICV, which is consistent with previous studies showing that these variables affect brain structural changes differently in CN and MCI/AD populations (Dukart et al., 2013;

Ferretti et al., 2018; Wolf et al., 2004). In addition, WMH burden is generally increased across progressive stages of cognitive decline (Kamal et al., 2023), and we observed that attribute features captured substantially greater variation in WMH. These findings suggest that InfoSepGAN disentangles general effects from those that specifically interact with AD. Full statistical results and an ablation version of InfoSepGAN without the context module, which showed that removing $E_c$ (context submodule) increases the influence of nonclinical factors within the attribute space and weakens its association with AD-specific variables, are provided together with Table 16 and Table 17.

Finally, the generation of synthetic twins provides individualized counterfactual reconstructions that highlight subject-specific pathological regional variations, showing progressive, severity-dependent changes along the learned attribute dimensions and offering an intuitive visual representation of learned disease patterns. The high concordance between synthetic twins and VBM results in key brain regions further supports the reliability of our model (Fig. 14, see Appendix B.10). Together, these experiments demonstrate that InfoSepGAN robustly identifies clinically meaningful neuroanatomical patterns and successfully disentangles AD-related atrophy from non-pathological variations in a fully observational data (see Appendix B.9).

### 5.4 LIMITATIONS AND PERSPECTIVES

A persistent challenge in contrastive analysis is the lack of formal identifiability guarantees (the conditions required for models to recover the true latent factors of the data generating process). Nonlinear models such as VAEs and GANs are generally non-identifiable (Locatello et al., 2019). Although recent work proposes achieving identifiability by imposing structural constraints on the encoder (Kivva et al., 2022) or by introducing auxiliary variables that render latent factors conditionally independent (Khemakhem et al., 2020), these strategies are not compatible with CA settings. Consequently, while InfoSepGAN effectively disentangles attribute from context factors, it cannot ensure recovery of all true generative factors, which is a limitation shared by all CA-based approaches and an important direction for future work.

## 6 CONCLUSION

In this study, we presented InfoSepGAN, a novel contrastive generative framework that disentangles heterogeneous pathological imaging patterns and non-pathological variation. By reducing bias arising from distributional shifts in the context space and applying regularization in the attribute space, the model robustly extracts continuous and interpretable disease representations. Experiments on synthetic and AD datasets confirm its superiority over existing methods, highlighting its promise for robust and interpretable heterogeneity analysis in neuroimaging and other complex biomedical data.

## ETHICS STATEMENT

All participants in this work, as well as the paper submission, adhere to the ICLR Code of Ethics (`https://iclr.cc/public/CodeOfEthics`).

## REPRODUCIBILITY STATEMENT

We affirm that all results in this work are fully reproducible. Appendix A provides the theoretical proofs and details of the model hyperparameter configurations. Appendix B describes the experimental setup in full. The source code will be made publicly available upon publication of the paper.

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

## A  APPENDIX: METHODS

### A.1  ADVERSARIAL GAN LOSS

This appendix provides the full technical specification of the adversarial component summarized in the main text. While the main paper focuses on the conceptual GAN design and the role of $G_{\text{bg}}$ and $G_{\text{diff}}$, here we detail the exact sampling procedures, architectural decomposition, and the complete adversarial loss used in InfoSepGAN.

The generator takes latent variables $(\mathbf{z}_a, \mathbf{z}_c)$ drawn from the prior $P(\mathbf{z}_a, \mathbf{z}_c)$ and produces synthetic BG and TG samples via: $\mathbf{y}' = G(\mathbf{z}_a, \mathbf{z}_c) = \mathbf{x}' + G_{\text{diff}}(\mathbf{z}_a, \mathbf{x}') = G_{\text{bg}}(\mathbf{z}_c) + G_{\text{diff}}(\mathbf{z}_a, G_{\text{bg}}(\mathbf{z}_c))$.

The constraint $G_{\text{diff}}(\mathbf{0}, \mathbf{x}') = \mathbf{0}$ ensures that $\mathbf{z}_a$ contributes exclusively to disease-related variability. The discriminator is similarly decomposed into domain-specific experts $D_{\text{bg}}$ and $D_{\text{tar}}$, allowing it to evaluate samples from background and target domains.

Besides sampling from the prior, we have another way to generate TG data: $\tilde{\mathbf{y}} = \mathbf{x} + G_{\text{diff}}(\mathbf{z}_a, \mathbf{x})$, which can be regarded as drawing $\mathbf{z}_c$ from the true posterior $P(\mathbf{z}_c|\mathbf{x})$ implicit in $\mathbf{x} = G_{\text{bg}}(\mathbf{z}_c^*)$ ($\mathbf{z}_c^*$ is the ground-truth context factor for $X$). Motivated by VAE-GAN (Larsen et al., 2016), we additionally introduce reconstruction samples for the discriminator from the posterior distributions $q_\phi(\mathbf{z}_a, \mathbf{z}_c|\cdot)$, that is, $\hat{\mathbf{x}} = G(E(\mathbf{x}))$ and $\hat{\mathbf{y}} = G(E(\mathbf{y}))$. We observe empirically that discriminating based on these diverse generated samples from $P(\mathbf{z}_a, \mathbf{z}_c)$, $P(\mathbf{z}_c|\mathbf{x})$, and $q_\phi(\mathbf{z}_a, \mathbf{z}_c|\cdot)$ provide more useful learning signals, leading to stable training and improved synthesis. The adversarial loss is therefore defined as:

$$
\begin{aligned}
\mathcal{L}_{\text{Adv}}(D, G) &= \mathbb{E}_{\mathbf{x} \sim P(\mathbf{x})}\big[\log D_{\text{bg}}(\mathbf{x})\big] + \mathbb{E}_{\mathbf{z}_c \sim P_X(\mathbf{z}_c)}\big[\log\big(1 - D_{\text{bg}}(\mathbf{x}')\big)\big] \\
&\quad + \mathbb{E}_{\mathbf{x} \sim P(\mathbf{x}), \hat{\mathbf{z}}_c = E_c(\mathbf{x})}\big[\log\big(1 - D_{\text{bg}}(\hat{\mathbf{x}})\big)\big] \\
&\quad + \mathbb{E}_{\mathbf{y} \sim P(\mathbf{y})}\big[\log D_{\text{tar}}(\mathbf{y})\big] + \mathbb{E}_{\mathbf{z}_a, \mathbf{z}_c \sim P_Y(\mathbf{z}_a, \mathbf{z}_c)}\big[\log\big(1 - D_{\text{tar}}(\mathbf{y}')\big)\big] \\
&\quad + \mathbb{E}_{\mathbf{y} \sim P(\mathbf{y}), (\hat{\mathbf{z}}_a, \hat{\mathbf{z}}_c) = E(\mathbf{y})}\big[\log\big(1 - D_{\text{tar}}(\hat{\mathbf{y}})\big)\big] \\
&\quad + \mathbb{E}_{\mathbf{x} \sim P(\mathbf{x}), \mathbf{z}_a \sim P_Y(\mathbf{z}_a)}\big[\log\big(1 - D_{\text{tar}}(\tilde{\mathbf{y}})\big)\big].
\end{aligned}
\tag{4}
$$

### A.2  INFORMATION REGULARIZATION LOSS

In the main text, we defined the mutual-information objective that encourages the context factors $\mathbf{z}_c$ to capture shared variability and the attribute factors $\mathbf{z}_a$ to encode target-specific structure. Here we provide the full derivation of the variational lower bounds used to approximate these mutual-information terms and the resulting information regularization loss. Because $\mathbf{z}_a$ and $\mathbf{z}_c$ are independent by construction, the joint mutual information decomposes as: $I((\mathbf{z}_a, \mathbf{z}_c); \cdot) = I(\mathbf{z}_a; \cdot) + I(\mathbf{z}_c; \cdot)$ (Appendix A.3). We approximate these terms by variational lower bounds with auxiliary encoders $E_a$ and $E_c$, modeled as factorized distributions (Lin et al., 2020). This brings to a reconstruction loss, where we adopt $\ell_1$ loss (a factorized Laplace distribution $\mathbf{L}(\mu, 1)$, Appendix A.4).

For generated background samples $G(\mathbf{0}, \mathbf{z}_c)$, we maximize $I(\mathbf{z}_c; \mathbf{x}')$, while enforcing that the attribute encoder outputs a constant value (e.g., $\mathbf{0}$), reflecting the information-less constraint on $\mathbf{z}_a$ for BG samples. For generated target samples $G(\mathbf{z}_a, \mathbf{z}_c)$, we maximize both $I(\mathbf{z}_c; \mathbf{y}')$ and $I(\mathbf{z}_a; \mathbf{y}')$ using the same variational bounds. Moreover, to guarantee that real background samples carry no target-specific information, we explicitly constrain $\mathbf{z}_a$ to equal a constant $\mathbf{0}$ when encoding $\mathbf{x}$, and include $I(\mathbf{z}_a = \mathbf{0}; \mathbf{x})$ as an additional regularization. Finally, to better train $E_a$, we also include a term based on $\tilde{\mathbf{y}}$ (see section 3.1), which corresponds to maximizing $I(\mathbf{z}_a; \tilde{\mathbf{y}})$. The overall information regularization loss is defined as:

$$
\begin{aligned}
\mathcal{L}_{\text{Info}}(E, G) &= \mathbb{E}_{\mathbf{z}_c \sim P_Y(\mathbf{z}_c)}\big[w_c \|E_c(G(\mathbf{0}, \mathbf{z}_c)) - \mathbf{z}_c\|_1 + w_a \|E_a(G(\mathbf{0}, \mathbf{z}_c)) - \mathbf{0}\|_1\big] \\
&\quad + \mathbb{E}_{\mathbf{z}_a, \mathbf{z}_c \sim P_Y(\mathbf{z}_a, \mathbf{z}_c)}\big[w_c \|E_c(G(\mathbf{z}_a, \mathbf{z}_c)) - \mathbf{z}_c\|_1 + w_a \|E_a(G(\mathbf{z}_a, \mathbf{z}_c)) - \mathbf{z}_a\|_1\big] \\
&\quad + w_a \mathbb{E}_{\mathbf{x} \sim P(\mathbf{x}), \mathbf{z}_a \sim P_Y(\mathbf{z}_a)}\big[\|E_a(\mathbf{x} + G_{\text{diff}}(\mathbf{z}_a, \mathbf{x})) - \mathbf{z}_a\|_1\big] \\
&\quad + w_a \mathbb{E}_{\mathbf{x} \sim P(\mathbf{x})}\big[\|E_a(\mathbf{x}) - \mathbf{0}\|_1\big].
\end{aligned}
\tag{5}
$$

### A.3 LEMMA1

Since $\mathbf{z}_a$ and $\mathbf{z}_c$ are supposed to be independent, the mutual information $I((\mathbf{z}_a, \mathbf{z}_c); \cdot)$ can be decomposed into the sum of the two mutual information $I(\mathbf{z}_a; \cdot) + I(\mathbf{z}_c; \cdot)$. Take $I((\mathbf{z}_a, \mathbf{z}_c); \mathbf{y}')$ as an example:

$$
\begin{aligned}
I((\mathbf{z}_a, \mathbf{z}_c); \mathbf{y}') &= -H((\mathbf{z}_a, \mathbf{z}_c) \mid \mathbf{y}') + H(\mathbf{z}_a, \mathbf{z}_c) \\
&= -H((\mathbf{z}_a, \mathbf{z}_c) \mid \mathbf{y}') + H(\mathbf{z}_a) + H(\mathbf{z}_c) \\
&= \iiint P(\mathbf{z}_a, \mathbf{z}_c, \mathbf{y}') \log\left(\frac{P(\mathbf{z}_a, \mathbf{z}_c, \mathbf{y}')}{P(\mathbf{y}')}\right) d\mathbf{z}_a \, d\mathbf{z}_c \, d\mathbf{y}' + H(\mathbf{z}_a) + H(\mathbf{z}_c) \\
&= \int P(\mathbf{y}') \iint P(\mathbf{z}_a \mid \mathbf{y}') P(\mathbf{z}_c \mid \mathbf{y}') \log\bigl(P(\mathbf{z}_a \mid \mathbf{y}') P(\mathbf{z}_c \mid \mathbf{y}')\bigr) d\mathbf{z}_a \, d\mathbf{z}_c \, d\mathbf{y}' \\
&\quad + H(\mathbf{z}_a) + H(\mathbf{z}_c) \\
&= \int P(\mathbf{y}') \iint P(\mathbf{z}_a \mid \mathbf{y}') P(\mathbf{z}_c \mid \mathbf{y}') \left(\log P(\mathbf{z}_a \mid \mathbf{y}') + \log P(\mathbf{z}_c \mid \mathbf{y}')\right) d\mathbf{z}_a \, d\mathbf{z}_c \, d\mathbf{y}' \\
&\quad + H(\mathbf{z}_a) + H(\mathbf{z}_c) \\
&= \int_{\mathbf{y}'} P(\mathbf{y}') \left[ \int_{\mathbf{z}_a} P(\mathbf{z}_a \mid \mathbf{y}') \log P(\mathbf{z}_a \mid \mathbf{y}') \, d\mathbf{z}_a + \int_{\mathbf{z}_c} P(\mathbf{z}_c \mid \mathbf{y}') \log P(\mathbf{z}_c \mid \mathbf{y}') \, d\mathbf{z}_c \right] d\mathbf{y}' \\
&\quad + H(\mathbf{z}_a) + H(\mathbf{z}_c) \\
&= \mathbb{E}_{\mathbf{y}' \sim P(\mathbf{y}'), \, \mathbf{z}_a \sim P(\mathbf{z}_a \mid \mathbf{y}')} \log P(\mathbf{z}_a \mid \mathbf{y}') + \mathbb{E}_{\mathbf{y}' \sim P(\mathbf{y}'), \, \mathbf{z}_c \sim P(\mathbf{z}_c \mid \mathbf{y}')} \log P(\mathbf{z}_c \mid \mathbf{y}') \\
&\quad + H(\mathbf{z}_a) + H(\mathbf{z}_c) \\
&= -H(\mathbf{z}_a \mid \mathbf{y}') - H(\mathbf{z}_c \mid \mathbf{y}') + H(\mathbf{z}_a) + H(\mathbf{z}_c) \\
&= I(\mathbf{z}_a; \mathbf{y}') + I(\mathbf{z}_c; \mathbf{y}').
\end{aligned}
\tag{6}
$$

### A.4 LEMMA2

We proof that minimizing $\mathcal{L}_{\text{Info}}(E, G)$ (Eq. 5) is equivalent to maximizing the lower bound of the mutual information $I((\mathbf{z}_a, \mathbf{z}_c); \cdot)$. Taking $I(\mathbf{z}_a; \mathbf{y}')$ as an example, we can derive:

$$
\begin{aligned}
I(\mathbf{z}_a; \mathbf{y}') &= -H(\mathbf{z}_a \mid \mathbf{y}') + H(\mathbf{z}_a) = \mathbb{E}_{\mathbf{y}' \sim P(\mathbf{y}'), \, \mathbf{z}_a \sim P(\mathbf{z}_a \mid \mathbf{y}')} \log P(\mathbf{z}_a \mid \mathbf{y}') + H(\mathbf{z}_a) \\
&= \int_{\mathbf{y}'} P(\mathbf{y}') \int_{\mathbf{z}_a} P(\mathbf{z}_a \mid \mathbf{y}') \log(P(\mathbf{z}_a \mid \mathbf{y}')) \, d\mathbf{z}_a \, d\mathbf{y}' + H(\mathbf{z}_a) \\
&= \int_{\mathbf{y}'} P(\mathbf{y}') \int_{\mathbf{z}_a'} P(\mathbf{z}_a' \mid \mathbf{y}') \log(P(\mathbf{z}_a' \mid \mathbf{y}')) \, d\mathbf{z}_a' + H(\mathbf{z}_a') \\
&= \int_{\mathbf{y}'} \int_{\mathbf{z}_a} \int_{\mathbf{z}_c} P(\mathbf{y}', \mathbf{z}_a, \mathbf{z}_c) \, d\mathbf{z}_c \, d\mathbf{z}_a \, d\mathbf{y}' \int_{\mathbf{z}_a'} P(\mathbf{z}_a' \mid \mathbf{y}') \log(P(\mathbf{z}_a' \mid \mathbf{y}')) \, d\mathbf{z}_a' + H(\mathbf{z}_a') \\
&= \int_{\mathbf{z}_c} P(\mathbf{z}_c) \int_{\mathbf{z}_a} P(\mathbf{z}_a) \int_{\mathbf{y}'} P(\mathbf{y}' \mid \mathbf{z}_a, \mathbf{z}_c) \int_{\mathbf{z}_a'} P(\mathbf{z}_a' \mid \mathbf{y}') \log(P(\mathbf{z}_a' \mid \mathbf{y}')) \, d\mathbf{z}_a' \, d\mathbf{y}' \, d\mathbf{z}_a \, d\mathbf{z}_c \\
&\quad + H(\mathbf{z}_a') \\
&= \int_{\mathbf{z}_c} P(\mathbf{z}_c) \int_{\mathbf{z}_a} P(\mathbf{z}_a) \int_{\mathbf{y}'} P(\mathbf{y}' \mid \mathbf{z}_a, \mathbf{z}_c) \int_{\mathbf{z}_a'} P(\mathbf{z}_a' \mid \mathbf{y}') \log\left(\frac{P(\mathbf{z}_a' \mid \mathbf{y}')}{Q(\mathbf{z}_a' \mid \mathbf{y}')} Q(\mathbf{z}_a' \mid \mathbf{y}')\right) d\mathbf{z}_a' \, d\mathbf{y}' \, d\mathbf{z}_a \, d\mathbf{z}_c \\
&\quad + H(\mathbf{z}_a') \\
&= \underbrace{\mathbb{E}_{\mathbf{z}_c \sim P(\mathbf{z}_c), \, \mathbf{z}_a \sim P(\mathbf{z}_a), \, \mathbf{y}' \sim P(\mathbf{y}' \mid \mathbf{z}_a, \mathbf{z}_c)} KL(P(\mathbf{z}_a' \mid \mathbf{y}') \,\|\, Q(\mathbf{z}_a' \mid \mathbf{y}'))}_{\geq 0} \\
&\quad + \mathbb{E}_{\mathbf{z}_c \sim P(\mathbf{z}_c), \, \mathbf{z}_a \sim P(\mathbf{z}_a), \, \mathbf{y}' \sim P(\mathbf{y}' \mid \mathbf{z}_a, \mathbf{z}_c)} \mathbb{E}_{\mathbf{z}_a' \sim P(\mathbf{z}_a' \mid \mathbf{y}')} \log Q(\mathbf{z}_a' \mid \mathbf{y}') + H(\mathbf{z}_a') \\
&\geq \mathbb{E}_{\mathbf{z}_c \sim P(\mathbf{z}_c), \, \mathbf{z}_a \sim P(\mathbf{z}_a), \, \mathbf{y}' \sim P(\mathbf{y}' \mid \mathbf{z}_a, \mathbf{z}_c)} \mathbb{E}_{\mathbf{z}_a' \sim P(\mathbf{z}_a' \mid \mathbf{y}')} \log Q(\mathbf{z}_a' \mid \mathbf{y}') + H(\mathbf{z}_a').
\end{aligned}
\tag{7}
$$

where we have introduced an auxiliary distribution $Q(\mathbf{z}_a' \mid \mathbf{y}')$, parameterized as a neural network, to approximate the posterior $P(\mathbf{z}_a' \mid \mathbf{y}')$ (which is difficult to compute), and we have made the hypothesis

that $\mathbf{z}_a$ does not depend on $\mathbf{z}_c$ (i.e., $P(\mathbf{z}_a \mid \mathbf{z}_c) = P(\mathbf{z}_a)$). To further remove also the need to sample from $P(\mathbf{z}'_a|\mathbf{y}')$ (which would be impossible in most cases), Chen et al. (2016) propose a simple, yet effective, modification of the previous variational lower bound. In their algorithm, they actually compute and maximize $\mathcal{L}(G, Q)$:

$$\mathcal{L}(G, Q) = \mathbb{E}_{\mathbf{z}_c \sim P(\mathbf{z}_c),\, \mathbf{z}_a \sim P(\mathbf{z}_a),\, \mathbf{y}' \sim P(\mathbf{y}'|\mathbf{z}_a,\mathbf{z}_c)} \mathbb{E}_{\mathbf{z}'_a \sim P(\mathbf{z}'_a|\mathbf{y}')} \log Q(\mathbf{z}'_a \mid \mathbf{y}') + H(\mathbf{z}'_a).$$

$$= \int P(\mathbf{z}_c) \int \int \int P(\mathbf{z}_a) P(\mathbf{y}'|\mathbf{z}_a,\mathbf{z}_c) P(\mathbf{z}'_a|\mathbf{y}') \log(Q(\mathbf{z}'_a|\mathbf{y}'))\, d\mathbf{z}'_a\, d\mathbf{y}'\, d\mathbf{z}_a\, d\mathbf{z}_c + H(\mathbf{z}'_a)$$

$$= \int P(\mathbf{z}_c) \int \int \int P(\mathbf{y}', \mathbf{z}_a|\mathbf{z}_c) P(\mathbf{z}'_a|\mathbf{y}') \log(Q(\mathbf{z}'_a|\mathbf{y}'))\, d\mathbf{z}'_a\, d\mathbf{y}'\, d\mathbf{z}_a\, d\mathbf{z}_c + H(\mathbf{z}'_a)$$

$$= \int P(\mathbf{z}_c) \int \int P(\mathbf{y}'|\mathbf{z}_c) P(\mathbf{z}'_a|\mathbf{y}', \mathbf{z}_c) \log(Q(\mathbf{z}'_a|\mathbf{y}'))\, d\mathbf{z}'_a\, d\mathbf{y}'\, d\mathbf{z}_c + H(\mathbf{z}'_a)$$

$$= \int P(\mathbf{z}_c) \int \int P(\mathbf{y}', \mathbf{z}'_a|\mathbf{z}_c) \log(Q(\mathbf{z}'_a|\mathbf{y}'))\, d\mathbf{z}'_a\, d\mathbf{y}'\, d\mathbf{z}_c + H(\mathbf{z}'_a)$$

$$= \int P(\mathbf{z}_c) \int \int P(\mathbf{y}', \mathbf{z}_a|\mathbf{z}_c) \log(Q(\mathbf{z}_a|\mathbf{y}'))\, d\mathbf{z}_a\, d\mathbf{y}'\, d\mathbf{z}_c + H(\mathbf{z}_a) \qquad (8)$$

$$= \int P(\mathbf{z}_c) \int P(\mathbf{z}_a) \int P(\mathbf{y}'|\mathbf{z}_a, \mathbf{z}_c) \log(Q(\mathbf{z}_a|\mathbf{y}'))\, d\mathbf{y}'\, d\mathbf{z}_a\, d\mathbf{z}_c + H(\mathbf{z}_a)$$

$$= \mathbb{E}_{\mathbf{z}_c \sim P(\mathbf{z}_c), \mathbf{z}_a \sim P(\mathbf{z}_a), \mathbf{y}' \sim P(\mathbf{y}'|\mathbf{z}_a,\mathbf{z}_c)} \log(Q(\mathbf{z}_a|\mathbf{y}')) + \underbrace{H(\mathbf{z}_a)}_{\text{constant}}$$

$$= \mathbb{E}_{\mathbf{z}_c \sim P(\mathbf{z}_c), \mathbf{z}_a \sim P(\mathbf{z}_a)} \log\left(Q(\mathbf{z}_a \mid G(\mathbf{z}_a, \mathbf{z}_c))\right) + C_1$$

$$\approx -\frac{1}{NM} \sum_{i=1}^{N} \sum_{j=1}^{M} \sqrt{\left(\mathbf{z}_a^{(i)} - E_a(G(\mathbf{z}_a^{(i)}, \mathbf{z}_c^{(j)}))\right)^T \left(\mathbf{z}_a^{(i)} - E_a(G(\mathbf{z}_a^{(i)}, \mathbf{z}_c^{(j)}))\right)} + C$$

Following Chen et al. (2016), we opt for simplicity by fixing the latent variable distribution and treating $H(\mathbf{z}_a)$ as a constant. Moreover, assuming that $Q(\mathbf{z}'_a \mid \mathbf{y}')$ follows a factorized Laplace distribution, $\mathbf{L}(\mu, 1)$, we have $Q(\mathbf{z}'_a \mid \mathbf{y}') = \prod_i Q((\mathbf{z}'_a)_i \mid \mathbf{y}') = \prod_i \mathbf{L}(\mu_i(\mathbf{y}'), 1)$. Under this assumption, maximizing the corresponding lower bound is equivalent to minimizing the information loss $\mathcal{L}_{\text{Info}}(E, G)$.

### A.5 LEMMA3

As in Kingma & Welling (2013), we derive the variational lower bounds presented in section 3.3 in the main text. Take $\log P(\mathbf{y})$ as an example and $\log P(\mathbf{x})$ has a similar derivation:

$$\log P(\mathbf{y}) = \log \int P(\mathbf{y}, \mathbf{z}_a, \mathbf{z}_c)\, d\mathbf{z}_a\, d\mathbf{z}_c$$

$$= \log \int P(\mathbf{y} \mid \mathbf{z}_a, \mathbf{z}_c) P(\mathbf{z}_a, \mathbf{z}_c) \frac{Q(\mathbf{z}_a, \mathbf{z}_c \mid \mathbf{y})}{Q(\mathbf{z}_a, \mathbf{z}_c \mid \mathbf{y})}\, d\mathbf{z}_a\, d\mathbf{z}_c$$

$$= \log \mathbb{E}_{(\mathbf{z}_a, \mathbf{z}_c) \sim Q(\mathbf{z}_a, \mathbf{z}_c|\mathbf{y})} \left[ \frac{P(\mathbf{y} \mid \mathbf{z}_a, \mathbf{z}_c) P(\mathbf{z}_a, \mathbf{z}_c)}{Q(\mathbf{z}_a, \mathbf{z}_c \mid \mathbf{y})} \right] \qquad (9)$$

$$\geq \mathbb{E}_{(\mathbf{z}_a, \mathbf{z}_c) \sim Q(\mathbf{z}_a, \mathbf{z}_c|\mathbf{y})} \log \frac{P(\mathbf{y} \mid \mathbf{z}_a, \mathbf{z}_c) P(\mathbf{z}_a, \mathbf{z}_c)}{Q(\mathbf{z}_a, \mathbf{z}_c \mid \mathbf{y})}$$

$$= \mathbb{E}_{(\mathbf{z}_a, \mathbf{z}_c) \sim Q(\mathbf{z}_a, \mathbf{z}_c|\mathbf{y})} \log P(\mathbf{y} \mid \mathbf{z}_a, \mathbf{z}_c) - D_{\text{KL}}\left(Q(\mathbf{z}_a, \mathbf{z}_c \mid \mathbf{y}) \,\|\, P(\mathbf{z}_a, \mathbf{z}_c)\right)$$

$$= \mathbb{E}_{(\mathbf{z}_a, \mathbf{z}_c) \sim Q(\mathbf{z}_a, \mathbf{z}_c|\mathbf{y})} \left[\log P(\mathbf{y} \mid \mathbf{z}_a, \mathbf{z}_c)\right] - D_{\text{KL}}\left(Q(\mathbf{z}_a \mid \mathbf{y}) \,\|\, P(\mathbf{z}_a)\right) - D_{\text{KL}}\left(Q(\mathbf{z}_c \mid \mathbf{y}) \,\|\, P(\mathbf{z}_c)\right)$$

Our reconstruction loss is therefore:

$$\mathcal{L}_{\text{recons}}(E, G) = \mathbb{E}_{\mathbf{x} \sim P(\mathbf{x}),\, \hat{\mathbf{z}}_c = E_c(\mathbf{x})} \left[ \|G(\mathbf{0}, \hat{\mathbf{z}}_c) - \mathbf{x}\|_1 \right]$$
$$+ \mathbb{E}_{\mathbf{y} \sim P(\mathbf{y}),\, (\hat{\mathbf{z}}_a, \hat{\mathbf{z}}_c) = E(\mathbf{y})} \left[ \|G(\hat{\mathbf{z}}_a, \hat{\mathbf{z}}_c) - \mathbf{y}\|_1 \right]. \qquad (10)$$

## A.6 PRIOR LOSS

In section 3.3, we have three KL divergence terms, one of which $D_{\mathrm{KL}}\big(q_\phi(\mathbf{z}_a|\mathbf{y}) \,\|\, \mathcal{U}[0,1]^{D_a}\big)$ has no closed-form solution, so we use Cramer-Wold (CW) distance to control the distance between distributions (Knop et al., 2020).

Let $I = I_c \cup I_d = \{1, \cdots, p\}$ be an index set of the variables, where $I_c$ and $I_d$ are the index sets of continuous and discrete variables, respectively. $\mathbf{s}_j \in \mathbb{R}$ for $j \in I_c$ and $\mathbf{s}_j \in \{0,1\}^{T_j}$ for $j \in I_d$, where $T_j$ denotes the number of levels, meaning that the discrete variables are transformed into a one-hot vector.

Let $\mathbf{s} = (\mathbf{s}_1, \cdots, \mathbf{s}_p) \in \mathbb{R}^D$ be an observation consisting of continuous and one-hot encoded discrete variables, and $D = |I_c| + \sum_{j \in I_d} T_j$. We denote the ground-truth underlying distribution (probability density function, PDF) as $p(\mathbf{s})$. Let $\mathbf{u}$ be a latent variable, where $\mathbf{u} \in \mathbb{R}^d$ and $d < D$. The prior distribution of $\mathbf{u}$ is assumed to be $p(\mathbf{u}) := \mathcal{N}(\mathbf{u}|\mathbf{0}, \mathbf{I})$ (standard Gaussian distribution), where $\mathbf{I}$ is $d \times d$ identity matrix. Let $\sigma_D$ be the normalized surface measure on $S_D$, where $S_D$ denotes the unit sphere in $\mathbb{R}^D$. For a sample $R = \{r_i\}_{i=1}^n \subset \mathbb{R}$, the smoothed distribution with a Gaussian kernel is defined as:

$$sm_\gamma(R) := \frac{1}{n} \sum_{i=1}^n N(r_i, \gamma),$$

which is a smoothen distribution with a Gaussian kernel $N(\cdot, \gamma)$ and $N(m, s)$ denotes the one-dimensional normal density with mean $m$ and variance $s$.

**Cramer-Wold distance.** The Cramer-Wold distance leverages the Cramer-Wold Theorem (Cramér & Wold, 1936) and the Radon Transform (Deans, 2007) to simplify the computation of the distance between two multivariate distributions into one-dimensional calculations. It shares similarity with the sliced-Wasserstein distribution (Deshpande et al., 2018). Notably, when each multivariate distribution is smoothed with a Gaussian kernel, the Cramer-Wold distance can be computed in a closed form without the need for sampling (Knop et al., 2020).

**Definition A.1.** *Let two PDFs $p(\mathbf{s})$ and $\hat{p}(\mathbf{s})$ be given, where $\mathbf{s} \in \mathbb{R}^D$. The Cramer-Wold distance $d_{CW}^2$ with $\sigma_D$ is defined as*

$$d_{CW}^2\big(p(\mathbf{s}), \hat{p}(\mathbf{s}); \sigma_D\big) := \int_{S_D} \big\|sm_\gamma(\nu^\top \mathbf{S}) - sm_\gamma(\nu^\top \hat{\mathbf{S}})\big\|_2^2 d\sigma_D(\nu),$$

*where $\nu \in S_D$, $\nu^\top \mathbf{S} := \{\nu^\top \mathbf{s}^{(i)}\}_{i=1}^n$ for $\mathbf{s}^{(i)} \sim p(\mathbf{s})$, and $\nu^\top \hat{\mathbf{S}} := \{\nu^\top \hat{\mathbf{s}}^{(i)}\}_{i=1}^n$ for $\hat{\mathbf{s}}^{(i)} \sim \hat{p}(\mathbf{s})$.*

Note that $\nu \in \mathbb{R}^p$ is a unit-norm projection vector such that $\|\nu\|_2 = 1$. The Cramer-Wold distance is assumed to be computed with finite $n$ samples. Additionally, in this section, we simplify notation by considering only continuous variables in explaining model descriptions and motivation without the loss of generality.

For the specific application of CW distance, following Knop et al. (2020), we can obtain the approximate analytical formula that expresses $d_{\mathrm{CW}}$ for a latent sample $U = \{u_i\}_{i=1,\ldots,n}$ and the standard Gaussian prior $\mathcal{N}(0, I)$. Specifically:

$$2\sqrt{\pi} d_{CW}^2(U, \mathcal{N}(0,I)) \approx \frac{1}{n^2} \sum_{ij} \left(\gamma_n + \frac{\|u_i - u_j\|^2}{2D_U - 3}\right)^{-\frac{1}{2}} + (1 + \gamma_n)^{-\frac{1}{2}}$$

$$- \frac{2}{n} \sum_i \left(\gamma_n + \frac{1}{2} + \frac{\|u_i\|^2}{2D_U - 3}\right)^{-\frac{1}{2}},$$

Following Knop et al. (2020), for two given samples $S = \{s_i\}_{i=1,\ldots,n}$ and $T = \{t_i\}_{i=1,\ldots,n}$ in domain $\Omega$, the approximate analytical formula expressing $d_{\mathrm{CW}}$ is:

$$2\sqrt{\pi} d_{CW}^2(S, T) \approx \frac{1}{n^2} \sum_{ij} \left(\gamma_n + \frac{\|s_i - s_j\|^2}{2D_\Omega - 3}\right)^{-\frac{1}{2}} + \frac{1}{n^2} \sum_{ij} \left(\gamma_n + \frac{\|t_i - t_j\|^2}{2D_\Omega - 3}\right)^{-\frac{1}{2}}$$

$$- \frac{2}{n^2} \sum_{ij} \left( \gamma_n + \frac{\|s_i - t_j\|^2}{2D_\Omega - 3} \right)^{-\frac{1}{2}},$$

where $D_\Omega = \dim \Omega$ and $\gamma_n$ is calculated using the standard deviation of joined $S$ and $T$ samples.

Therefore, we can define $\mathcal{L}_{\text{prior}}$ to constrain the distance between the learned posterior $q_\phi(\mathbf{z}_a, \mathbf{z}_c | \cdot)$ and the corresponding priors $P(\mathbf{z}_a, \mathbf{z}_c)$. We take the logarithm of the Cramer-Wold distance to improve balance as in Knop et al. (2020):

$$\mathcal{L}_{\text{prior}}(E_a, E_c) = \log\big(d_{CW}^2(q_\phi(\mathbf{z}_a | \mathbf{y}), \mathcal{U}(0,1))\big) + D_{\text{KL}}(q_\phi(\mathbf{z}_c | \mathbf{y}) \| \mathcal{N}(0,1)) \\ + D_{\text{KL}}(q_\phi(\mathbf{z}_c | \mathbf{x}) \| \mathcal{N}(0,1)) \tag{11}$$

### A.7 ATTRIBUTE SPACE REGULARIZATION

A continuous latent variable, by itself, does not guarantee interpretable or clinically meaningful patterns. To ensure that attribute space reflect plausible pathological processes, we constrain the function class of the generator $G$ and the encoder $E$ with several regularization terms, inspired by SurrealGAN (Yang et al., 2024). These constraints are designed to make the attribute factors $\mathbf{z}_a$ correspond to distinct, localized, and progressively interpretable changes in neuroimaging data.

**Sparsity.** To reflect the clinical fact that pathology usually affects localized regions, we impose an $\ell_1$ penalty to encourage sparse changes, ensuring that only a subset of brain regions is significantly altered.

$$\mathcal{L}_{\text{sparse}}(G) = \mathbb{E}_{\mathbf{z}_a \sim P_Y(\mathbf{z}_a), \mathbf{x} \sim P(\mathbf{x})}\big[\|G_{\text{diff}}(\mathbf{z}_a, \mathbf{x})\|_1\big] + \mathbb{E}_{\mathbf{z}_a, \mathbf{z}_c \sim P_Y(\mathbf{z}_a, \mathbf{z}_c)}\big[\|G_{\text{diff}}(\mathbf{z}_a, \mathbf{x}')\|_1\big] \tag{12}$$

**Pattern Separation.** We regularize $G_{\text{diff}}$ such that different components of $\mathbf{z}_a$ give rise to distinct and orthogonal patterns, ensuring each latent dimension $(\mathbf{z}_a)_i$ encodes a unique and identifiable anatomical variation. This is achieved by penalizing correlations between normalized change patterns. Let $\mathbf{d}^j = G_{\text{diff}}(\mathbf{a}^j, \boldsymbol{x})$, where $\mathbf{a}^j$ is a vector whose $j$-th entry $\mathbf{a}_j^j = (\mathbf{z}_a)_j$ and $\mathbf{a}_k^j = 0, \forall k \neq j$. We define the matrix $\boldsymbol{D}_{G_{\text{diff}}(\mathbf{z}_a, \mathbf{x})}$ whose $j$-th column is $(\boldsymbol{D}_{G_{\text{diff}}(\mathbf{z}_a, \mathbf{x})})_{:,j} = |\mathbf{d}^j|/\|\mathbf{d}^j\|_2$. The pattern separation loss is then given by:

$$\mathcal{L}_{\text{disen}}(G) = \mathbb{E}_{\mathbf{z}_a \sim P_Y(\mathbf{z}_a), \mathbf{x} \sim P(\mathbf{x})}\big[\|\boldsymbol{D}_{G_{\text{diff}}(\mathbf{z}_a, \mathbf{x})}^T \boldsymbol{D}_{G_{\text{diff}}(\mathbf{z}_a, \mathbf{x})} - \boldsymbol{I}\|_F\big] \\ + \mathbb{E}_{\mathbf{z}_a, \mathbf{z}_c \sim P_Y(\mathbf{z}_a, \mathbf{z}_c)}\big[\|\boldsymbol{D}_{G_{\text{diff}}(\mathbf{z}_a, \mathbf{x}')}^T \boldsymbol{D}_{G_{\text{diff}}(\mathbf{z}_a, \mathbf{x}')} - \boldsymbol{I}\|_F\big]. \tag{13}$$

**Monotonicity.** We hypothesize that as a disease pattern becomes more severe, the absolute value of the change in regional volume will only increase monotonically or remain constant. To enforce this property, we use a monotonicity loss that penalizes cases where a less severe latent variable value leads to larger anatomical changes than a more severe one. This ensures a positive correlation between the values in the latent variable and the severity of the generated pattern. Therefore, we sample another latent variable $\tilde{\mathbf{z}}_a \sim P(\tilde{\mathbf{z}}_a | \mathbf{z}_a)$ conditioned on $\mathbf{z}_a$, such that $(\tilde{\mathbf{z}}_a)_i \geq (\mathbf{z}_a)_i$ for all $1 \leq i \leq D_a$. The monotonicity loss is then defined as:

$$\mathcal{L}_{\text{mono}}(G) = \mathbb{E}_{\mathbf{z}_a \sim P_Y(\mathbf{z}_a), \tilde{\mathbf{z}}_a \sim P(\tilde{\mathbf{z}}_a | \mathbf{z}_a), \mathbf{x} \sim P(\mathbf{x})}\Big[\big\|\min\big(|G_{\text{diff}}(\tilde{\mathbf{z}}_a, \mathbf{x}) - G_{\text{diff}}(\mathbf{z}_a, \mathbf{x})|, 0\big)\big\|_2\Big] \\ + \mathbb{E}_{\mathbf{z}_a, \mathbf{z}_c \sim P_Y(\mathbf{z}_a, \mathbf{z}_c), \tilde{\mathbf{z}}_a \sim P(\tilde{\mathbf{z}}_a | \mathbf{z}_a)}\Big[\big\|\min\big(|G_{\text{diff}}(\tilde{\mathbf{z}}_a, \mathbf{x}') - G_{\text{diff}}(\mathbf{z}_a, \mathbf{x}')|, 0\big)\big\|_2\Big]. \tag{14}$$

To be appropriate for the constraint $G_{\text{diff}}(\mathbf{0}, \mathbf{x}') = \mathbf{0}$, We sample in the $\delta$–neighborhood of $\mathbf{0}$, $\mathbf{z}_a \sim \mathcal{U}[0, \delta]^{D_a} = P_\delta(\mathbf{z}_a)$. The background loss is then defined as:

$$\mathcal{L}_{\text{bg}}(G) = \mathbb{E}_{\mathbf{z}_a \sim P_\delta(\mathbf{z}_a), \mathbf{x} \sim P(\mathbf{x})}\big[\|G_{\text{diff}}(\mathbf{z}_a, \mathbf{x})\|_1\big] + \mathbb{E}_{\mathbf{z}_a \sim P_\delta(\mathbf{z}_a), \mathbf{z}_c \sim P_Y(\mathbf{z}_c)}\big[\|G_{\text{diff}}(\mathbf{z}_a, \mathbf{x}')\|_1\big]. \tag{15}$$

**Lipschitz Continuity.** We constrain the parameters of functions $\theta_E$ and $\theta_G$ to compact spaces $\Theta_E$ and $\Theta_G$ in order to ensure their $M$-Lipschitz continuity, where the Lipschitz constants $M_E$ and $M_G$

are determined by $\Theta_E$ and $\Theta_G$. Specifically, compactness is achieved by restricting the weight space to a fixed box $\Theta = [-c, c]^D$. In our implementation, we set $c = 0.5$ for both $\Theta_E$ and $\Theta_G$.

As an example, consider the encoder $E_a$ and the generator $G_{\text{diff}}$. When $G_{\text{diff}}$ is $M_{G_{\text{diff}}}$-Lipschitz continuous, we have that, for a fixed attribute variable $\mathbf{z}_a = \boldsymbol{a}$ and for any $\mathbf{x}'_1, \mathbf{x}'_2 \sim P_{\mathbf{z}_c \sim P_X(\mathbf{z}_c)}(G_{\text{bg}}(\mathbf{z}_c))$, $\|G_{\text{diff}}(\boldsymbol{a}, \mathbf{x}'_1) - G_{\text{diff}}(\boldsymbol{a}, \mathbf{x}'_2)\| \leq M_{G_{\text{diff}}}\|\mathbf{x}'_1 - \mathbf{x}'_2\|$.

Thus, by controlling the constant $M_{G_{\text{diff}}}$, the same mapping direction preserves the original distance between background data by transforming them into a compact set, but does not excessively disperse the synthesized target data.

Similarly, when $E_a$ is $M_{E_a}$-Lipschitz continuous, we can derive that when the information loss is close to zero, for any two distinct latent codes $(\mathbf{z}_a)_1, (\mathbf{z}_a)_2 \sim P_{\mathbf{y}}(\mathbf{z}_a)$ and any $\mathbf{x}' \sim P_{\mathbf{z}_c \sim P_{\mathbf{x}}(\mathbf{z}_c)}(G_{\text{bg}}(\mathbf{z}_c))$, the distance $\|G_{\text{diff}}((\mathbf{z}_a)_1, \mathbf{x}') - G_{\text{diff}}((\mathbf{z}_a)_2, \mathbf{x}')\|$ admits a constant lower bound $L_0$. This ensures that different attribute directions remain distinguishable in the generated data.

To see this more concretely, we consider $E_a$ and $G_{\text{diff}}$. When the information loss is close to zero, the following inequality holds:

$$\|G_{\text{diff}}(\mathbf{x}', (\mathbf{z}_a)_1) - G_{\text{diff}}(\mathbf{x}', (\mathbf{z}_a)_2)\| \geq \frac{1}{M_{E_a}} \|E_a(G_{\text{diff}}(\mathbf{x}', (\mathbf{z}_a)_1)) - E_a(G_{\text{diff}}(\mathbf{x}', (\mathbf{z}_a)_2))\|$$

$$\geq \frac{1}{M_{E_a}} \Big( \|(\mathbf{z}_a)_1 - (\mathbf{z}_a)_2\| - \|E_a(G_{\text{diff}}(\mathbf{x}', (\mathbf{z}_a)_1)) - (\mathbf{z}_a)_1\| - \|E_a(G_{\text{diff}}(\mathbf{x}', (\mathbf{z}_a)_2)) - (\mathbf{z}_a)_2\| \Big)$$

$$= \frac{\sqrt{2}}{M_{E_a}} - \frac{1}{M_{E_a}} \underbrace{\Big( \|E_a(G_{\text{diff}}(\mathbf{x}', (\mathbf{z}_a)_1)) - (\mathbf{z}_a)_1\| + \|E_a(G_{\text{diff}}(\mathbf{x}', (\mathbf{z}_a)_2)) - (\mathbf{z}_a)_2\| \Big)}_{\rightarrow 0} \quad (16)$$

$$= L_0.$$

Consequently, the proposed framework alleviates the mode collapse problem that is prone to occur in generative tasks. The difference between mapping directions is controlled to a non-trivial level; in other words, the same background data are mapped to significantly different target data when guided by distinct attribute codes.

**Decomposition.** We introduce a decomposition loss to enhance the semantic learning capability of the encoder $E_a$ by decomposing it into a composite mapping: $E_a = E_{a_2} \circ E_{a_1}$, where $E_{a_1}$ and $E_{a_2}$ are sub-encoders.

The decomposer $E_{a_1}$ is designed to reconstruct the changes synthesized by each component $\mathbf{d}^j$. For each latent dimension $i$, we define the change vector $\mathbf{d}^j = G_{\text{diff}}(\mathbf{a}^j, \boldsymbol{x})$, where $\mathbf{a}^j$ is a vector whose $j$-th entry satisfies $\mathbf{a}^j_j = (\mathbf{z}_a)_j$ and $\mathbf{a}^j_k = 0$ for all $k \neq j$, as in part Pattern Separation. We then concatenate these individual change vectors to form $\hat{\mathbf{d}}_{G_{\text{diff}}(\mathbf{z}_a, \mathbf{x})} = [\mathbf{d}_1^T, \mathbf{d}_2^T, \cdots, \mathbf{d}_{D_a}^T]^T$. The decomposition loss, which regularizes $E_{a_1}$ to accurately capture these individual changes, is defined as:

$$\mathcal{L}_{\text{decom}}(G_{\text{diff}}, E_{a_1}) = \mathbb{E}_{\mathbf{z}_a \sim P_Y(\mathbf{z}_a), \mathbf{x} \sim P(\mathbf{x})} \left[ \left\| E_{a_1}(\mathbf{x} + G_{\text{diff}}(\mathbf{z}_a, \mathbf{x})) - \hat{\mathbf{d}}_{G_{\text{diff}}(\mathbf{z}_a, \mathbf{x})} \right\|_2 \right]$$
$$+ \mathbb{E}_{\mathbf{z}_a, \mathbf{z}_c \sim P_Y(\mathbf{z}_a, \mathbf{z}_c)} \left[ \left\| E_{a_1}(G(\mathbf{z}_a, \mathbf{z}_c)) - \hat{\mathbf{d}}_{G_{\text{diff}}(\mathbf{z}_a, \mathbf{x}')} \right\|_2 \right] \quad (17)$$

The second sub-encoder $E_{a_2}$ further processes the output of $E_{a_1}$ to reconstruct each component of the original latent variable $\mathbf{z}_a$. Specifically, let $K$ denote the number of regions of interest (ROIs), then $E_a$ is given by:

$$E_a(G_{\text{diff}}(\mathbf{z}_a, \mathbf{x})) = \left[ E_{a_2}\left( E_{a_1}(G_{\text{diff}}(\mathbf{z}_a, \mathbf{x}))_{0:K} \right), \cdots, E_{a_2}\left( E_{a_1}(G_{\text{diff}}(\mathbf{z}_a, \mathbf{x}))_{K*(D_a-1):K*D_a} \right) \right]^T$$

This composite structure ensures that $E_a$ not only captures the overall variation encoded in $\mathbf{z}_a$ but also preserves the semantic meaning of each individual latent dimension, maintaining alignment with the pattern separation regularization described earlier.

## A.8 TRAINING DETAILS AND ARCHITECTURES

Table 2 displays network architectures of the encoders $E$, the generator $G$, and the discriminator $D$, where $K$ denotes the number of ROIs, $D_a$ and $D_c$ denote the dimension of $\mathbf{z}_a$ and $\mathbf{z}_c$.

All models were implemented in `PyTorch` (Lee et al., 2009). The training process for each iteration followed a two-step update scheme using a single mini-batch of data: First, update background discriminator ($D_{\text{bg}}$) and target discriminator ($D_{\text{tar}}$) with background/target samples; Second, update generators and encoders by jointly encoding samples to derive latent attribute ($\mathbf{z}_a$) and context ($\mathbf{z}_c$) factors, then synthesizing outputs.

We used the ADAM optimizer ($\beta_1 = 0.5$, $\beta_2 = 0.999$) (Kinga et al., 2015): learning rate was $2 \times 10^{-3}$ for generators/encoders and $2 \times 10^{-4}$ for discriminators (1/10 of the former). Weight decay was $5 \times 10^{-3}$ for discriminators and $2.5 \times 10^{-3}$ for generators/encoders. Gradient clipping (max norm = 100) was applied to prevent exploding gradients and stabilize training, with a fixed batch size of 300. In addition, to ensure the encoder and generator satisfy Lipschitz continuity, we applied weight clipping to their parameters. Specifically, all weights of $\theta_E$ and $\theta_G$ were restricted to lie within a compact box $[-c, c]^D$, with $c = 0.5$ in our implementation. This compactness constraint enforces boundedness of the functions, thereby maintaining their $M$-Lipschitz property.

Default hyperparameters for InfoSepGAN were set as follows: $\lambda_{\text{recon}} = 2$, $\lambda_{\text{info}(z_a)} = w_a * \lambda_{\text{info}} = 40$, $\lambda_{\text{info}(z_c)} = w_c * \lambda_{\text{info}} = 10$, $\lambda_{\text{decomp}} = 80$, $\lambda_{\text{sparse}} = 1$, $\lambda_{\text{bg}} = 6$, $\lambda_{\text{prior}} = 0.08$, $\lambda_{\text{mono}} = 500$, and $\lambda_{\text{disen}} = 0.6$. The definitions of $w_a$ and $w_c$ are given in Eq. 5. These settings were used consistently across both synthetic and real datasets. For the synthetic experiments, the latent dimensions were set to $D_a = 3$ and $D_c = 3$. For the real dataset, we explored different values of the key hyperparameters $D_a$ and $D_c$, and selected the optimal configuration based on model agreement (ACI) across multiple runs, resulting in the final choice of $D_a = 2$ and $D_c = 5$. As shown in Section B.5, the model remains robust under variations of these hyperparameters. The model was trained for at least 30,000 epochs. Model checkpoints were saved only when all three criteria were simultaneously satisfied, with thresholds measured in terms of the $\ell_2$ loss: a reconstruction loss below $2 \times 10^{-2}$, an attribute information loss below $5 \times 10^{-3}$, and a monotonicity loss below $5 \times 10^{-3}$. $D_a$ and $D_c$ denote the dimension of $\mathbf{z}_a$ and $\mathbf{z}_c$.

Table 2: Architecture of encoders, generators and discriminators

| | Layer | Input Size | Bias Term | leaky relu $\alpha$ | Output Size |
|---|---|---|---|---|---|
| **Encoders** | | | | | |
| $E_{a_1}$ | Leaky-Relu+Linear | $K * 1$ | Yes | 0.2 | $(D_a * K) * 1$ |
| | Reshape | $(D_a * K) * 1$ | NA | NA | $D_a * K$ |
| $E_{a_2}$ | Linear1+Leaky-Relu | $K * 1$ | Yes | 0.2 | $\lfloor K/2 \rfloor * 1$ |
| | Linear2+Leaky-Relu | $\lfloor K/2 \rfloor * 1$ | Yes | 0.2 | $\lfloor K/4 \rfloor * 1$ |
| | Linear3+Leaky-Relu | $\lfloor K/4 \rfloor * 1$ | Yes | 0.2 | $D_a * 1$ |
| $E_c$ | Linear1 | $K * 1$ | Yes | NA | $\lfloor K/2 \rfloor * 1$ |
| | Leaky-Relu+Linear2 | $\lfloor K/2 \rfloor * 1$ | Yes | 0.2 | $\lfloor K/4 \rfloor * 1$ |
| | Leaky-Relu+Linear3 (for $\mu$) | $\lfloor K/4 \rfloor * 1$ | Yes | 0.2 | $D_c * 1$ |
| | Leaky-Relu+Linear3 (for $\log \sigma^2$) | $\lfloor K/4 \rfloor * 1$ | Yes | 0.2 | $D_c * 1$ |
| **Generators** | | | | | |
| $G_{\text{bg}}$ | Linear1+Leaky-Relu | $D_c * 1$ | No | 0.2 | $\lfloor K/4 \rfloor * 1$ |
| | Linear2+Leaky-Relu | $\lfloor K/4 \rfloor * 1$ | No | 0.2 | $\lfloor K/2 \rfloor * 1$ |
| | Linear3 | $\lfloor K/2 \rfloor * 1$ | No | NA | $K * 1$ |
| $G_{\text{diff}}$ | Linear1+Leaky-Relu | $K * 1$ | No | 0.2 | $\lfloor K/2 \rfloor * 1$ |
| | Linear2+Leaky-Relu | $\lfloor K/2 \rfloor * 1$ | No | 0.2 | $\lfloor K/4 \rfloor * 1$ |
| | Linear3+Sigmoid | $D_a * 1$ | No | NA | $\lfloor K/4 \rfloor * 1$ |
| | Linear4+Leaky-Relu | $\lfloor K/4 \rfloor * 1$ | No | 0.2 | $\lfloor K/2 \rfloor * 1$ |
| | Linear4 | $\lfloor K/2 \rfloor * 1$ | No | NA | $K * 1$ |
| **Discriminators** | | | | | |
| $D_{\text{bg}}/D_{\text{tar}}$ | Linear1+Leaky-Relu | $K * 1$ | Yes | 0.2 | $\lfloor K/2 \rfloor * 1$ |
| | Linear2+Leaky-Relu | $\lfloor K/2 \rfloor * 1$ | Yes | 0.2 | $\lfloor K/4 \rfloor * 1$ |
| | Linear3 | $\lfloor K/4 \rfloor * 1$ | Yes | NA | $2 * 1$ |

## A.9 EVALUATION METRICS

In this section, we provide a detailed explanation of the evaluation metrics used in InfoSepGAN, namely the **Pattern-c-index (PCI)** and the **Pattern-agr-index (ACI)**.

**Permutation and Alignment.** For two independent runs producing attribute matrices $(\mathbf{z}_a)^1$ and $(\mathbf{z}_a)^2 \in \mathbb{R}^{D_a}$, a permutation search is performed to find the optimal alignment of dimensions:

$$\pi^* = \arg\max_{\pi \in \Pi} \frac{1}{D_a} \sum_{k=1}^{D_a} \mathrm{CI}\big((\mathbf{z}_{a,k})^1, (\mathbf{z}_{a,\pi(k)})^2\big),$$

where $\Pi$ is the set of permutation functions that reorder the $D_a$ dimensions, and $\mathrm{CI}(\cdot, \cdot)$ is the concordance index. Since the order of indices in the derived representation is not important, we can always reorder them to find the best matching. Therefore, without loss of generality, we can simply write $\mathbf{z}_{a,\pi(k)} = \mathbf{z}_{a,k}$ after alignment.

**Pattern-c-index (PCI).** The PCI quantifies how well the inferred attribute factors $\mathbf{z}_a$ recover the ground-truth patterns $\mathbf{s}$ in synthetic or semi-synthetic experiments. A permutation search is first applied between $\mathbf{z}_a$ and $\mathbf{s}$ to find the optimal one-to-one correspondence between dimensions. For each aligned dimension, compute the concordance index:

$$\mathrm{CI}_k = \mathrm{CI}(\mathbf{z}_{a,k}, s_k), \quad k = 1, \ldots, D_a.$$

Average across all $D_a$ dimensions:

$$\mathrm{PCI} = \frac{1}{D_a} \sum_{k=1}^{D_a} \mathrm{CI}_k.$$

PCI thus serves as the primary metric for assessing the accuracy of pattern recovery.

**Pattern-agr-index (ACI).** The ACI measures the agreement of inferred attribute factors across multiple independent runs, thereby quantifying the stability and reproducibility of the learned patterns. For two independent runs $(\mathbf{z}_a)^{(m)}$ and $(\mathbf{z}_a)^{(n)}$, after permutation alignment, the pairwise consistency is defined as:

$$\text{pairwise } \mathrm{CI}\big((\mathbf{z}_a)^{(m)}, (\mathbf{z}_a)^{(n)}\big) = \frac{1}{D_a} \sum_{k=1}^{D_a} \mathrm{CI}\big((\mathbf{z}_{a,k})^{(m)}, (\mathbf{z}_{a,k})^{(n)}\big).$$

For $M$ independent runs, the ACI of model $m$ is the average consistency with all others:

$$\mathrm{ACI}^{(m)} = \frac{1}{M-1} \sum_{\substack{n=1 \\ n \neq m}}^{M} \text{pairwise } \mathrm{CI}\big((\mathbf{z}_a)^{(m)}, (\mathbf{z}_a)^{(n)}\big).$$

A global measure can be defined as the average across all $M$ runs:

$$\mathrm{ACI} = \frac{1}{M} \sum_{m=1}^{M} \mathrm{ACI}^{(m)}.$$

We also validate that the ACI among multiple trained models is a good indicator of model accuracy for parameter selection (see Fig. 8), and thus can effectively guide hyperparameter tuning in real-world applications.

**Model selection using ACI.** The framework automatically determines the optimal hyperparameters by identifying the configuration that achieves the highest overall ACI. Given this configuration, the specific model whose $\mathbf{z}_a$ exhibits the highest agreement (quantified by $\mathrm{ACI}^{(m)}$) with all other repetitions is selected to produce the final Attrs.

**Pattern-Pearson-Correlations (PPC).** PPC quantifies the linear correlation between two sets of attribute patterns across corresponding dimensions. For two independent runs producing aligned attribute matrices $(\mathbf{z}_a)^1$ and $(\mathbf{z}_a)^2$, PPC is computed as the average Pearson correlation across all $D_a$ dimensions:

$$\text{PPC} = \frac{1}{D_a} \sum_{k=1}^{D_a} \rho\big((\mathbf{z}_{a,k})^1, (\mathbf{z}_{a,k})^2\big),$$

where $\rho(\cdot, \cdot)$ denotes the Pearson correlation.

**Pattern-Difference-Correlations (PDC).** PDC captures the agreement of pairwise differences across dimensions, providing a complementary measure of structural consistency. For aligned attribute matrices $(\mathbf{z}_a)^1$ and $(\mathbf{z}_a)^2$, PDC is defined as the average Pearson correlation over all pairs of dimensions $(i, j)$ with $i < j$:

$$\text{PDC} = \frac{2}{D_a(D_a - 1)} \sum_{i=1}^{D_a} \sum_{j=i+1}^{D_a} \rho\Big((\mathbf{z}_{a,i})^1 - (\mathbf{z}_{a,j})^1, (\mathbf{z}_{a,i})^2 - (\mathbf{z}_{a,j})^2\Big).$$

PPC and PDC thus extend the evaluation framework by quantifying both direct dimension-wise similarity and the preservation of inter-dimensional differences, respectively.

In a word, PCI and PPC provide metrics for evaluating the accuracy, while ACI and PDC provide complementary metrics for evaluating the reliability and structural consistency of the InfoSepGAN outputs, respectively.

# B APPENDIX: EXPERIMENTS

## B.1 SYNTHETIC EXPERIMENTS

**Data Generation Process.** To evaluate whether our model can robustly recover disease-related atrophy patterns under unbalanced confounding effects, we designed synthetic experiments. We generated baseline ROI data for 1,800 subjects (900 BG and 900 Pseudo-TG). Each subject had 100 ROI features, where each ROI value was sampled from a normal distribution: $v_{ij} \sim \mathcal{N}(1, 0.1)$, with $v_{ij}$ denoting the baseline volume of the $j$-th ROI for the $i$-th subject. On top of these baseline values, we simulated two types of effects: (i) disease-related atrophy (applied only to Pseudo-TG subjects) and (ii) confounding factors (applied to subsets of both groups).

Table 3: Synthetic experimental scenarios with varying confounding strengths, proportions, and patterns.

| Experi | Conf. | $\mathbf{c}_k^{\mathbf{bg}}$ | $\mathbf{c}_k^{\mathbf{tar}}$ | $\alpha_{\mathbf{bg}}$ | $\alpha_{\mathbf{tar}}$ | Pattern |
|---|---|---|---|---|---|---|
| Basic | N | - | - | - | - | - |
| Conf | Y | $\mathcal{U}[0,1]^2$ | $\mathcal{U}[0,1]^2$ | 33% | 33% | Non-overlap |
| Conf Ovlp | Y | $\mathcal{U}[0,1]^2$ | $\mathcal{U}[0,1]^2$ | 33% | 33% | Overlap |
| Conf Sev Ovlp | Y | $\mathcal{U}[0,1]^2$ | $\mathcal{U}[0,1]^2$ | 20% | 80% | Overlap |
| Conf DiffPop | Y | $\mathcal{U}[0,0.4]^2 \cup \mathcal{U}[0.7,0.8]^2$ | $\mathcal{U}[0.7,1]^2$ | 33% | 67% | Non-overlap |
| Conf Sev DiffPop | Y | $\mathcal{U}[0,0.4]^2$ | $\mathcal{U}[0.7,1]^2$ | 33% | 67% | Non-overlap |
| Conf Extreme | Y | - | $\mathcal{U}[0,1]^2$ | 0% | 100% | Non-overlap |

**Simulation of Disease-Related Atrophy.** For each Pseudo-TG subject $i$, we simulated three distinct atrophy patterns. Their severities were represented by a three-dimensional vector: $\mathbf{s}_i = (s_{i1}, s_{i2}, s_{i3}) \sim \mathcal{U}[0,1]^3$, where $s_{ik}$ indicates the severity of the $k$-th pattern. For each pattern $k$ and each ROI $j$ belonging to that pattern, the adjusted volume was computed as: $v_{ij} = v_{ij} - v_{ij} * s_{ik} * \epsilon_{ijk} * 0.3$, where $\epsilon_{ijk} \sim \mathcal{N}(1, 0.1)$ introduces stochastic variability. Each pattern involved 12 ROIs, with 4 ROIs shared between different patterns to reflect realistic pathological overlaps (see Fig. 5a).

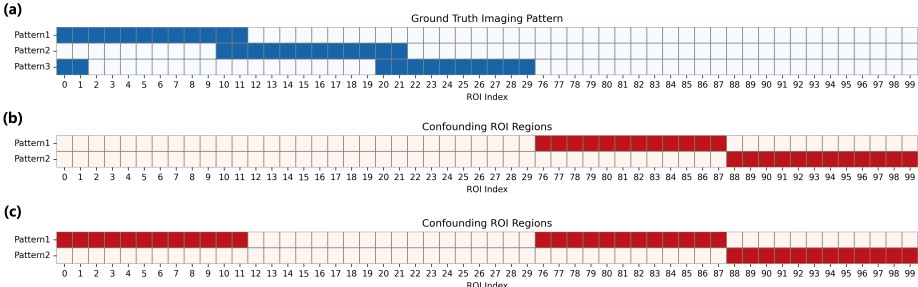

Figure 5: Illustration of synthetic experimental setups. (a) Three distinct simulated atrophy patterns. (b) Confounding pattern with no regional overlap with the atrophic regions. (c) Confounding pattern overlapping with the atrophic regions.

**Simulation of Confounding Factors.** Confounding effects were applied to random subsets of both BG and Pseudo-TG subjects. For each selected subject $i$, we sampled a confounding severity vector: $\mathbf{c}_i = (c_{i1}, c_{i2}) \sim U[0,1]^2$, where $c_{ik}$ denotes the severity of the $k$-th confounding pattern. For ROIs $j$ within the designated confounding area, the volumes were further modified as: $v_{ij} = v_{ij} - v_{ij} * c_{ik} * 0.3$. The confounding ROI area could overlap with the atrophic regions, thereby introducing additional ambiguity.

**Experimental Scenarios.** We constructed 7 representative scenarios by systematically varying three factors: (1) the confounding strengths $\mathbf{c}_k^{\mathrm{bg}}$ and $\mathbf{c}_k^{\mathrm{tar}}$; (2) the fractions of BG and TG subjects affected by confounding, $\alpha_{\mathrm{bg}}$ and $\alpha_{\mathrm{tar}}$; (3) the confounding patterns, either non-overlapping with or overlapping with atrophic regions (see Fig. 5b and Fig. 5c). Specifically, for each pattern, we calculated

a c-index between the inferred attribute factors and the ground truth $s_{ik}$. The detailed setups for each scenario are provided in Table 3, enabling systematic evaluation of model performance under different levels of confounding obscurity.

## B.2 SEMI-SYNTHETIC EXPERIMENTS

**UKBB Image Processing Pipeline.** The UK Biobank (UKBB) dataset included 63,618 T1-weighted 3D MRI volumes from participants aged 40 to 85 years. T1 image preprocessing (see also Alfaro-Almagro et al. (2018); Miller et al. (2016)) consisted of bias correction with FAST (Zhang et al., 2002), brain extraction using BET (Smith, 2002), and linear registration to MNI space via the FLIRT toolbox (Jenkinson et al., 2002). MRI and clinical data were obtained directly from the UKBB resource. To quantify regional brain volumes, we used imaging-derived phenotypes (IDPs) computed by the UK Biobank with FreeSurfer (Fischl, 2012). Specifically, we extracted volumetric measures from the FreeSurfer ASEG and DKT atlases, which together provide detailed segmentations of subcortical and cortical structures. Based on neuroanatomical relevance and consistency with widely used reference atlases, we retained 95 regions of interest (ROIs), excluding mainly non-brain parenchyma regions and global volume indicators, leaving primarily gray matter, white matter, and key subcortical nuclei. No additional segmentation or parcellation was applied beyond the UKBB-provided preprocessing.

We further identified individuals diagnosed with one or more chronic diseases from the UKBB, covering 14 categories: mild cognitive impairment/dementia, stroke, multiple sclerosis, hypertension, diabetes, depression, bipolar disorder, schizophrenia, Parkinson's disease, chronic obstructive pulmonary disease (COPD), osteoarthritis, chronic kidney disease (CKD), osteoporosis, and ischemic heart disease. Participants without these conditions were classified as the background group (BG). To establish ground-truth disease patterns and severity, we generated semi-synthetic data by introducing simulated disease-related atrophy into real healthy control (HC) data, thereby retaining the natural variation present in BG subjects. For each experiment, we sampled 1,200 HC individuals according to the experimental setting, then introduced artificial atrophy in 600 of them to create pseudo-TG data. Several semi-synthetic experiments were performed to assess the robustness of the improved InfoSepGAN model across multiple scenarios, making use of the remaining **30,858** HC individuals (age on average $63.39 \pm 6.12$).

Table 4: ROIs included in semi-synthetic patterns

| ROI | Default | | | Large overlapping | | | Scarce Region | | |
|---|---|---|---|---|---|---|---|---|---|
| | P0 | P1 | P2 | P0 | P1 | P2 | P0 | P1 | P2 |
| Amygdala | ✓ | | | ✓ | ✓ | | ✓ | | |
| Hippocampus | ✓ | | | ✓ | ✓ | | ✓ | | |
| Entorhinal | ✓ | | | ✓ | | | | | |
| Parahippocampal | ✓ | | | ✓ | | | | | |
| Medial orbitofrontal | ✓ | ✓ | | ✓ | | ✓ | | | |
| Superior frontal | ✓ | | ✓ | | | | | | |
| Lateral orbitofrontal | | ✓ | | | ✓ | | | | |
| Caudal middle frontal | | ✓ | | | | ✓ | | ✓ | |
| Rostral middle frontal | | ✓ | | | ✓ | | | | |
| Pars opercularis | | ✓ | ✓ | | | | | ✓ | |
| Pars triangularis | | ✓ | | | | | | | |
| Inferior temporal | | | ✓ | | | ✓ | | | |
| Inferior parietal | | | ✓ | ✓ | | ✓ | | | ✓ |
| Supramarginal | | | ✓ | | ✓ | ✓ | | | ✓ |
| Superior parietal | | | ✓ | | ✓ | ✓ | | | |

**Pattern-based Experiments.** We designed five experiments to evaluate robustness against varying disease-pattern complexities, focusing on participants aged 45 to 75 years: (1) **Basic**: Atrophy volumes were computed as $v_{ij} = v_{ij} - v_{ij} * s_{ik} * \mathcal{N}(1, 0.1) * 0.3$. Each pattern included 12 ROIs, with four overlapping across different patterns. (2) **Larger Noise**: Same as Basic, but noise standard deviation increased, i.e., $\mathcal{N}(1, 0.3)$. (3) **Larger Overlap**: Each pattern included 12 ROIs, with 8 ROIs overlapping across patterns. (4) **Mild**: Atrophy volumes were computed as $v_{ij} = v_{ij} - v_{ij} * s_{ik} * \mathcal{N}(1, 0.1) * 0.2$, reducing the effect size. (5) **Scarce**: Each pattern affected only

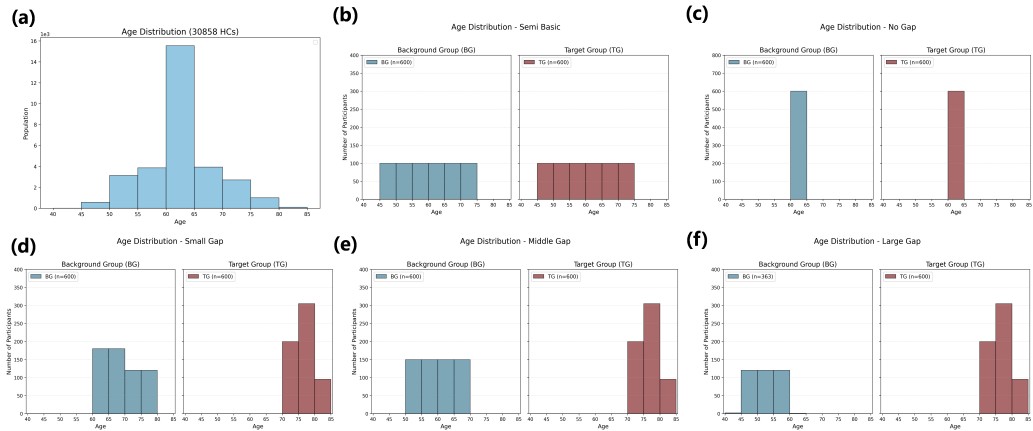

Figure 6: This figure presents age distributions across different datasets and experimental settings. (a) It depicts the age distribution of healthy control (HC) participants from real UK Biobank (UKBB) data; (b) It shows the age distribution histogram for Pattern-based Experiments; (c) to (f) illustrate age distributions for Age-confounding Experiments, where (c) is "No Age Gap", (d) is "Small Age Gap", (e) is "Middle Age Gap", and (f) is "Large Age Gap".

4 ROIs, with no overlap between patterns. All other settings matched the Basic scenario. Different regions were included in different patterns as shown in Table 4.

**Age-confounding Experiments.** We simulated unbalanced confounding effects by varying the degree of overlap between BG and pseudo-TG age distributions: (1) **No Age Gap**: Both BG and pseudo-TG ages ranged from 60 to 65 years. (2) **Small Age Gap**: BG ages ranged from 60 to 80 years, while pseudo-TG ages ranged from 70 to 85 years. (3) **Middle Age Gap**: BG ages ranged from 50 to 70 years, while pseudo-TG ages ranged from 70 to 85 years. (4) **Large Age Gap**: BG ages ranged from 40 to 60 years, while pseudo-TG ages ranged from 70 to 85 years. Due to the limited availability of 40–50-year-old subjects, only 363 BG samples were included in this setting. For all age-confounding experiments, atrophy volumes were computed as $v_{ij} = v_{ij} - v_{ij} * s_{ik} * \mathcal{N}(1, 0.1) * 0.3$, using a total of 1,200 individuals, with 600 per group.

In both experimental settings, analyses were conducted both with and without linear adjustment for age, sex, and intracranial volume (ICV), allowing us to examine the extent to which the model can inherently account for these confounding factors in the absence of explicit correction. The age distributions for each experimental setting are shown in Fig. 6.

### B.3 Data Preprocessing for Baseline Methods

To ensure fair comparison across different baseline models, we tailored the preprocessing steps to each method according to their assumptions and implementation requirements. For SurrealGAN, the data preprocessing procedure was identical to that of our InfoSepGAN framework, as both methods operate under the same generative modeling setting. We use the recommended hyperparameter settings for SurrealGAN, as specified in its official repository https://github.com/zhijian-yang/SurrealGAN. For cPCA (Abid et al., 2018), ROI-level features were first standardized to the range [0, 1] before model fitting. For classical dimensionality reduction approaches, we followed widely adopted practices in neuroimaging applications. Specifically, for the FA model, z-scored ROI volumes (with respect to the BG group) were directly used as input. For NMF, op-NMF, and LDA, we incorporated prior neurobiological knowledge that disease-related effects are predominantly characterized by regional atrophy. Accordingly, we inverted the sign of the z-scores, retained only the positive values, and set negative entries to zero, thus focusing on reductions in brain volume. For LDA, the features were further discretized by multiplying the transformed scores by 10 and mapping them to the nearest integer, following the procedure described in Zhang et al. (2016).

In addition, the general Contrastive Analysis (CA) family methods, including CAVAE (Aglinskas et al., 2022), DoubleInfoGAN (Carton et al., 2024), and MMCAVAE (Weinberger et al., 2022), were processed using the same pipeline as InfoSepGAN. This ensures that all CA-family models receive identical input features and preprocessing, allowing for a fair and controlled comparison of their performance in downstream analyses. The hyperparameters for all CA baseline models were set to their recommended values.

### B.4 BASELINE COMPARISONS AND GENERALIZATION TESTS

This section presents the results of all baseline models involved in the paper (each model was independently run 10 times for every experiment) and finally validates the generalization capability of InfoSepGAN.

**Synthetic Data Experiment** To comprehensively evaluate model performance under controlled synthetic settings, we conducted experiments across 7 representative scenarios (from balanced confounding to extreme unbalanced confounding). Table 5 summarizes the performance metrics (c-index) of all models, with values formatted as "mean ± standard deviation".

Table 5: Performance of All Models in Synthetic Data Experiments (c-index, mean ± std). **Bold** indicates the best mean performance, and $^{\#}$ indicates the second-best mean performance across all models. The statistical significance of the best versus the second-best model is denoted by asterisks: $^{***}p < 0.0001$, $^{**}p < 0.001$, $^{*}p < 0.05$, and $^{\text{n.s.}}$ (not significant).

| Experi Name | InfoSepGAN | SurrealGAN | NMF | OpNMF | FA | LDA | cPCA |
|---|---|---|---|---|---|---|---|
| Basic | $0.889^{\#}$ $\pm 0.002$ | $\mathbf{0.897}^{**}$ $\pm \mathbf{0.003}$ | $0.838$ $\pm 0.036$ | $0.803$ $\pm 0.000$ | $0.542$ $\pm 0.009$ | $0.753$ $\pm 0.016$ | $0.702$ $\pm 0.000$ |
| Conf | $\mathbf{0.891}^{\text{n.s.}}$ $\pm \mathbf{0.001}$ | $0.887^{\#}$ $\pm 0.030$ | $0.809$ $\pm 0.003$ | $0.806$ $\pm 0.000$ | $0.559$ $\pm 0.015$ | $0.613$ $\pm 0.031$ | $0.642$ $\pm 0.000$ |
| Conf Ovlp | $\mathbf{0.881}^{\text{n.s.}}$ $\pm \mathbf{0.001}$ | $0.873^{\#}$ $\pm 0.019$ | $0.784$ $\pm 0.035$ | $0.749$ $\pm 0.000$ | $0.625$ $\pm 0.042$ | $0.648$ $\pm 0.037$ | $0.721$ $\pm 0.000$ |
| Conf Sev Ovlp | $\mathbf{0.852}^{***}$ $\pm \mathbf{0.012}$ | $0.826^{\#}$ $\pm 0.010$ | $0.668$ $\pm 0.020$ | $0.687$ $\pm 0.000$ | $0.575$ $\pm 0.000$ | $0.631$ $\pm 0.023$ | $0.586$ $\pm 0.000$ |
| Conf DiffPop | $\mathbf{0.871}^{***}$ $\pm \mathbf{0.032}$ | $0.710^{\#}$ $\pm 0.014$ | $0.649$ $\pm 0.031$ | $0.680$ $\pm 0.000$ | $0.659$ $\pm 0.000$ | $0.609$ $\pm 0.016$ | $0.656$ $\pm 0.000$ |
| Conf Sev DiffPop | $\mathbf{0.835}^{***}$ $\pm \mathbf{0.044}$ | $0.6940^{\#}$ $\pm 0.018$ | $0.660$ $\pm 0.008$ | $0.652$ $\pm 0.000$ | $0.674$ $\pm 0.000$ | $0.611$ $\pm 0.016$ | $0.639$ $\pm 0.000$ |
| Conf Extreme | $\mathbf{0.711}^{**}$ $\pm \mathbf{0.047}$ | $0.622^{\#}$ $\pm 0.027$ | $0.563$ $\pm 0.010$ | $0.553$ $\pm 0.000$ | $0.447$ $\pm 0.000$ | $0.582$ $\pm 0.049$ | $0.601$ $\pm 0.000$ |

Key observations from Table 5 are as follows:
1. In balanced scenarios (Basic, Conf, Conf Ovlp), InfoSepGAN (0.881–0.891) and SurrealGAN (0.873–0.897) perform comparably, outperforming traditional methods (NMF: 0.784–0.838; FA: 0.542–0.625); 2. As confounding becomes unbalanced (Conf Sev Ovlp to Conf Extreme), InfoSepGAN maintains superior performance (0.711–0.852) compared to SurrealGAN (0.622–0.826) and all baselines.

**Semi-Synthetic Data Experiment** To further validate the model's performance in more realistic scenarios, we conducted semi-synthetic experiments (combining real UKBB data with simulated atrophy patterns) under two settings: *Pattern-based experiments* and *Age-confounding experiments*. Results are split into four tables (with/without linear adjustment for each experiment type), formatted as "mean ± std".

The semi-synthetic tables (Table 6–9) provide detailed numerical results across all scenarios. Consistent with the main text, InfoSepGAN generally outperforms SurrealGAN and classical baselines in both pattern-based and age-confounding settings. Performance differences are most pronounced under challenging conditions (e.g., Mild, Large Age Gap, or Scarce). Importantly, InfoSepGAN's superiority is preserved with and without covariate adjustment, underscoring its robustness to pre-processing choices, whereas SurrealGAN shows stronger dependence on explicit adjustment.

**Contrastive Analysis Baselines Results** We also benchmarked InfoSepGAN against three representative methods from the general Contrastive Analysis (CA) family: CAVAE (Abid & Zou, 2019;

Table 6: Pattern-based Semi-Synthetic Experiments (**with** linear adjustment, c-index, mean ± std). **Bold** indicates the best mean performance, and # indicates the second-best mean performance across all models. The statistical significance of the best versus the second-best model is denoted by asterisks: $^{***}p < 0.0001$, $^{**}p < 0.001$, $^{*}p < 0.05$, and $^{n.s.}$ (not significant).

| Experi Name | InfoSepGAN | SurrealGAN | NMF | OpNMF | FA | LDA | cPCA |
|---|---|---|---|---|---|---|---|
| Semi Basic | $0.851^{\#}$ $\pm 0.018$ | $\mathbf{0.855^{n.s.}}$ $\pm \mathbf{0.012}$ | 0.778 $\pm 0.013$ | 0.790 $\pm 0.000$ | 0.545 $\pm 0.000$ | 0.686 $\pm 0.047$ | 0.680 $\pm 0.000$ |
| Large Overlap | $\mathbf{0.819^{*}}$ $\pm \mathbf{0.016}$ | $0.773^{\#}$ $\pm 0.038$ | 0.713 $\pm 0.027$ | 0.739 $\pm 0.000$ | 0.475 $\pm 0.000$ | 0.561 $\pm 0.033$ | 0.438 $\pm 0.000$ |
| Large Noise | $0.831^{\#}$ $\pm 0.029$ | $\mathbf{0.841^{n.s.}}$ $\pm \mathbf{0.013}$ | 0.773 $\pm 0.001$ | 0.774 $\pm 0.000$ | 0.659 $\pm 0.000$ | 0.689 $\pm 0.042$ | 0.485 $\pm 0.000$ |
| Mild | $\mathbf{0.780^{***}}$ $\pm \mathbf{0.026}$ | 0.716 $\pm 0.026$ | 0.679 $\pm 0.041$ | $0.718^{\#}$ $\pm 0.000$ | 0.527 $\pm 0.000$ | 0.585 $\pm 0.033$ | 0.578 $\pm 0.000$ |
| Scarce | $\mathbf{0.774^{*}}$ $\pm \mathbf{0.009}$ | $0.746^{\#}$ $\pm 0.035$ | 0.723 $\pm 0.009$ | 0.715 $\pm 0.000$ | 0.574 $\pm 0.001$ | 0.552 $\pm 0.015$ | 0.517 $\pm 0.000$ |

Table 7: Pattern-based Semi-Synthetic Experiments (**without** linear adjustment, c-index, mean ± std). **Bold** indicates the best mean performance, and # indicates the second-best mean performance across all models. The statistical significance of the best versus the second-best model is denoted by asterisks: $^{***}p < 0.0001$, $^{**}p < 0.001$, $^{*}p < 0.05$, and $^{n.s.}$ (not significant).

| Experi Name | InfoSepGAN | SurrealGAN | NMF | OpNMF | FA | LDA | cPCA |
|---|---|---|---|---|---|---|---|
| Semi Basic | $\mathbf{0.851^{*}}$ $\pm \mathbf{0.027}$ | $0.802^{\#}$ $\pm 0.036$ | 0.744 $\pm 0.005$ | 0.749 $\pm 0.000$ | 0.585 $\pm 0.000$ | 0.622 $\pm 0.046$ | 0.624 $\pm 0.000$ |
| Large Overlap | $\mathbf{0.821^{**}}$ $\pm \mathbf{0.027}$ | $0.767^{\#}$ $\pm 0.033$ | 0.654 $\pm 0.001$ | 0.653 $\pm 0.000$ | 0.607 $\pm 0.000$ | 0.556 $\pm 0.031$ | 0.525 $\pm 0.000$ |
| Large Noise | $\mathbf{0.835^{*}}$ $\pm \mathbf{0.025}$ | $0.779^{\#}$ $\pm 0.043$ | 0.696 $\pm 0.052$ | 0.646 $\pm 0.000$ | 0.610 $\pm 0.001$ | 0.607 $\pm 0.035$ | 0.548 $\pm 0.000$ |
| Mild | $\mathbf{0.773^{***}}$ $\pm \mathbf{0.023}$ | $0.681^{\#}$ $\pm 0.013$ | 0.611 $\pm 0.007$ | 0.604 $\pm 0.000$ | 0.494 $\pm 0.000$ | 0.548 $\pm 0.018$ | 0.565 $\pm 0.000$ |
| Scarce | $\mathbf{0.773^{***}}$ $\pm \mathbf{0.012}$ | $0.705^{\#}$ $\pm 0.028$ | 0.664 $\pm 0.021$ | 0.643 $\pm 0.000$ | 0.516 $\pm 0.000$ | 0.521 $\pm 0.013$ | 0.517 $\pm 0.000$ |

Table 8: Age-confounding Semi-Synthetic Experiments (**with** linear adjustment, c-index, mean ± std). **Bold** indicates the best mean performance, and # indicates the second-best mean performance across all models. The statistical significance of the best versus the second-best model is denoted by asterisks: $^{***}p < 0.0001$, $^{**}p < 0.001$, $^{*}p < 0.05$, and $^{n.s.}$ (not significant).

| Experi Name | InfoSepGAN | SurrealGAN | NMF | OpNMF | FA | LDA | cPCA |
|---|---|---|---|---|---|---|---|
| No Age Gap | $\mathbf{0.857^{n.s.}}$ $\pm \mathbf{0.004}$ | $0.838^{\#}$ $\pm 0.028$ | 0.679 $\pm 0.006$ | 0.674 $\pm 0.000$ | 0.551 $\pm 0.000$ | 0.616 $\pm 0.045$ | 0.660 $\pm 0.000$ |
| Small Age Gap | $\mathbf{0.845^{n.s.}}$ $\pm \mathbf{0.010}$ | $0.815^{\#}$ $\pm 0.049$ | 0.767 $\pm 0.011$ | 0.778 $\pm 0.000$ | 0.608 $\pm 0.000$ | 0.663 $\pm 0.045$ | 0.489 $\pm 0.000$ |
| Middle Age Gap | $\mathbf{0.825^{*}}$ $\pm \mathbf{0.047}$ | $0.773^{\#}$ $\pm 0.061$ | 0.759 $\pm 0.011$ | 0.770 $\pm 0.000$ | 0.555 $\pm 0.000$ | 0.629 $\pm 0.055$ | 0.518 $\pm 0.000$ |
| Large Age Gap | $\mathbf{0.796^{***}}$ $\pm \mathbf{0.028}$ | 0.740 $\pm 0.035$ | 0.757 $\pm 0.011$ | $0.768^{\#}$ $\pm 0.000$ | 0.622 $\pm 0.000$ | 0.615 $\pm 0.059$ | 0.553 $\pm 0.000$ |

Aglinskas et al., 2022), DoubleInfoGAN (Carton et al., 2024), and MMCAVAE (Weinberger et al., 2022). These methods are primarily designed for information disentanglement, but lack specific inductive biases for modeling the progressive, monotonic, and sparse nature of pathological changes. To evaluate their applicability in disease pattern discovery, we tested these models on our semi-synthetic task using the PCI metric. As shown in Table 10, InfoSepGAN significantly outperformed all three CA baseline methods in all experimental scenarios.

The general CA models had PCI scores of approximately 0.49 to 0.58, indicating limited ability to spontaneously capture continuous monotonic progression of disease severity. Lacking explicit constraints such as monotonicity, sparsity and pattern separation, these models are fail to identify coherent pathological patterns. In contrast, InfoSepGAN consistently maintains a high PCI score.

Table 9: Age-confounding Semi-Synthetic Experiments (**without** linear adjustment, c-index, mean ± std). **Bold** indicates the best mean performance, and $^{\#}$ indicates the second-best mean performance across all models. The statistical significance of the best versus the second-best model is denoted by asterisks: $^{***}p < 0.0001$, $^{**}p < 0.001$, $^{*}p < 0.05$, and $^{\text{n.s.}}$ (not significant).

| Experi Name | InfoSepGAN | SurrealGAN | NMF | OpNMF | FA | LDA | cPCA |
|---|---|---|---|---|---|---|---|
| No Age | **0.855**$^{**}$ | 0.787$^{\#}$ | 0.739 | 0.725 | 0.588 | 0.590 | 0.546 |
| Gap | **± 0.023** | ± 0.035 | ± 0.015 | ± 0.000 | ± 0.000 | ± 0.049 | ± 0.000 |
| Small Age | **0.838**$^{**}$ | 0.770$^{\#}$ | 0.659 | 0.621 | 0.581 | 0.589 | 0.551 |
| Gap | **± 0.021** | ± 0.040 | ± 0.040 | ± 0.000 | ± 0.000 | ± 0.046 | ± 0.000 |
| Middle Age | **0.816**$^{***}$ | 0.711 | 0.712$^{\#}$ | 0.700 | 0.562 | 0.586 | 0.527 |
| Gap | **± 0.020** | ± 0.025 | ± 0.012 | ± 0.000 | ± 0.001 | ± 0.053 | ± 0.000 |
| Large Age | **0.759**$^{***}$ | 0.677$^{\#}$ | 0.645 | 0.654 | 0.572 | 0.585 | 0.537 |
| Gap | **± 0.021** | ± 0.021 | ± 0.010 | ± 0.000 | ± 0.000 | ± 0.044 | ± 0.000 |

This confirms that our specially designed regularization framework is crucial for transforming general disentanglement into clinically interpretable disease progression models.

Table 10: Performance of InfoSepGAN vs. Contrastive Analysis (CA) Baselines in Semi-Synthetic Experiments (**without** linear adjustment, c-index, mean ± std). **Bold** indicates the best mean performance, and $^{\#}$ indicates the second-best mean performance across all models. The statistical significance of the best versus the second-best model is denoted by asterisks: $^{***}p < 0.0001$, $^{**}p < 0.001$, $^{*}p < 0.05$, and $^{\text{n.s.}}$ (not significant).

| Experiment | InfoSepGAN | CAVAE | DoubleInfoGAN | MMCAVAE |
|---|---|---|---|---|
| Semi Basic | **0.851**$^{***}$ **± 0.027** | 0.532 ± 0.041 | 0.557$^{\#}$ ± 0.049 | 0.555 ± 0.028 |
| Large Noise | **0.835**$^{***}$ **± 0.025** | 0.492 ± 0.062 | 0.554$^{\#}$ ± 0.053 | 0.543 ± 0.032 |
| Mild | **0.773**$^{***}$ **± 0.023** | 0.517 ± 0.033 | 0.527 ± 0.021 | 0.528$^{\#}$ ± 0.021 |
| Large Overlap | **0.821**$^{***}$ **± 0.027** | 0.491 ± 0.068 | 0.537 ± 0.030 | 0.538$^{\#}$ ± 0.031 |
| Scarce | **0.773**$^{***}$ **± 0.012** | 0.531$^{\#}$ ± 0.014 | 0.525 ± 0.011 | 0.527 ± 0.022 |
| No Age Gap | **0.855**$^{***}$ **± 0.023** | 0.532 ± 0.042 | 0.583$^{\#}$ ± 0.063 | 0.575 ± 0.037 |
| Small Age Gap | **0.838**$^{***}$ **± 0.021** | 0.520 ± 0.063 | 0.537 ± 0.029 | 0.542$^{\#}$ ± 0.035 |
| Middle Age Gap | **0.816**$^{***}$ **± 0.020** | 0.534 ± 0.049 | 0.527 ± 0.028 | 0.547$^{\#}$ ± 0.048 |
| Large Age Gap | **0.759**$^{***}$ **± 0.021** | 0.509 ± 0.032 | 0.521 ± 0.022 | 0.547$^{\#}$ ± 0.042 |

**Generalization Evaluation of InfoSepGAN** To assess the generalization capability of InfoSep-GAN, we evaluated its performance on both training and independent test sets across synthetic, pattern-based semi-synthetic, and age-confounding semi-synthetic experiments. For the synthetic data, test sets were independently generated under the same simulation settings, each containing 200 subjects (100 BG and 100 TG). For the semi-synthetic data, test sets were constructed by sampling non-overlapping subsets of 200 individuals (100 BG and 100 TG) from the corresponding UKBB age ranges. This design ensured that train and test sets did not share samples, providing a more rigorous test of out-of-sample generalization.

Notably, InfoSepGAN learns attribute representations in an unsupervised manner, further underscoring the reliability of its generalization performance. Results are summarized in the following three tables (formatted as "mean ± std").

Key conclusions from generalization evaluation (Table 11-13). InfoSepGAN demonstrates strong and stable generalization across all settings, exhibiting minimal training-test set discrepancies, insensitivity to covariate manipulation, and robustness to imbalanced pathologically unrelated variation—highlighting its adaptability to unseen data and its superiority over previous models.

Table 11: Generalization of InfoSepGAN in Synthetic Experiments

| Experi Name | Train Set | Test Set |
|---|---|---|
| Basic | $0.889 \pm 0.002$ | $0.877 \pm 0.002$ |
| Conf | $0.891 \pm 0.001$ | $0.883 \pm 0.002$ |
| Conf Ovlp | $0.881 \pm 0.001$ | $0.876 \pm 0.002$ |
| Conf Sev Ovlp | $0.852 \pm 0.012$ | $0.852 \pm 0.013$ |
| Conf DiffPop | $0.871 \pm 0.032$ | $0.870 \pm 0.026$ |
| Conf Sev DiffPop | $0.835 \pm 0.044$ | $0.835 \pm 0.045$ |
| Conf Extreme | $0.711 \pm 0.047$ | $0.710 \pm 0.038$ |

Table 12: Generalization of InfoSepGAN in Pattern-based Semi-Synthetic Experiments

| Experi Name | Train (w. covar) | Test (w. covar) | Train (w.o. covar) | Test (w.o. covar) |
|---|---|---|---|---|
| Semi Basic | 0.851 $\pm$ 0.018 | 0.845 $\pm$ 0.028 | 0.851 $\pm$ 0.027 | 0.849 $\pm$ 0.024 |
| Large Overlap | 0.819 $\pm$ 0.016 | 0.819 $\pm$ 0.015 | 0.821 $\pm$ 0.027 | 0.820 $\pm$ 0.027 |
| Large Noise | 0.831 $\pm$ 0.029 | 0.822 $\pm$ 0.029 | 0.835 $\pm$ 0.025 | 0.824 $\pm$ 0.025 |
| Mild | 0.780 $\pm$ 0.026 | 0.776 $\pm$ 0.024 | 0.773 $\pm$ 0.023 | 0.769 $\pm$ 0.022 |
| Scarce | 0.774 $\pm$ 0.009 | 0.772 $\pm$ 0.018 | 0.773 $\pm$ 0.012 | 0.773 $\pm$ 0.014 |

Table 13: Generalization of InfoSepGAN in Age-confounding Semi-Synthetic Experiments

| Experi Name | Train (w. covar) | Test (w. covar) | Train (w.o. covar) | Test (w.o. covar) |
|---|---|---|---|---|
| No Age Gap | 0.857 $\pm$ 0.004 | 0.852 $\pm$ 0.009 | 0.855 $\pm$ 0.023 | 0.843 $\pm$ 0.028 |
| Small Age Gap | 0.845 $\pm$ 0.010 | 0.843 $\pm$ 0.014 | 0.838 $\pm$ 0.021 | 0.836 $\pm$ 0.025 |
| Middle Age Gap | 0.825 $\pm$ 0.047 | 0.823 $\pm$ 0.048 | 0.816 $\pm$ 0.020 | 0.815 $\pm$ 0.023 |
| Large Age Gap | 0.796 $\pm$ 0.028 | 0.793 $\pm$ 0.030 | 0.759 $\pm$ 0.021 | 0.753 $\pm$ 0.028 |

## B.5 ABLATION & ROBUSTNESS

We evaluated both ablation and robustness on two representative settings: the *Semi Basic* (easy case) and the *Middle Age Gap* (hard case). In all cases, the model was trained 10 times, and we reported average pattern-c-index (PCI).

**Ablation.** To assess the contribution of each core component, we performed ablation by setting key hyperparameters to zero or removing specific modules while keeping all other parameters fixed. Tested configurations include: $\lambda_{\text{recon}} = 0$ (no Recon), $\lambda_{\text{bg}} = 0$ (no Bg), $\lambda_{\text{prior}} = 0$ (no Prior), $\lambda_{\text{mono}} = 0$ (no Mono), $\lambda_{\text{disen}} = 0$ (no Disen), $\lambda_{\text{info}} = 0$ (no Two), and removing the context submodule (no Context, reducing InfoSepGAN to a "1-to-k" mapping framework). Fig. 7 shows that eliminating any single term consistently degrades performance. Notably, removing either $\lambda_{\text{disen}}$ or $\lambda_{\text{info}}$ leads to a marked drop in PCI, confirming their central role in ensuring accurate factor representation. Moreover, excluding the context submodule leads to significant performance degradation in the case of uneven distribution of age (Middle Age Gap), highlighting its importance in adaptively mitigating the effects of confounding and maintaining generalization across heterogeneous scenarios.

**Robustness.** We further evaluated the sensitivity of InfoSepGAN to hyperparameter variations by scaling each of the six core terms from 50% to 150% of their default values (section A.8), while keeping the others fixed. Across both the easy and hard settings (Fig. 8), the model generally preserved stable performance, indicating strong robustness. In the more challenging scenario, the parameter $\lambda_{\text{info}(z_a)}$ showed relatively higher variations in performances. Importantly, the agreement

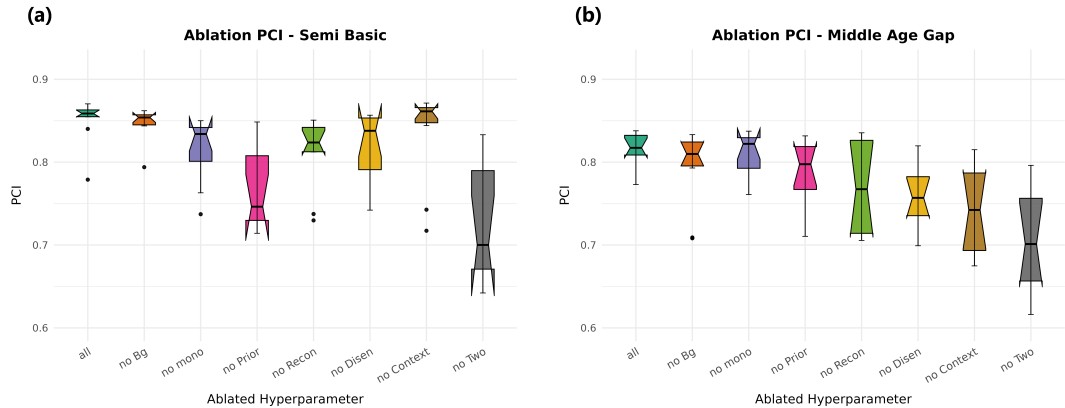

Figure 7: Impact of removing individual regularization terms on representation accuracy (removing the context sub-module also eliminates all associated context-related losses).

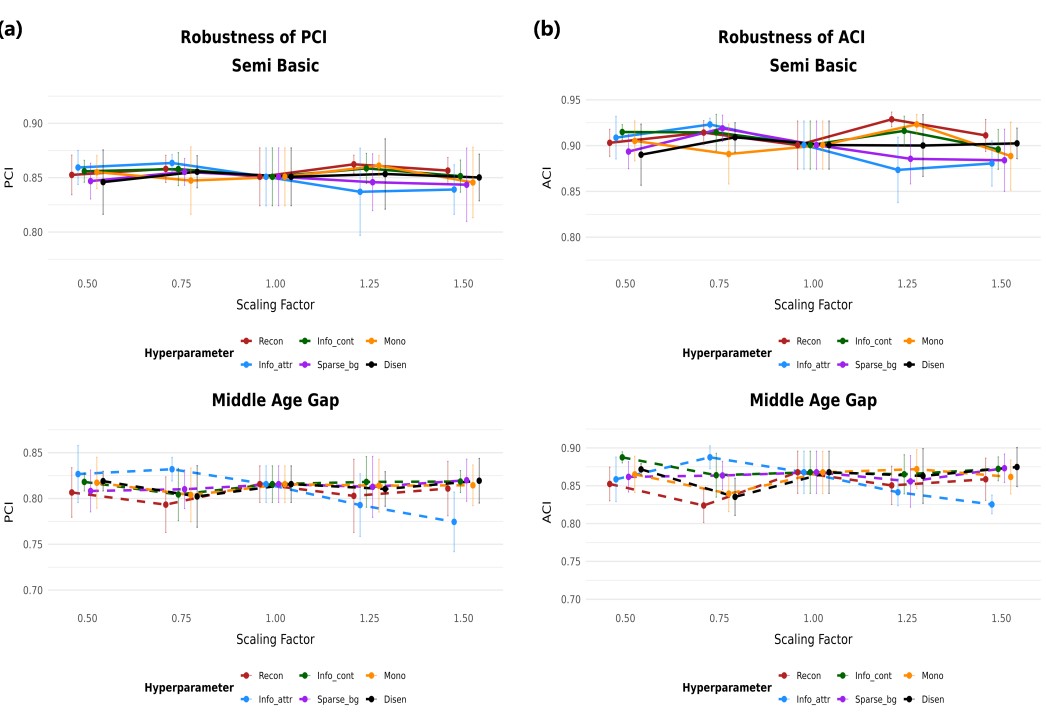

Figure 8: Robustness to hyperparameter scaling. (a) Representation accuracy (PCI) under varying hyperparameters; (b) Agreements among repetitively trained models (ACI) on different cases under the same settings. Results are reported as mean ± standard error.

index (ACI) consistently provided a reliable guide for model selection, suggesting that it can be effectively used to tune hyperparameters in real-world datasets.

### B.6 REAL DATA PREPROCESSING

We collected structural MRI (T1-weighted, T1w) data from the Alzheimer's Disease Neuroimaging Initiative (ADNI) database and implemented a systematic preprocessing pipeline, resulting in a final cohort of 2,587 participants (Cognitively Normal (CN) as BG ($N = 988$); Mild Cognitive Impair-

ment (MCI) or Alzheimer's Disease (AD) as TG ($N = 1599$). Data processing diagram is shown in Fig. 9.

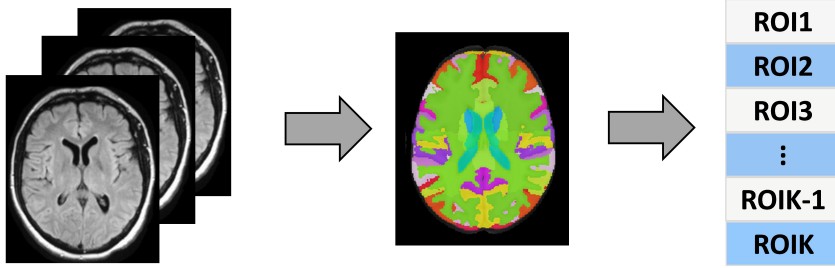

Figure 9: Procedure for deriving InfoSepGAN input features (ROI volumes): Structural MRI scans were preprocessed and skull-stripped using ANTs. Regional brain volumes (ROIs) were then extracted from segmented gray and white matter structures, which served as the input features for InfoSepGAN.

**T1w image preprocessing.** First, all baseline MRIs were corrected for intensity inhomogeneities (Sled et al., 2002), followed by skull-stripping with ANTs (Avants et al., 2009). Next, anatomical regions of interest (ROIs) were segmented using the multi-atlas label fusion framework (MUSE) (Doshi et al., 2016), producing 139 ROIs (119 gray matter, 20 white matter). Symmetric ROIs from the left and right hemispheres were merged to yield 72 volumetric features as input to InfoSepGAN.
**Quality control.** We adopted a result-variable-guided QC process to identify and exclude low-quality cases, focusing on key structural metrics and regional volumes. The QC targeted deep brain structures, subcortical parcellations, and ICV. For each dataset, volumetric measures were ICV-corrected and z-score normalized. Outliers were defined as ROI volumes deviating more than three standard deviations from the study mean. All scans containing at least one anomalous ROI were flagged for manual inspection to confirm low-quality data before exclusion.
**Voxel-wise RAVENS maps.** Voxel-wise regional volumetric maps (RAVENS) (Davatzikos et al., 2001) for gray and white matter were generated by registering skull-stripped images to a single-subject brain template using a deformable alignment method (Ou et al., 2011).
**Site harmonization.** Given the multi-site nature of ADNI, scanner-related biases were corrected using Combat-GAM (Pomponio et al., 2020). Parameters for harmonization (mean and variance) were estimated from the BG subset and then applied to the full cohort (including MCI/AD subjects), controlling for key covariates such as age, sex, and ICV.
**Feature standardization.** Finally, volumetric features of both BG and TG groups were standardized relative to the CN baseline. This alignment ensured that subsequent analyses emphasized deviations associated with disease progression rather than population variability.
**VBM visualization analysis.** To further localize disease-related atrophy, we performed voxel-based morphometry (VBM) analyses using the Python package Nilearn (Abraham et al., 2014). Specifically, we tested voxel-wise associations between regional tissue density and each attribute factor, while controlling for age, sex, intracranial volume (ICV), and the remaining one attribute factor. Multiple comparison correction was applied using the Benjamini–Hochberg procedure, with the false discovery rate (FDR) controlled at 0.05.

### B.7 REPRODUCIBILITY ON REAL DATASET

To validate the reproducibility of InfoSepGAN's results on real-world data, we first performed a random half-split of the ADNI dataset (described in Section 4.1) into training and test sets, ensuring equal distribution of cognitively normal (BG) and patient (TG) samples across both sets to avoid class imbalance bias. The experiment followed the exact same procedure as real-data analysis in main text: we trained InfoSepGAN on the training set (with $D_a = 2, D_c = 5$ as determined in section 5.3) and then applied the trained model to the test set to infer the two-dimensional attribute factors ($\mathbf{z}_a$) for all BG and TG samples. For visualization of the associated gray matter atrophy patterns, we conducted voxel-wise t-tests in the test set, adjusting for age, sex, ICV and the remaining one attribute factor.
Results (Fig. 10) demonstrate that the atrophy patterns derived from the test set are highly consistent

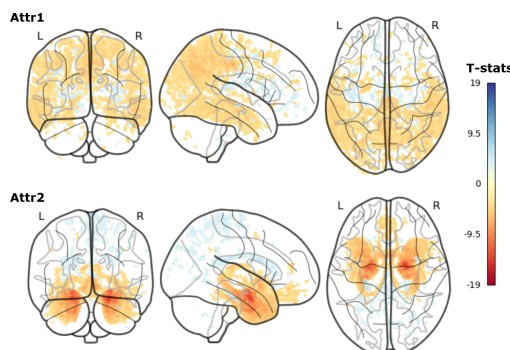

Figure 10: Reproducibility analysis on real-world ADNI data. The ADNI dataset was randomly split into training and test sets, and InfoSepGAN was trained on the former and evaluated on the latter. The patterns are shown via voxel-wise t-tests performed for each attribute factor while controlling for age, sex, intracranial volume (ICV), and the other attribute factor. Results are FDR-corrected at $p < 0.05$.

with those presented in the main text (Fig. 4). Although the smaller sample size of the test set led to a slight reduction in statistical power, the core pathological patterns linked to the attribute factors were reliably reproduced—confirming the stability and reproducibility of InfoSepGAN's results on real clinical data.

### B.8 External Dataset experiment

To further assess the generalizability of InfoSepGAN beyond ADNI, we conducted two external validation experiments using the UK Biobank (UKBB). Prior to analysis, both ADNI and UKBB images were harmonized using ComBat-GAM (Pomponio et al., 2020) to remove cohort effects while preserving biological variability.

**Cross-dataset reproducibility.** The UKBB dataset lacks AD-specific phenotypic information, such as diagnostic labels, CSF, or PET measures, making it challenging to define a control set free of AD pathology or a clear AD pathology cohort. However, as InfoSepGAN is a robust semi-supervised method, the control group only needs to be "generally healthy," as the model results are not significantly affected by a small minority of subjects exhibiting pathological aging.

To assess cross-dataset reproducibility, we first defined a rigorously selected healthy control in the UKBB dataset based on the exclusion criteria established in Chen et al. (2023a). These criteria included: not carrying the Apoe-E4 allele, no family history of dementia, no reported neurological or psychiatric disorders, and good performance on AD-related cognitive tests. We then randomly selected **1000** individuals from this control group as an independent reference group for retraining. The target group (TG) was defined as patients with mild cognitive impairment (MCI) or Alzheimer's disease (AD) from the ADNI cohort. Subsequently, we retrained InfoSepGAN 10 runs independently following the steps described in Section 4.2.

We compared the newly derived latent representations from this cross-cohort training with the original representations (Fig. 4a) using the four core pattern evaluation metrics: Pattern-c-index (PCI), Pattern-agr-index (ACI), Pattern-Pearson-Correlations (PPC), and Pattern-Difference-Correlations (PDC). Qualitatively, the atrophy maps derived from the retrained model (Fig. 11) consistently replicated the original findings: Attr1 reflected diffuse cortical atrophies across parietal, occipital, and temporal regions, and Attr2 highlighted focal medial temporal and subcortical atrophies. Quantitatively, an external reference group from UKBB in model retraining did not yield significant changes in identified patterns (**PCI: 0.774 ± 0.041; PPC: 0.819 ± 0.012; ACI: 0.806 ± 0.031; PDC: 0.403 ± 0.053**). This confirms that InfoSepGAN's ability to identify meaningful and stable disease patterns is robust to the choice of the healthy control reference cohort, thereby validating the model's reproducibility across distinct datasets.

**Application to asymptomatic individuals.** To further explore the cognitive relevance of the learned attribute factors in a broad asymptomatic population, we applied the model trained above to the general population of UKBB (N = 47,004), which excluded individuals who self-reported having neurological or psychiatric disorders (as defined in Chen et al. (2023a)).

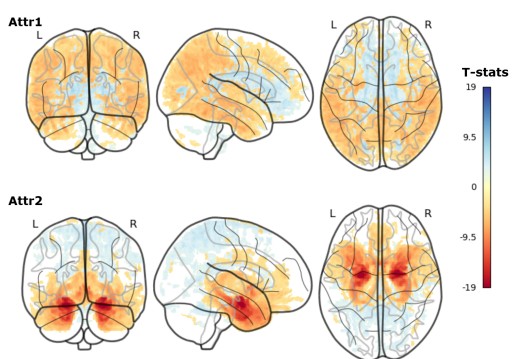

Figure 11: Cross-dataset reproducibility analysis using UKBB as an external reference cohort. InfoSepGAN was retrained using 1000 randomly selected healthy controls from UKBB and the MCI/AD group from ADNI. These patterns are shown via voxel-wise t-tests performed for each attribute factor while controlling for age, sex, intracranial volume (ICV), and the other attribute factor. Results are FDR-corrected at $p < 0.05$.

For each attribute dimension of $\mathbf{z}_a$ (Attr1 and Attr2), we evaluated its statistical association with available cognitive variables and WMH volumes in UKBB. Partial correlations (two-sided) were computed while adjusting for age and sex. From the cognitive tests available in UKBB, we focused on three domains: reaction time (RT), executive function (Trail Making Test A and B, TMT-A/B), and delayed memory (Prospective Memory Test, PMT), in addition to the Digit Symbol Substitution Test (DSST).

Table 14: Association between InfoSepGAN's attribute factors and cognitive variables and WMH in the UK Biobank general population. Significant $p$-values after Bonferroni correction are highlighted. (n.s. indicates not significant)

| Feature | Attr1 | Attr2 |
|---|---|---|
| WMH | 0.0569 (p < 0.0001) | 0.1344 (p < 0.0001) |
| Reaction Time (RT) | 0.0214 (p < 0.0001) | 0.0274 (p < 0.0001) |
| TMT-B | 0.0217 (p = 0.0002) | 0.0192 (p = 0.0017) |
| TMT-A | 0.0185 (p = 0.0029) | 0.0101 (n.s.) |
| DSST | -0.0234 (p = 0.0034) | -0.0126 (n.s.) |
| Prospective Memory | 0.0142 (p = 0.0201) | 0.0235 (p < 0.0001) |

Attr1 and Attr2 exhibits correlation with WMH, which is commonly associated with vascular risk factors and cognitive decline (Prins & Scheltens, 2015), and has also been shown to serve as an early predictor of AD risk (Mortamais et al., 2014). Additionally, significant correlations were found with several cognitive variables, particularly those measuring processing speed (RT, DSST), prospective memory (PMT) and executive function (TMT-B), which are well-established cognitive domains affected in the earliest, pre-clinical stages of dementia.

In summary, external validation results on UKBB confirm the generalization ability of the InfoSepGAN framework, demonstrating that the learned neuroanatomical patterns reflect meaningful clinical biological variations across different cohorts.

## B.9 ASSOCIATION WITH CLINICAL VARIABLES

**Partial Correlation Analysis.** For each attribute dimension of $\mathbf{z}_a$, we evaluated its statistical associations with clinical and biomarker variables using the target data (TG) from ADNI. Partial correlations (two-sided) were computed while adjusting for age and sex. The clinical variables included CSF biomarkers (CSF-Abeta42, CSF-Tau, CSF-PTau), cognitive scores (MMSE, TMT-A, TMT-B, DSST, ADNI-MEM, ADNI-EF, ADNI-LAN, ADNI-VS), years of education, and WMH volumes. ApoE-E4 allele carrier status was included as a dichotomous genetic risk factor, with point-biserial correlations applied in this case. All other variables were treated as continuous measures. To account for multiple comparisons, the false discovery rate was controlled at 5% using the

Benjamini–Hochberg procedure for CSF/plasma biomarkers due to their small sample sizes, while Bonferroni correction was applied for all other variables to control the family-wise error rate. These variables are widely recognized markers of Alzheimer's disease progression and are available in ADNI. Results (Table 15) reveal that the two attribute dimensions exhibit significant associations with multiple AD-related biomarkers and cognitive scores, supporting their clinical interpretability.

Table 15: Associations between AD pattern severity and clinical measures. (n.s. indicates not significant)

| Feature | Attr1 | Attr2 |
|---|---|---|
| CSF-Abeta | **-0.3456 (p $<$ 0.0010)** | **-0.2841 (p $<$ 0.0010)** |
| CSF-Tau | **0.1626 (p = 0.0022)** | 0.0321 (n.s.) |
| CSF-PTau | **0.1785 (p = 0.0010)** | 0.0239 (n.s.) |
| MMSE | **-0.4072 (p $<$ 0.0010)** | **-0.3582 (p $<$ 0.0010)** |
| TMT-A | **0.2281 (p $<$ 0.0010)** | **0.1340 (p = 0.0015)** |
| TMT-B | **0.2414 (p $<$ 0.0010)** | **0.1848 (p $<$ 0.0010)** |
| DSST | **-0.2351 (p $<$ 0.0010)** | **-0.1223 (p = 0.0215)** |
| ADNI-MEM | **-0.4350 (p $<$ 0.0010)** | **-0.4262 (p $<$ 0.0010)** |
| ADNI-EF | **-0.3089 (p $<$ 0.0010)** | **-0.2020 (p $<$ 0.0010)** |
| ADNI-LAN | **-0.3113 (p $<$ 0.0010)** | **-0.2795 (p $<$ 0.0010)** |
| ADNI-VS | **-0.1862 (p $<$ 0.0010)** | **-0.1328 (p = 0.0063)** |
| Apoe-E4 Alleles Carrier | **0.2357 (p $<$ 0.0010)** | **0.1983 (p $<$ 0.0010)** |
| Education years | -0.0328 (n.s.) | -0.0248 (n.s.) |
| WMH volumes | **0.1791 (p $<$ 0.0010)** | **0.0965 (p = 0.0139)** |

**Kendall $\tau$ Correlation Analysis.** Following Aglinskas et al. (2022), we further applied representational similarity analysis (RSA) to probe the disentanglement between attribute and context features. Specifically, we trained 10 independent InfoSepGAN models and, for each, constructed representational dissimilarity matrices (RDMs) based on attribute and context embeddings separately. These RDMs were then compared with dissimilarity matrices derived from clinical and nonclinical variables using Kendall's rank correlation coefficient. Statistical significance was evaluated by two-sided one-sample $t$-tests (whether correlations were greater than zero) and paired $t$-tests (comparing attribute vs. context).

Table 16: Supplementary RSA analyses for InfoSepGAN.

| Feature | N | Attribute $>$ Context |
|---|---|---|
| CSF-Tau | 628 | $\Delta\tau$=0.0629, $t(9)$=21.0577, **p $<$ .0001** |
| CSF-PTau | 626 | $\Delta\tau$=0.0828, $t(9)$=23.5879, **p $<$ .0001** |
| DSST | 608 | $\Delta\tau$=0.0445, $t(9)$=6.8524, **p $<$ .0001** |
| ADNI-MEM | 1173 | $\Delta\tau$=0.1400, $t(9)$=27.1296, **p $<$ .0001** |
| ADNI-EF | 1170 | $\Delta\tau$=0.0837, $t(9)$=13.8592, **p $<$ .0001** |
| ADNI-LAN | 1173 | $\Delta\tau$=0.1039, $t(9)$=23.0180, **p $<$ .0001** |
| Chronological Age | 2587 | $\Delta\tau$=-0.0049, $t(9)$=-1.4082, p = 0.1927 |
| Sex | 2587 | $\Delta\tau$=-0.0542, $t(9)$=-21.7755, **p $<$ .0001** |
| ICV | 2587 | $\Delta\tau$=-0.0667, $t(9)$=-11.8150, **p $<$ .0001** |
| Scanner Type | 2587 | $\Delta\tau$=-0.0224, $t(9)$=-8.3143, **p $<$ .0001** |
| Site | 2587 | $\Delta\tau$=-0.0155, $t(9)$=-17.4878, **p $<$ .0001** |
| ApoE-E4 | 2128 | $\Delta\tau$=0.0720, $t(9)$=33.8263, **p $<$ .0001** |
| WMH volumes | 1865 | $\Delta\tau$=0.0849, $t(9)$=15.5428, **p $<$ .0001** |

Results (Table 16) show that context features align more strongly with nonclinical variation shared across TG and BG participants (e.g., sex, site, scanner type, ICV), whereas attribute features capture AD-specific variation, including ApoE-E4 genotype, CSF biomarkers (e.g. CSF-Tau, CSF-PTau) and cognitive impairment (e.g. ADNI-MEM, ADNI-EF, ADNI-LAN). Although scanner type is primarily captured by the context features, we also observe a small but statistically significant correlation within the attribute space. The residual association between attribute features and scanner type may stem from the non-random distribution of diagnostic groups across different MRI protocols and scanner types in ADNI. Since later-phase, predominantly 3T scanners tend to include more MCI/AD subjects than earlier 1.5T protocols, scanner type becomes weakly informative of disease severity, making complete removal of scanner-related variance challenging. Importantly,

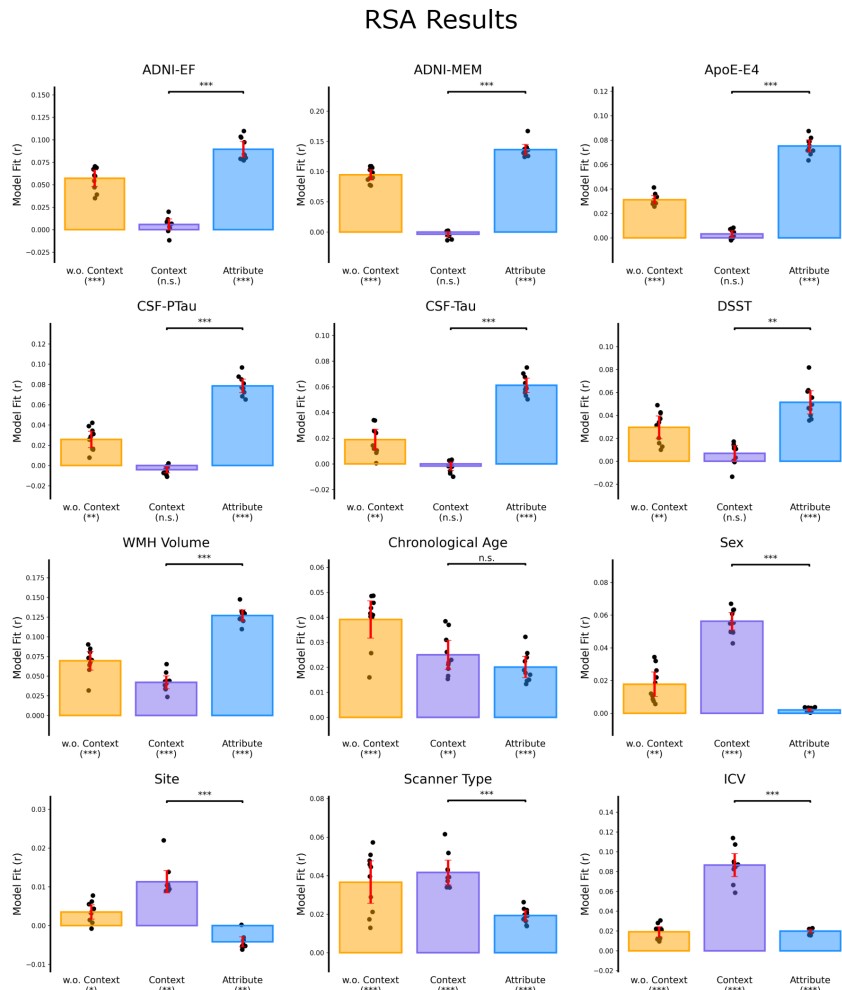

Figure 12: Representational Similarity Analysis (RSA) results (ADNI dataset). Bar plots show representational similarity between attribute/context features from InfoSepGAN, as well as attribute features without the context encoder $E_c$ (w.o. Context, a "1-to-$k$" mapping version of InfoSepGAN, only have attribute encoder $E_a$), and clinical or non-clinical variables. Red vertical lines indicate 95% confidence intervals. Statistical significance is indicated by stars (* $p < 0.05$, ** $p < 0.001$, *** $p < 0.0001$).

these residual effects are consistently much weaker than those encoded by the context features, and paired comparisons robustly confirm that scanner variation is overwhelmingly represented in the context space.

We also conducted additional analyses to assess whether attribute features remain independent from context features. Specifically, we computed pairwise pearson correlations between every attribute and context dimension. All cross-correlations were found to be very small ($|r| \leq 0.12$), indicating that the two embedding spaces share minimal information (Fig. 13).

Considering that past CA methods primarily focused on information disentanglement, we also added comparisons of their RSA results on the real ADNI dataset (Table 18). Specifically, InfoSepGAN demonstrates superior disentanglement capabilities: it consistently captures AD-specific attributes (e.g., CSF biomarkers, cognitive scores, and ApoE-E4) with the highest effect sizes and statistical significance in the attribute factors. Simultaneously, for nonclinical features (e.g., age, sex, ICV, site, scanner type), InfoSepGAN achieves comparable or superior performance to general CA baselines in isolating these shared variations within the context space. Crucially, unlike general CA mod-

Table 17: Comparison of InfoSepGAN attribute/context features and an ablation version of InfoSepGAN (w.o. Context).

| Feature | Attribute > Attr (w.o. Context) | Context > Attr (w.o. Context) |
|---|---|---|
| CSF-Tau | $\Delta\tau$=0.0422, $t(9)$=7.8321, **p** $<$ **.0001** | $\Delta\tau$=-0.0207, $t(9)$=-5.5993, **p** $<$ **.0001** |
| CSF-PTau | $\Delta\tau$=0.0528, $t(9)$=9.5269, **p** $<$ **.0001** | $\Delta\tau$=-0.0300, $t(9)$=-8.1736, **p** $<$ **.0001** |
| DSST | $\Delta\tau$=0.0219, $t(9)$=3.1094, **p** $=$ **0.0136** | $\Delta\tau$=-0.0226, $t(9)$=-4.1125, **p** $=$ **0.0034** |
| ADNI-MEM | $\Delta\tau$=0.0417, $t(9)$=7.9931, **p** $<$ **.0001** | $\Delta\tau$=-0.0983, $t(9)$=-21.4522, **p** $<$ **.0001** |
| ADNI-EF | $\Delta\tau$=0.0323, $t(9)$=5.3655, **p** $<$ **.0001** | $\Delta\tau$=-0.0514, $t(9)$=-9.0064, **p** $<$ **.0001** |
| ADNI-LAN | $\Delta\tau$=0.0439, $t(9)$=9.9011, **p** $<$ **.0001** | $\Delta\tau$=-0.0600, $t(9)$=-14.8059, **p** $<$ **.0001** |
| Chronological Age | $\Delta\tau$=-0.0190, $t(9)$=-4.6678, **p** $<$ **.0001** | $\Delta\tau$=-0.0141, $t(9)$=-2.9379, **p** $=$ **0.0179** |
| Sex | $\Delta\tau$=-0.0157, $t(9)$=-5.0398, **p** $<$ **.0001** | $\Delta\tau$=0.0385, $t(9)$=8.1586, **p** $<$ **.0001** |
| ICV | $\Delta\tau$=0.0006, $t(9)$=0.2563, p=0.8035 | $\Delta\tau$=0.0673, $t(9)$=12.5409, **p** $<$ **.0001** |
| Scanner Type | $\Delta\tau$=-0.0173, $t(9)$=-3.2592, **p** $=$ **0.0116** | $\Delta\tau$=0.0051, $t(9)$=0.9978, p=0.3444 |
| Site | $\Delta\tau$=-0.0077, $t(9)$=-6.7959, **p** $<$ **.0001** | $\Delta\tau$=0.0078, $t(9)$=4.6097, **p** $<$ **.0001** |
| ApoE-E4 | $\Delta\tau$=0.0440, $t(9)$=14.4567, **p** $<$ **.0001** | $\Delta\tau$=-0.0281, $t(9)$=-13.5466, **p** $<$ **.0001** |
| WMH volumes | $\Delta\tau$=0.0576, $t(9)$=10.6413, **p** $<$ **.0001** | $\Delta\tau$=-0.0273, $t(9)$=-3.9951, **p** $<$ **.0001** |

els, InfoSepGAN possesses the unique advantage of interpretable disease progression modeling. Notably, SurrealGAN, which models progression without an explicit context space, its attribute features showed negligible and mostly non-significant correlations with key clinical biomarkers (e.g., CSF-Tau, ADNI-MEM), highlighting the difficulty of learning disease progression directions without effectively isolating background anatomical variation. While baseline models like CAVAE, MM-CAVAE and DoubleInfoGAN show some abilities to separate features, they generally exhibit weaker correlations with clinical variables in the attribute space, whereas InfoSepGAN provides a more robust and clinically meaningful separation.

Finally, we evaluated an ablation version of InfoSepGAN without the context module $E_c$ ("1-to-$k$" mapping version of InfoSepGAN, only have attribute encoder $E_a$). As shown in Fig. 12 and Table 17, removing the context module weakened the association of attribute factors with AD-related clinical variables (e.g. ADNI-MEM, ADNI-EF, ADNI-LAN, DSST, CSF-Tau, CSF-PTau), while increasing their association with nonclinical variation (e.g. age, sex, site, scanner type, ICV). This highlights the necessity of the context module for proper disentanglement and clinically meaningful attribute representations.

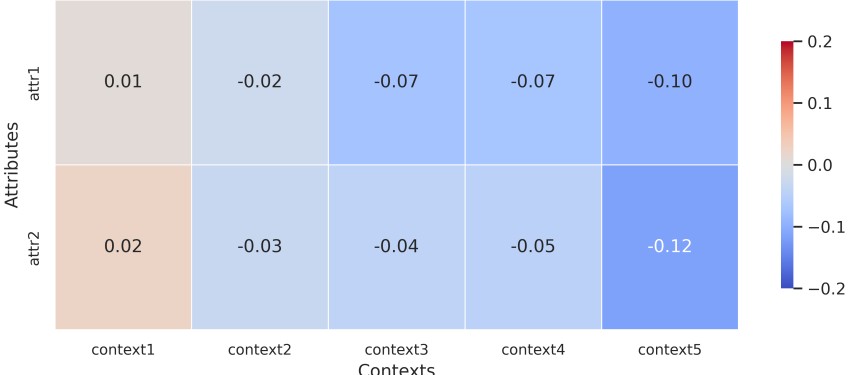

Figure 13: Correlation Matrix between Attribute and Context Embeddings

Table 18: RSA results across clinical or non-clinical features. Values are reported as Mean Kendall's $\tau \pm$ std, and significance level. Significance codes: $(\ast\ast\ast)\, p < 0.0001$, $(\ast\ast)\, p < 0.001$, $(\ast)\, p < 0.05$, (n.s.) not significant. Attribute (A) and Context (C) are reported for each model (SurrealGAN only has Attribute). **Bold** indicates the best-performing model for each feature. The arrows denote the desired direction of disentanglement: $A \uparrow C \downarrow$ indicates that the feature is expected to be primarily captured by the Attribute and minimally by the Context; $A \downarrow C \uparrow$ indicates the opposite preference.

| Feature | InfoSepGAN A | InfoSepGAN C | SurrealGAN A | SurrealGAN C | CAVAE A | CAVAE C | MMCAVAE A | MMCAVAE C | DoubleInfoGAN A | DoubleInfoGAN C |
|---|---|---|---|---|---|---|---|---|---|---|
| CSF-Tau $A\uparrow C\downarrow$ | **0.0612** ±0.0074 (***) | -0.0018 ±0.0043 (n.s.) | 0.0034 ±0.0048 (n.s.) | — | -0.0007 ±0.0109 (n.s.) | **-0.0073** ±0.0057 (*) | -0.0003 ±0.0086 (n.s.) | -0.0048 ±0.0059 (n.s.) | -0.0102 ±0.0103 (n.s.) | -0.0040 ±0.0079 (n.s.) |
| CSF-PTau $A\uparrow C\downarrow$ | **0.0786** ±0.0089 (***) | -0.0042 ±0.0040 (n.s.) | 0.0090 ±0.0049 (*) | — | -0.0037 ±0.0049 (n.s.) | **-0.0099** ±0.0052 (**) | 0.0011 ±0.0101 (n.s.) | -0.0076 ±0.0061 (*) | -0.0103 ±0.0111 (n.s.) | -0.0027 ±0.0074 (n.s.) |
| DSST $A\uparrow C\downarrow$ | **0.0514** ±0.0134 (***) | **0.0069** ±0.0089 (n.s.) | 0.0069 ±0.0076 (n.s.) | — | 0.0259 ±0.0077 (***) | 0.0152 ±0.0079 (**) | 0.0128 ±0.0113 (*) | 0.0091 ±0.0135 (n.s.) | 0.0143 ±0.0112 (n.s.) | 0.0111 ±0.0144 (n.s.) |
| ADNI-MEM $A\uparrow C\downarrow$ | **0.1363** ±0.0114 (***) | **-0.0037** ±0.0054 (n.s.) | 0.0040 ±0.0031 (n.s.) | — | 0.0524 ±0.0239 (**) | 0.0115 ±0.0081 (*) | 0.0124 ±0.0164 (*) | 0.0051 ±0.0128 (n.s.) | 0.0183 ±0.0166 (n.s.) | 0.0182 ±0.0182 (n.s.) |
| ADNI-EF $A\uparrow C\downarrow$ | **0.0895** ±0.0118 (***) | **0.0058** ±0.0082 (n.s.) | 0.0038 ±0.0034 (n.s.) | — | 0.0350 ±0.0199 (*) | 0.0185 ±0.0089 (**) | 0.0142 ±0.0129 (*) | 0.0079 ±0.0118 (n.s.) | 0.0123 ±0.0173 (n.s.) | 0.0128 ±0.0153 (n.s.) |
| ADNI-LAN $A\uparrow C\downarrow$ | **0.1014** ±0.0100 (***) | **-0.0025** ±0.0057 (n.s.) | 0.0061 ±0.0022 (**) | — | 0.0300 ±0.0134 (**) | 0.0106 ±0.0071 (*) | 0.0106 ±0.0131 (*) | 0.0016 ±0.0113 (n.s.) | 0.0089 ±0.0156 (n.s.) | 0.0092 ±0.0148 (n.s.) |
| Chronological Age $A\downarrow C\uparrow$ | 0.0201 ±0.0056 (***) | **0.0250** ±0.0076 (**) | -0.0022 ±0.0016 (*) | — | 0.0008 ±0.0028 (n.s.) | 0.0032 ±0.0066 (n.s.) | -0.0005 ±0.0058 (n.s.) | -0.0016 ±0.0043 (n.s.) | **-0.0031** ±0.0038 (n.s.) | -0.0003 ±0.0030 (n.s.) |
| Sex $A\downarrow C\uparrow$ | 0.0020 ±0.0013 (*) | **0.0562** ±0.0071 (***) | **0.0006** ±0.0003 (*) | — | 0.0024 ±0.0017 (*) | 0.0043 ±0.0015 (***) | 0.0041 ±0.0015 (**) | 0.0055 ±0.0017 (***) | 0.0031 ±0.0012 (**) | 0.0038 ±0.0013 (**) |
| ICV $A\downarrow C\uparrow$ | 0.0198 ±0.0023 (***) | **0.0865** ±0.0154 (***) | **-0.0009** ±0.0020 (n.s.) | — | 0.0110 ±0.0049 (**) | 0.0110 ±0.0050 (**) | 0.0130 ±0.0043 (***) | 0.0168 ±0.0034 (***) | 0.0078 ±0.0043 (*) | 0.0082 ±0.0062 (*) |
| Scanner Type $A\downarrow C\uparrow$ | 0.0193 ±0.0037 (***) | **0.0417** ±0.0084 (***) | **-0.0025** ±0.0027 (n.s.) | — | 0.0344 ±0.0134 (***) | 0.0160 ±0.0115 (*) | 0.0104 ±0.0091 (*) | 0.0247 ±0.0145 (**) | 0.0254 ±0.0161 (*) | 0.0157 ±0.0149 (n.s.) |
| Site $A\downarrow C\uparrow$ | **-0.0042** ±0.0017 (**) | 0.0113 ±0.0038 (**) | 0.0036 ±0.0009 (***) | — | 0.0030 ±0.0039 (*) | 0.0049 ±0.0038 (*) | -0.0013 ±0.0023 (n.s.) | 0.0044 ±0.0051 (*) | 0.0078 ±0.0036 (*) | 0.0045 ±0.0026 (*) |
| ApoE-E4 $A\uparrow C\downarrow$ | **0.0752** ±0.0066 (***) | 0.0032 ±0.0037 (n.s.) | -0.0028 ±0.0021 (*) | — | 0.0156 ±0.0133 (*) | **0.0019** ±0.0043 (n.s.) | 0.0053 ±0.0034 (*) | 0.0048 ±0.0037 (*) | 0.0075 ±0.0102 (n.s.) | 0.0048 ±0.0076 (n.s.) |
| WMH $A\uparrow C\downarrow$ | **0.1270** ±0.0094 (***) | 0.0421 ±0.0110 (***) | -0.0011 ±0.0029 (n.s.) | — | 0.0248 ±0.0174 (*) | 0.0186 ±0.0083 (**) | 0.0274 ±0.0095 (***) | 0.0255 ±0.0060 (***) | 0.0241 ±0.0142 (*) | **0.0170** ±0.0154 (n.s.) |

## B.10 SYNTHETIC COUNTERFACTUALS

A key application of InfoSepGAN is the generation of "synthetic twins" for individual subjects. This is enabled by the model's structured latent space, which disentangles attribute-specific signals ($\mathbf{z}_a$) from shared context factors ($\mathbf{z}_c$), and these counterfactuals are defined in the ROI domain.

**Construction.** For a subject with ROI data $\mathbf{y}_i$ (e.g., an AD patient), we first infer their attribute

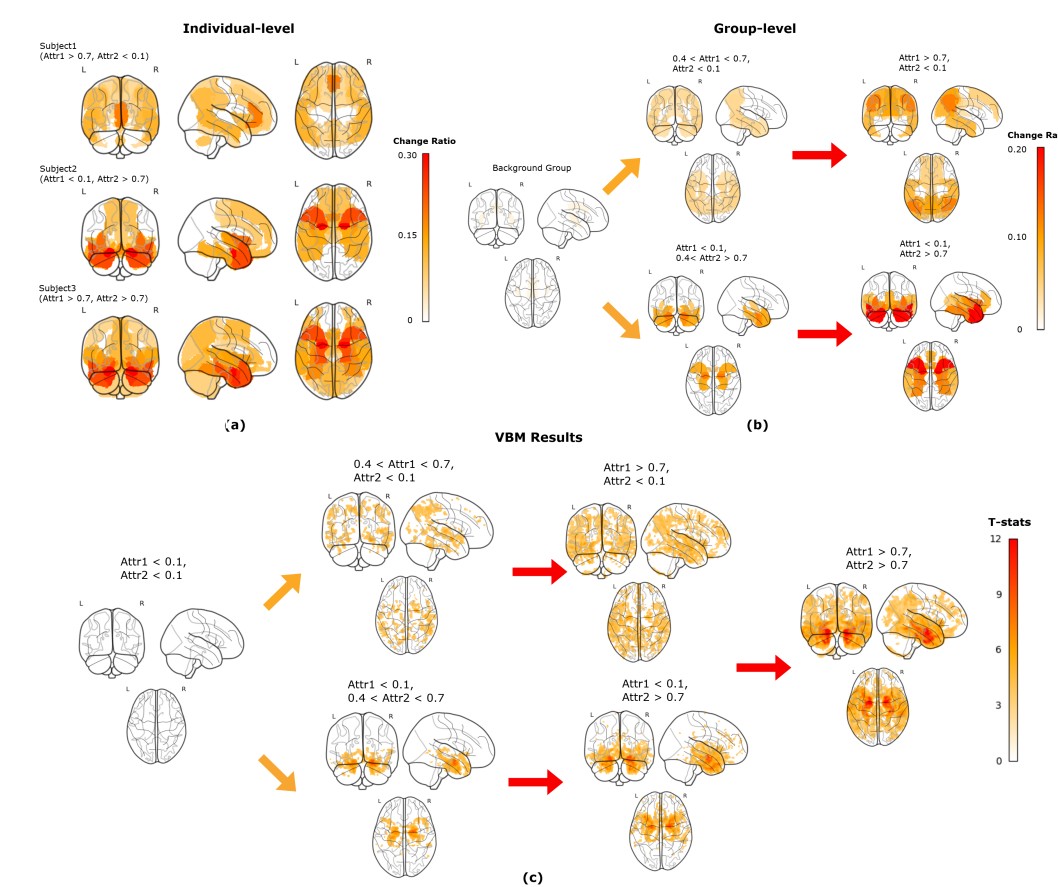

Figure 14: (a) Individual-level analysis of three representative MCI/AD subjects. ROIs are shown as change ratios between original and synthetic counterfactuals ($> 0.05$), highlighting attribute-specific atrophy patterns. (b) Group-level analysis across five subgroups. ROIs are shown as change ratios between original and synthetic counterfactuals, with only ratios $> 0.05$ visualized. Along each arrow direction, subjects with dominant expression of one attribute pattern were selected for visualization. (c) Voxel-wise statistical comparisons between selected subgroups and CN were performed with FDR correction at $p < 0.05$. Along each arrow direction, we increased one attribute factor while keeping the other fixed, enabling selection of subgroups with stronger expression of a given latent pattern for visualization. And an additional mixed-pattern subgroup is included to illustrate combined expression of both factors.

representation $\hat{\mathbf{z}}_{a,i} = E_a(\mathbf{y}_i)$ and context representation $\hat{\mathbf{z}}_{c,i} = E_c(\mathbf{y}_i)$. The generator then reconstructs the subject as $G(\hat{\mathbf{z}}_{a,i}, \hat{\mathbf{z}}_{c,i}) \approx \mathbf{y}_i$. A synthetic twin is reconstructed by setting attribute effects to zero and retaining the context: $\bar{\mathbf{y}}_i = G(\mathbf{0}, \hat{\mathbf{z}}_{c,i})$. Here, $\bar{\mathbf{y}}_i$ preserves the subject's shared features (e.g., age, sex, scanning-site and other factors discovered by the InfoSepGAN in a data-driven fashion), but omits disease-specific alterations (e.g., AD-related atrophy). Thus, $\bar{\mathbf{y}}_i$ functions as a precise, subject-matched control—an advantage over traditional population-level case–control studies, which cannot fully account for complex individual variability.

**Application in Clinical Analysis.** As in section 3.6, synthetic counterfactuals are useful at both the individual and group level: 1. Individual-level analysis. For a given subject, directly comparing $\mathbf{y}_i$ with $\bar{\mathbf{y}}_i$ reveals ROI-specific changes linked to the attribute of interest, supporting personalized assessment of pathological effects. 2. Group-level analysis. For a cohort $S$, we can summarize attribute-related effects by $\Delta_{\text{group}} = \frac{1}{N_S} \sum_{i \in S} ((\bar{\mathbf{y}}_i - \mathbf{y}_i)/\bar{\mathbf{y}}_i)$, which summarizes group-level deviations. This validates the biological relevance of $\mathbf{z}_a$.

To further illustrate the utility of synthetic counterfactuals, we visualize both individual- and group-

level analyses. At the individual level, we selected three representative patients: (i) a subject primarily driven by Pattern1 (Attr1 $> 0.7$, Attr2 $< 0.1$); (ii) a subject primarily driven by Pattern2 (Attr1 $< 0.1$, Attr2 $> 0.7$); and (iii) a subject with strong contributions from both patterns (Attr1 $> 0.7$, Attr2 $> 0.7$). For each subject, we compare the original ROI with its synthetic twin, highlighting attribute-specific atrophy regions that characterize distinct pathological signatures (shown in Fig. 14a).

At the group level, we further explored five subgroups: (1) a background group consisting of cognitively normal subjects, and four disease-related subgroups defined by attribute factors. Specifically, for each attribute $i$, we considered subgroups with $0.4 < \text{Attr}_i < 0.7$ and $\text{Attr}_i > 0.7$, with $\text{Attr}_j < 0.1$ for all $j \neq i$. Importantly, the captured attribute-driven patterns were highly consistent with VBM analyses in key regions, particularly in key ROIs implicated in AD pathology, further validating the interpretability of InfoSepGAN (shown in Fig. 14b, c).

**Voxel-wise analysis:** To support interpretability, we performed voxel-wise statistical comparisons between selected subgroups and CN group using t-tests. Results were FDR-corrected at $p < 0.05$ (Fig. 14c). This voxel-wise approach complements the ROI-level synthetic counterfactual visualization, confirming that the captured attribute-specific patterns align with known AD pathology.

### B.11 DATA AVAILABILITY

The raw imaging and clinical data analyzed in this study were obtained through formal data access agreements with the original data providers. Due to data sharing restrictions, we are not permitted to redistribute these datasets. Researchers may apply for access directly from the source repositories. The ADNI dataset is available upon registration and compliance with the data use agreement at the ADNI website (`http://adni.loni.usc.edu`). The UK Biobank dataset can be requested via formal application through the UK Biobank website (`https://www.ukbiobank.ac.uk/`).

