# OpenReview forum: "An Interpretable Contrastive GAN Approach for Identifying Heterogeneous Pathological Imaging Patterns"
_ICLR.cc/2026/Conference — Submitted to ICLR 2026_

### Official Review · Reviewer_1Tcw · 2025-10-29

**Soundness:** 3
**Presentation:** 3
**Contribution:** 3
**Rating:** 6
**Confidence:** 4

**Summary:**

The authors present a contrastive generative framework to identify, in a data-driven manner, disease characteristics and subtypes from brain MR images. The core idea is to separate imaging features into what is common to two populations (e.g., variability in brain size), termed the 'context', and what is specific to a patient cohort (e.g., reduced cortical thickness), termed 'attributes'. The principal contribution of this work is that this separation remains robust even when the context is unequally distributed between populations, such as controls and patients. This approach enables the identification of traits specific to a disease population and facilitates the reconstruction of pseudo-healthy features for patients to improve interpretability.

The proposed method builds upon a GAN framework, augmenting it with several key components. These include a mutual information-based regularisation, inspired by InfoGAN, to effectively disentangle context and attributes, alongside a reconstruction loss. To ensure the attributes reflect clinically plausible pathological processes, the authors introduce a suite of specialised regularisation terms. These enforce sparsity (as pathology is often localised), pattern separation (to reflect different patterns of variation), monotonicity (reflecting disease progression), background consistency (to prevent pathological changes in their absence), and decomposition (to mitigate mode collapse and encourage diverse, meaningful representations).

**Strengths:**

**Methods**:
The primary methodological strength is the focus on robustness to confounding factors,  i.e. non-pathological effects that differ between the two populations studied, e.g. young controls and old patients. The paper builds a contrastive generative framework upon a GAN base, explicitly designed to separate 'context' from 'attributes' in a way that is stable even when context is unequally distributed between populations. The framework is augmented with a set of well-justified regularisation terms, which are carefully chosen to ensure the derived attributes reflect clinically plausible pathological processes.

**Experiments**:
A key strength is the evaluation strategy, which is designed to progressively validate the model under increasingly challenging conditions. The authors begin with controlled experiments on synthetic data, systematically testing the model's sensitivity to confounding severity, the proportion of affected subjects, and the spatial overlap between confounding and pathological effects. This is followed by an assessment on semi-synthetic data based on the UK Biobank, providing a more realistic testbed. Finally, the framework is applied to real clinical data from ADNI, demonstrating its utility on a genuine problem.

**Results**:
The results validate the core contribution of the method. They show that the proposed approach achieves performance on a par with SOTA techniques when no confounding effects are present. More importantly, the results demonstrate the key strength: in the presence of confounders, the method maintains robust performance, whereas the compared SOTA methods show a marked degradation. This directly addresses a significant challenge in the field.

**Weaknesses:**

- **Limited baselines**: The comparison could be more comprehensive. While SurrealGAN (2024) is a relevant SOTA baseline, the others are from 2015-2018. The paper would be stronger by including contemporary methods from the "contrastive analysis" family (Abid & Zou, 2019; Aglinskas et al., 2022; Carton et al., 2024; Louiset et al., 2024; Vowels et al., 2020) to better benchmark its contribution to robust pattern discovery under confounding factors.
- **Unclear real-data evaluation**: The results on the real ADNI dataset lack clarity. It is not well-explained how the findings in Figure 4 were obtained.
- **Hyperparameter sensitivity unexplored**: The model uses many loss terms with weights spanning a wide range (0.08 to 500). The paper does not discuss the sensitivity of the results to these choices or the difficulty of tuning them, which is a significant concern for reproducibility and practical application.

**Questions:**

- **Real-data analysis**: Could you please clarify the exact procedure for obtaining the voxel-wise group differences shown in Figure 4?
- **Method comparison**: To better position the contribution, could the performance be compared against other recent contrastive analysis frameworks?
- **Hyperparameter tuning**: How sensitive are the results to the specific combination of loss weights? Some discussion or analysis of the tuning process and the model's robustness to these choices would be helpful.
- **Monotonicity loss**:  What are the implications of the use of this loss for diseases with less clear progression?

---

> ### Author Response · Authors · 2025-11-24
> **Response to Reviewer 1Tcw (Part1)**
>
> It is a pleasure to read the reviewer’s comprehensive summary and strong endorsement of our methodology and experimental design. We thank the reviewer for the constructive and insightful comments, particularly regarding baseline comparisons and hyperparameter sensitivity, which will significantly improve the rigor of our manuscript.
>
> 1. Limited baselines & Method comparison: The paper would be stronger by including contemporary methods from the "contrastive analysis" family to better benchmark its contribution to robust pattern discovery under confounding factors. Could the performance be compared against other recent contrastive analysis frameworks?
>
> Response:
> We appreciate the reviewers' suggestion to broaden the scope of comparisons.   Our initial choice of SurrealGAN focused on as the primary baseline because it is currently the closest state-of-the-art model specifically designed for learning disease dimensions and progression patterns. Most existing contrastive analysis (CA) methods (e.g., CAVAE, MMCAVAE, DoubleInfoGAN) focus primarily on information disentanglement rather than explicitly modeling disease progression (spatial and temporal variability). Without specific regularization terms (e.g., monotonicity, sparsity, and pattern separation), these general-purpose CA models struggle to spontaneously learn interpretable, progressive pathological patterns, which is precisely the core objective of InfoSepGAN.
>
> However, we fully agree that benchmarking against other methods in the CA field is valuable. We added comparisons with CAVAE, DoubleInfoGAN, and MMCAVAE in **Appendix B.4, Table 10**. We evaluated their performance on our semi-synthetic task, and as expected, they did not outperform InfoSepGAN in capturing disease progression patterns due to a lack of specific constraints. This confirms that our specially designed regularization framework is crucial for transforming general disentanglement into clinically interpretable disease progression models.
>
> Considering that past CA methods primarily focused on information disentanglement, we also added comparisons of their RSA results on the real ADNI dataset (**Appendix B.9, Table 18**). InfoSepGAN demonstrates superior disentanglement compared to all baselines. Its attribute space captures AD-specific features (e.g., CSF, cognitive scores) with the highest significance, while effectively isolating nonclinical features in the context space. In contrast, methods like SurrealGAN showed negligible correlations with key clinical biomarkers (e.g., CSF-Tau, ADNI-MEM). While baseline models like CAVAE, MMCAVAE and DoubleInfoGAN show some abilities to separate features, they generally exhibit weaker correlations with clinical variables in the attribute space, whereas InfoSepGAN provides a more robust and clinically meaningful separation.
>
> We did not include NestedVAE in the comparisons because it requires paired data, which is not suitable for our situation. Similarly, SepCLR relies on image augmentation techniques that cannot be directly applied to our ROI-level data.
>
> 2. Unclear real-data evaluation: The results on the real ADNI dataset lack clarity. It is not well-explained how the findings in Figure 4 were obtained. / Real-data analysis: Could you please clarify the exact procedure for obtaining the voxel-wise group differences shown in Figure 4?
>
> Response:
> We thank the reviewer for highlighting the need for greater clarity in our real-data analysis. We have revised **Section 5.3** and the caption of **Fig. 4** to better describe how the results were obtained.
>
> For **Fig. 4a**, the voxel-wise maps were generated by conducting voxel-wise t-tests between regional tissue density and each learned attribute factor, while controlling for age, sex, ICV, and the other attribute. VBM is used solely as an interpretability tool to visualize how each attribute relates to gray matter atrophy. The exact preprocessing and statistical procedure are fully documented in **Appendix B.6**.
>
> For **Fig. 4b in Section 5.3**, we substantially expanded the description of the RSA analysis. We clarify that we computed representational dissimilarity matrices (RDMs) for both attribute and context features, and compared them with RDMs derived from a wide range of clinical and nonclinical variables. This analysis shows that attribute features align with AD-related markers, whereas context features capture shared, non-pathological variation. The complete statistical results have been moved to **Appendix B.9** for clarity, with **Section 5.3** now providing a concise summary of the key findings.

---

> > ### Author Response · Authors · 2025-11-24
> > **Response to Reviewer 1Tcw (Part2)**
> >
> > 3. Hyperparameter sensitivity unexplored / Hyperparameter tuning: How sensitive are the results to the specific combination of loss weights?
> >
> > Response:
> > We thank the reviewers for their important points regarding model robustness and reproducibility, and sincerely apologize for not clearly emphasizing these analyses in the main text. We conducted a comprehensive sensitivity study, detailed in **Appendix B.5**, in which we systematically adjusted the key loss weight parameter ($\lambda$ value) from the default 50% to 150%. As shown in **Fig. 8**, the model's performance (measured by PCI) remained stable across the entire range. This indicates that InfoSepGAN is not particularly sensitive to the specific choice of loss weights and does not require fine-tuning of hyperparameters to operate effectively.
> >
> > To further address the challenge of tuning real-world datasets without true labels, we introduced the Pattern-agr-index (ACI). Experiments show that the agreement of independent model runs is highly correlated with actual performance (PCI), meaning that ACI provides a practical and data-driven approach to selecting hyperparameters in real-world applications. We have revised **Section 4.2 (L330 - L335)** accordingly to highlight this strategy and guide readers to the supporting analyses in the appendix.
> >
> > 4. Monotonicity loss: What are the implications of the use of this loss for diseases with less clear progression?
> >
> > Response:
> > We thank the reviewer for raising this insightful and forward-looking question. InfoSepGAN was primarily designed for neurodegenerative disorders such as Alzheimer’s disease that are known to exhibit progressive, severity-dependent pathological changes. In such cases, the monotonicity loss plays an important role in encouraging the latent dimension to align with this continuous disease severity axis. For disorders that do not exhibit a clear progression pattern (e.g., stable subtypes), we agree that enforcing monotonicity could potentially impose an artificial ordering that lacks biological interpretability. Importantly, the monotonicity constraint is not a mandatory structural constraint of the framework itself. Researchers can simply set the weight of the monotonicity loss weight to zero when applying the model to diseases without a progressive trajectory. In this setting, the remaining regularization terms, such as sparsity ($L_{\text{sparse}}$) and pattern separability ($L_{\text{disen}}$), continue to function effectively and can still identify meaningful but non-progressive patterns or subgroups.

---

> ### Comment · Reviewer_1Tcw · 2025-11-25
>
> I'd like to thank the authors for their responses and for clarifying several aspects, especially regarding the real-data analysis, comparisons with additional contrastive analysis methods, and the discussion of hyperparameter sensitivity and monotonicity loss. These additions address my main technical questions.
>
> However, while these clarifications and additional results are appreciated, they do not fundamentally change my overall assessment, which was already positive.

---

> > ### Author Response · Authors · 2025-11-25
> >
> > We sincerely thank reviewer 1Tcw for the thoughtful comments and positive assessment. Your positive evaluation and confirmation that these additions strengthen the manuscript are highly valued. We appreciate your time and expertise in reviewing our work.

---

### Official Review · Reviewer_7JoC · 2025-10-31

**Soundness:** 2
**Presentation:** 1
**Contribution:** 2
**Rating:** 2
**Confidence:** 4

**Summary:**

The authors attempt to address unmodeled confounders in neuroimaging data by creating a generative model that tries to explicitly model confounders and disease-specific attributes as independent. The model is evaluated on a simulated, semi-simulated, and real neuroimaging dataset. The latter is an Alzheimer's disease dataset with many known confounders and disease progression metrics.

**Strengths:**

I think the authors make some interesting, and good, design choices, such as modeling disease severity as a spectrum. Moreover, it is encouraging that the authors actually find some significant correlations with biomarkers in the CSF in Tables 12 and 13. If the authors can show better generalization and how their design choices affect model performance, I believe this paper can be impactful.

**Weaknesses:**

Major weaknesses:
1) Lack of ablations. The authors a lot of changes to previous models that they do not ablate, especially the extra regularization terms in their loss function Equation 4 are added, but not ablated.
2) Limited evaluation of generalizability. The authors use two simulation datasets, which is a good start, but they only evaluate their model on a single real neuroimaging dataset.
3) Unconvincing results. The authors only evaluate against one other GAN, and only on the semi-simulation data. To improve the work, the authors should include more baselines and include ablations for each of their proposed additions and regularization terms as mentioned in 1). Moreover, the authors do not compare their model to a GAN baseline at all on the only real dataset. This makes it hard to verify whether the model actually performs better on a realistic task. Moreover, the results show that age are captured in both the context and attribute embeddings, which means that the authors are not able to separate certain confounders. Similarly, the scanner type is still included in the attribute embedding space. It is potentially arguable that it is impossible to completely decouple age from Alzheimer's disease-specific attributes, but that is definitely not the case for scanner type. Moreover, the authors explicitly assume that the context and attribute embeddings are conditionally independent (L168-169). The model is clearly not learning conditionally independent embeddings, so the authors either need to change their design choices or improve their training objective to enforce the conditional independence.

Minor weaknesses/grammar/spelling errors:
- The authors spend almost 4/9 pages describing their model with mathematical equations, but the presentation, explanation, and intuition behind many of these choices is lacking. In general this section is quite hard to follow, and I am unsure why the authors decided to have the method section take up so much of the paper, especially because the authors leave little room to discuss their results. I would encourage the authors to keep the most important methodological advances, and mostly focus on explaining choices in the loss function. Many of the equations take up a lot of space, but are not novel theoretical results, specifically: equations 1, 2, and 3. Reading these subsections reminded me of the term 'mathiness' as coined by Lipton and Steinhardt [1]. Moreover, the simulation and baseline results are currently in the Appendix, the authors should move these results to the main text.
- The authors also only use a very specific evaluation metric as their main evaluation metric, the pattern-c-index (PCI). I believe Figure 3 would require multiple evaluation metrics.
- Figure 2 is unclear. What do all the symbols mean? What is the difference between a dashed and solid line etc.?
- The text in Figure 3 is hard to read, also the x-axis of Figure 3a is unclear without reading the text. The authors should add an explanation of each term to the caption. Moreover, what is InfoSepGAN (w.) vs InfoSepGan (w.o.)?
- On L138-139 the authors claim: "While known confounders can be corrected using linear adjustments, it is impractical to account for all potential confounding factors, ...". After reading the paper, it is unclear to me how the authors guarantee that their method actually learns all of the confounding factors, especially given the issue of shortcut learning in medical AI [2].
- The authors should move their discussion on Lines 338-347 to the discussion section.
- L107: "..., these approaches remain limitations ..." -> remain limited
- L210/211: "Maximizing these two terms serves following purposes, ..." -> serves the following
- L255/256: "We adapts and extends the terms ..." -> We adapt and extend


[1] Lipton, Z. C., & Steinhardt, J. (2019). Troubling Trends in Machine Learning Scholarship: Some ML papers suffer from flaws that could mislead the public and stymie future research. Queue, 17(1), 45-77. \
[2] Brown, A., Tomasev, N., Freyberg, J., Liu, Y., Karthikesalingam, A., & Schrouff, J. (2023). Detecting shortcut learning for fair medical AI using shortcut testing. Nature communications, 14(1), 4314.

**Questions:**

- Some specific choices are unclear, for example the authors use a uniform spectrum for disorders (why a uniform and not, for example a normal distribution?). I understand it is easier in the sense that it is a bounded distribution, and control subjects can be assigned as 0, but is there a neurobiological reason for this design choice? Moreover, for many spectrum disorders it is also believed that controls may lie somewhere on the same spectrum as people diagnosed with a neurological or psychiatric disorder. The authors currently set the attribute factors for controls to 0, have the authors thought about using a different prior on the control subject attributes?

---

> ### Author Response · Authors · 2025-11-24
> **Response to Reviewer 7JoC (Part1)**
>
> We sincerely thank reviewer for the detailed and thoughtful evaluation of our work. We greatly appreciate the reviewer’s careful reading, constructive comments, and insightful suggestions. Many of the issues raised—such as ablations, clarity of presentation, and the need for broader empirical evaluation—have helped us substantially improve both the methodology and the exposition. We address each point below and have incorporated corresponding revisions into the updated manuscript.
>
> 1. Lack of ablations. The authors a lot of changes to previous models that they do not ablate, especially the extra regularization terms in their loss function Equation 4 are added, but not ablated. Limited evaluation of generalizability. The authors use two simulation datasets, which is a good start, but they only evaluate their model on a single real neuroimaging dataset.
>
> Response:
>
> (1) Ablation experiment:
>
> We apologize for not making our ablation studies more prominent in the main text. We conducted extensive ablation and robustness analyses, which are detailed in **Appendix B.5**. We performed ablation experiments on both the "basic" and the more challenging "middle age gap" scenarios. As shown in **Fig. 7**, removing key regularization terms or the context encoder ($E_c$) significantly degrades performance. This effect is particularly pronounced in the "middle age gap" scenario, where distributional shifts between BG and TG are larger, underscoring the necessity of our full model architecture for robust disentanglement. Additionally, we systematically varied key hyperparameters across a range of 50% to 150% of their default values. **Fig. 8** demonstrates that the model's performance (measured by PCI) remains stable across this range, indicating that InfoSepGAN is robust and does not require precise tuning of these weights to function effectively. To address the challenge of hyperparameter selection in real-world datasets lacking ground truth labels, we introduced the Pattern-agr-index (ACI). Our experiments confirm that ACI—which measures the agreement between independent model runs—correlates strongly with representation accuracy (PCI). This provides a practical, data-driven criterion for model selection (**Appendix A.9, Fig. 8**). We have revised **Section 4.2 (L330 - L335)** in the main text to explicitly highlight these findings and direct readers to the detailed results in the Appendix.
>
> (2) External dataset experiment:
>
> To strengthen the evaluation of model generalizability, we additionally conducted external validation experiments using the UK Biobank (UKBB), as detailed in **Appendix B.8**. Unlike ADNI, UKBB does not contain AD-specific phenotypic information, such as diagnostic labels, CSF, or PET measures, making it difficult to reliably define AD cases or a “pure” healthy control cohort. Following the exclusion criteria established in [1], we defined the healthy controls. Furthermore, to mitigate these cohort-level differences, we used the widely validated ComBat-GAM [2] method to harmonize the two datasets. This method effectively removes site effects while preserving biologically meaningful variability.
>
> Specifically, we retrained InfoSepGAN using **1,000** randomly selected healthy controls from UKBB as the reference group and the ADNI MCI/AD cohort as the target group. The resulting attribute representations were compared against the original ADNI-only results using all four pattern evaluation metrics (PCI, ACI, PPC, PDC). Qualitatively, the atrophy maps derived from the retrained model (**Appendix B.8, Fig. 11**) consistently replicated the original findings: Attr1 reflected diffuse cortical atrophies across parietal, occipital, and temporal regions, and Attr2 highlighted focal medial temporal and subcortical atrophies. Quantitatively, an external reference group from UKBB in model retraining did not yield significant changes in identified patterns (**PCI = 0.774 ± 0.041; PPC = 0.819 ± 0.012**), confirming that InfoSepGAN is robust to the choice of reference cohort and generalizes well across distinct imaging populations.
>
> We further applied the UKBB-trained model to 47,004 asymptomatic individuals in UKBB and observed significant associations between the learned attribute factors and clinically features—including white-matter hyperintensities and multiple cognitive measures (**Appendix B.8, Table 14**)—demonstrating that the learned representations capture biologically meaningful variation even outside the ADNI dataset.
>
> Due to the limited time available for revision, we were able to include external validation on only one additional dataset. Nevertheless, UKBB is a large-scale, population-based cohort with substantial demographic variability, and thus provides strong evidence for the generalizability of InfoSepGAN. Extending the evaluation to additional multi-site datasets is an important direction that we plan to pursue in future work.

---

> > ### Author Response · Authors · 2025-11-24
> > **Response to Reviewer 7JoC (Part2)**
> >
> > 2. Unconvincing results. The authors only evaluate against one other GAN, and only on the semi-simulation data. To improve the work, the authors should include more baselines and include ablations for each of their proposed additions and regularization terms as mentioned in 1). Moreover, the authors do not compare their model to a GAN baseline at all on the only real dataset. This makes it hard to verify whether the model actually performs better on a realistic task.
> >
> > Response:
> > We appreciate the reviewers’ suggestion to broaden the comparisons. In the revised manuscript, we substantially expanded both the baseline evaluations.
> >
> > Our updates include a rigorous evaluation on both semi-synthetic and real datasets. On the semi-synthetic dataset, we supplement the comparisons by adding current contrastive analysis (CA) methods, including **CAVAE, DoubleInfoGAN, and MMCAVAE (Appendix B.4, Table 10)**. Since these general CA models primarily target information disentanglement rather than disease progression modeling, they did not outperform InfoSepGAN in capturing the underlying progression patterns, confirming that the specialized regularization terms in InfoSepGAN are crucial for transforming generic disentanglement into clinically interpretable disease progression models.
> >
> > Furthermore, we extended the comparison of all models to the real ADNI dataset using Representational Similarity Analysis (RSA) (**Appendix B.9, Table 18**), which provided a crucial benchmark on a realistic task. In this analysis, InfoSepGAN consistently demonstrated superior disentanglement, capturing AD-specific biomarkers (e.g., CSF-Tau, ADNI-MEM) with the strongest correlations in the attribute space while simultaneously isolating nonclinical variability in the context space. SurrealGAN, which models progression without an explicit context space, showed negligible and mostly non-significant correlations with key clinical biomarkers, highlighting the inherent difficulty of learning meaningful disease progression directions without effectively isolating background anatomical variation, and while other CA baselines showed some ability to separate features, they generally exhibited weaker correlations with clinical variables in the attribute space, confirming InfoSepGAN provides a more robust and clinically meaningful separation.
> >
> > Finally, to validate the necessity of our core architectural innovation, we performed an **ablation study on the real ADNI dataset** by implementing a "1-to-k" version of InfoSepGAN that removes the context encoder ($E_c$). Results (**Fig. 12 and Table 17**) show that removing the context module weakened the attribute associations with AD-related biomarkers and simultaneously increased the leakage of nonclinical variation (age, sex, site, scanner type) into the attribute space. These findings highlight the necessity of the context module for properly handling background anatomical variability between the BG and TG, which is a key mechanism for **mitigating biases** introduced by non-pathological variation, and this result confirms the core contribution of InfoSepGAN's design on a realistic task.
> >
> >
> > [1] Hanyi Chen, Alexandra Young, Neil P Oxtoby, Frederik Barkhof, Daniel C Alexander, Andre Altmann, et al. Transferability of alzheimer’s disease progression subtypes to an independent population cohort. NeuroImage, 271:120005, 2023a.
> >
> > [2] Raymond Pomponio, Guray Erus, Mohamad Habes, Jimit Doshi, Dhivya Srinivasan, Elizabeth Mamourian, Vishnu Bashyam, Ilya M Nasrallah, Theodore D Satterthwaite, Yong Fan, et al. Harmonization of large mri datasets for the analysis of brain imaging patterns throughout the lifespan. NeuroImage, 208:116450, 2020.

---

> > > ### Author Response · Authors · 2025-11-24
> > > **Response to Reviewer 7JoC (Part3)**
> > >
> > > 3. Moreover, the results show that age is captured in both the context and attribute embeddings, which means that the authors are not able to separate certain confounders. Similarly, the scanner type is still included in the attribute embedding space. It is potentially arguable that it is impossible to completely decouple age from Alzheimer's disease-specific attributes, but that is definitely not the case for scanner type.
> > >
> > > Response:
> > > We thank the reviewer for this critical observation regarding the separation of age and scanner type, as this touches upon fundamental challenges in disentangling factors within real-world, complex medical datasets. We provide a detailed justification for our findings, asserting that our method still provides the most clinically meaningful separation.
> > >
> > > For **Age**, we note that our RSA results showed that the difference in association for age between the attribute and context space is statistically non-significant ($\Delta\tau$=-0.0049, $p = 0.1927$), indicating that $z_a$ does not preferentially encode age (**Table 16**). However, the slight residual association of age in the attribute space is not merely due to disentanglement failure, but has a biological explanation. Previous literature has confirmed that age affects brain structural changes in AD patients, implying that age interacts with AD-specific atrophy patterns, rather than a simple independent confounding factor [3]. Therefore, the $z_a$ space encodes age-related AD atrophy patterns.
> > >
> > > For **Scanner Type**, we acknowledge that our RSA results show a small but statistically significant correlation within the attribute space. However, this effect is consistently **much weaker** than that encoded by the context features, as confirmed by paired comparisons ($\Delta\tau$=-0.0224, $p < 0.0001$). The scanner type is **overwhelmingly represented in the context space**, which aligns with its role in capturing variation unrelated to disease status. This weak residual correlation in $z_a$ likely stems from subtle distributional characteristics inherent to the ADNI cohort, where later-phase, predominantly 3T scanners tend to include more MCI/AD subjects than earlier 1.5T protocols; consequently, scanner type becomes weakly informative of disease information, making its complete removal extremely challenging. Crucially, the presence of this minimal leakage in $z_a$ does not affect our core conclusion that the attribute space remains highly specific to AD-related biological variation.
> > >
> > > We emphasize that InfoSepGAN’s overall disentanglement is superior to existing baselines. While some other models like SurrealGAN or DoubleInfoGAN may not show significant associations with several single confounding factors (e.g., age or scanner type) in their attribute space, they simultaneously exhibit negligible or non-significant correlations with key AD-specific biomarkers (e.g., ADNI-MEM, CSF-Tau), demonstrating that their attribute space lacks clinical utility. Conversely, InfoSepGAN’s attribute features show the highest and most consistent correlations with AD-specific properties while InfoSepGAN’s context features show the strongest overall alignment with nonclinical variation (**Table 18**). These findings suggest that InfoSepGAN effectively disentangles general effects from those that specifically interact with AD.
> > >
> > >
> > > [3] Juergen Dukart, Karsten Mueller, Arno Villringer, Ferath Kherif, Bogdan Draganski, Richard Frackowiak, Matthias L Schroeter, Alzheimer’s Disease Neuroimaging Initiative, et al. Relationship between imaging biomarkers, age, progression and symptom severity in Alzheimer’s disease. NeuroImage: Clinical, 3:84–94, 2013.

---

> > > > ### Author Response · Authors · 2025-11-24
> > > > **Response to Reviewer 7JoC (Part4)**
> > > >
> > > > 4. Moreover, the authors explicitly assume that the context and attribute embeddings are conditionally independent (L168-169). The model is clearly not learning conditionally independent embeddings, so the authors either need to change their design choices or improve their training objective to enforce the conditional independence.
> > > >
> > > > Response:
> > > > We thank the reviewer for raising this important point regarding the conditional independence assumption between the context and attribute embeddings. To assess this, we first examined the empirical dependence between the two spaces. Pairwise pearson correlations between every attribute and context dimension were extremely small ($|r| \leq 0.12$), indicating that the spaces share minimal information and behave close to conditionally independent in practice (**Fig. 13**).
> > > >
> > > > We initially experimented with enforcing stricter independence constraints, such as the“null mutual information”regularizer proposed in **SepCLR** [4]. Interestingly, imposing such constraints consistently led to a notable decline in performance and degrading the model's ability to learn accurate disease patterns. These observations may suggest that overly strict independence may remove subtle but biologically relevant interactions necessary for capturing AD-related variation.
> > > >
> > > > Given these findings, we adopted a more balanced and practically effective strategy. As described in **Section 3.2**, we introduce an information-theoretic loss to promote disentanglement between context and attribute factors. In addition,, $z_a$ and $z_c$ are sampled independently by construction during training, which further limits potential information leakage while preserving representational flexibility.
> > > >
> > > > 5. The authors spend almost 4/9 pages describing their model with mathematical equations, but the presentation, explanation, and intuition behind many of these choices is lacking. I would encourage the authors to keep the most important methodological advances, and mostly focus on explaining choices in the loss function.
> > > >
> > > > Response:
> > > > We appreciate the valuable suggestions from our reviewers. Our initial intention was to fully demonstrate the implementation of InfoSepGAN, including the components of the loss function and the training objective. However, we agree that excessive mathematical detail in the main text could reduce readability and distract readers from the key methodological contributions.
> > > >
> > > > In the revised manuscript, we have significantly streamlined **Sections 3.1 to 3.3**, retaining only the essential components that constitute the main methodological advancements. We have simplified the mathematical expressions as much as possible and moved the derivations and underlying implementation details to the appendix for reference. Furthermore, we have expanded the corresponding explanations, more clearly illustrating the logic behind each design choice in the loss function.
> > > >
> > > > We believe these revisions improve the clarity of the article, helping readers better understand the motivations behind the loss function design without being bogged down by technical details, while ensuring the complete reproducibility of the experiments through the content in the appendix.
> > > >
> > > >
> > > > [4] Robin Louiset, Edouard Duchesnay, Antoine Grigis, and Pietro Gori. Separating common from salient patterns with contrastive representation learning. In International Conference on Learning Representations (ICLR), 2024.

---

> > > > > ### Author Response · Authors · 2025-11-24
> > > > > **Response to Reviewer 7JoC (Part5)**
> > > > >
> > > > > 6. The simulation and baseline results are currently in the Appendix, the authors should move these results to the main text. The authors also only use a very specific evaluation metric as their main evaluation metric, the pattern-c-index (PCI). I believe Figure 3 would require multiple evaluation metrics.
> > > > >
> > > > > Response:
> > > > > We thank the reviewer for the constructive suggestion regarding the presentation of our results and evaluation metrics. We agree that the most crucial evidence should be easily accessible in the main text and that relying solely on one metric is insufficient for a complete assessment.
> > > > >
> > > > > We conducted a total of **25** synthetic and semi-synthetic experiments. Due to space limitations in the main text, it would be inappropriate to include all of them. We have moved only the most essential semi-synthetic results, specifically those reflecting the raw performance of the models without linear adjustments, into **Table 1** in the main text. The full set of extensive synthetic and additional semi-synthetic results remains available in the Appendix for completeness and reproducibility.
> > > > >
> > > > > Regarding the evaluation metrics, we acknowledge that the PCI, while suitable for quantifying the accuracy of disease-generative pattern recovery, is not enough on its own. We have therefore expanded the quantitative evaluation presented in **Table 1** to include the Pattern-agr-index (ACI), Pattern-Pearson-Correlations (PPC), and Pattern-Difference-Correlations (PDC). This expanded set offers a more rigorous assessment by capturing complementary aspects of performance: PCI and PPC measure the accuracy of pattern recovery, while ACI and PDC assess the reliability and structural consistency of the learned representations. The formal definitions and implementation details of all these metrics are now provided in **Appendix A.9**. We believe this expanded evaluation ensures a more complete and rigorous comparison across all methods.
> > > > >
> > > > > 7. Figure 2 is unclear. What do all the symbols mean? What is the difference between a dashed and solid line etc? The text in Figure 3 is hard to read, also the x-axis of Figure 3a is unclear without reading the text. The authors should add an explanation of each term to the caption. Moreover, what is InfoSepGAN (w.) vs InfoSepGan (w.o.)?
> > > > >
> > > > > Response:
> > > > > We thank the reviewer for the helpful suggestions regarding **Fig. 2 and 3**. We have updated both figures to improve clarity and readability. In **Fig. 2**, we revised the visual design and expanded the caption to explicitly explain the meaning of all symbols, the distinction between solid and dashed lines. The solid lines now represent the generative assumptions made by each paradigm, whereas the dashed lines denote inference paths from observed data to latent factors. The caption also clarifies the interpretation of the shared context space C and the attribute spaces A, as well as the meaning of E and G.
> > > > >
> > > > > For **Fig. 3**, we clarified the x-axis of **Fig. 3a**, which now indicates increasing confounding difficulty from basic settings to more severe overlap scenarios. We further revised the caption to define all terms appearing in the figure. In particular, we clarified that “InfoSepGAN (w.)” and “InfoSepGAN (w.o.)” refer to results with and without linear correction for age, sex, and ICV, respectively. These revisions ensure that readers can interpret the figures without relying on the main text.

---

> > > > > > ### Author Response · Authors · 2025-11-24
> > > > > > **Response to Reviewer 7JoC (Part6)**
> > > > > >
> > > > > > 8. On L138-139 the authors claim: "While known confounders can be corrected using linear adjustments, it is impractical to account for all potential confounding factors, ...". After reading the paper, it is unclear to me how the authors guarantee that their method actually learns all of the confounding factors, especially given the issue of shortcut learning in medical AI.
> > > > > >
> > > > > > Response:
> > > > > > We appreciate the reviewers' contribution in raising this crucial question, which explores the theoretical challenge of robust pattern discovery under complex and unknown confounding factors, especially given the risk of learning shortcuts in the field of medical AI. We emphasize that InfoSepGAN's core contribution lies in establishing a methodological framework to **mitigate** the impact of unknown and nonlinear confounding factors, **rather than** providing theoretical guarantees that every confounding factor can be explicitly identified. We have revised the wording of the original text (**L140-L142**).
> > > > > >
> > > > > > First, we emphasize that InfoSepGAN is based on a contrastive analysis paradigm, which has shown great potential in the study of heterogeneity in neuroimaging. Prior “1-to-k” methods, by jointly modeling data from healthy individuals and patients, overcome the limitations of unsupervised clustering (these methods applied directly in the patient data, confront main limitations in avoiding potential disease-unrelated brain variations), thus focusing on disease-specific variations rather than variations caused by age, scanner differences, or other confounding factors. The effectiveness of this approach has been validated by numerous works, such as HYDRA, CHIMERA, SmileGAN, and SurrealGAN (cited in the manuscript). InfoSepGAN builds on this paradigm but introduces a more principled framework: instead of modeling only a disease-specific attribute space, our model explicitly decomposes the generative factors into a shared context space that captures variation common to TG and BG, and a disease-specific attribute space that captures variation unique to the target group. This formulation, together with the information-theoretic loss, encourages the attribute representation to focus on pathology-related structure and encourages the context representation to absorb shared non-disease variation. Compared to "1-to-k" mapping, InfoSepGAN provides stronger disentangle capabilities and reduces the risk of non-pathological factors leaking into disease-specific representations in a wider range of cases (e.g., $P(z_c|x) \neq P(z_c|y)$).
> > > > > > Furthermore, our model design naturally resists utilizing simple metadata shortcuts, as the input for inferring both $z_a$ and $z_c$ is solely the neuroimaging data, without providing any clinical or non-clinical individual features (such as age, scanner type, or cognitive scores) as explicit labels. The model must learn the latent factors strictly from the data structure itself.
> > > > > >
> > > > > > The effectiveness of this architecture is confirmed by our ablation study (**Table 17**), which shows that removing the context module ($E_c$), thereby reverting to a traditional "1-to-$k$" design, significantly increases the influence of non-clinical factors in $z_a$ and weakens its correlation with AD-specific variables. This empirical evidence validates that the $E_c$ module is critical for successfully mitigating biases introduced by non-pathological variation, a key step for ensuring the accuracy and fairness of subsequent clinical heterogeneity analysis.
> > > > > >
> > > > > > Finally, we acknowledge that a persistent challenge in contrastive analysis is the lack of formal identifiability guarantees, as non-linear models like VAEs and GANs are generally non-identifiable. While InfoSepGAN is empirically shown to have effective disentanglement ability, achieving the complete recovery of all true generative factors remains an inherent theoretical limitation shared by all CA-based approaches, and it represents an important direction for future theoretical work in the field. We discuss this point in **Section 5.4**.

---

> > > > > > > ### Author Response · Authors · 2025-11-24
> > > > > > > **Response to Reviewer 7JoC (Part7)**
> > > > > > >
> > > > > > > 9. The authors should move their discussion on Lines 338-347 to the discussion section.  L107: "..., these approaches remain limitations ..." -> remain limited; L210/211: "Maximizing these two terms serves following purposes, ..." -> serves the following; L255/256: "We adapts and extends the terms ..." -> We adapt and extend.
> > > > > > >
> > > > > > > Response:
> > > > > > > We thank the reviewer for these suggestions. We have moved the discussion from **Lines 338-347 to Section 5.3**, and we corrected the grammatical errors in the main text as noted (**L107; L107, we have significantly reduced the content of section 3.2; L255/256, now in L230/L231**).
> > > > > > >
> > > > > > >
> > > > > > > 10. Some specific choices are unclear, for example the authors use a uniform spectrum for disorders (why a uniform and not, for example a normal distribution?). I understand it is easier in the sense that it is a bounded distribution, and control subjects can be assigned as 0, but is there a neurobiological reason for this design choice?
> > > > > > >
> > > > > > > Response:
> > > > > > > We thank the reviewer for this question. The choice of a uniform prior for the disorder spectrum is primarily motivated by interpretability and ease of downstream analysis. We map 1 to the most severe expression and 0 to the absence of pathology, creating a clear and monotonic representation that facilitates analyses such as testing correlations with clinical variables. These values indicate how a patient deviates from the reference group (healthy controls) along a specific dimension (Attr1 or Attr2) at both imaging pattern and severity levels.
> > > > > > > Using a uniform distribution also provides a bounded space that simplifies sampling, which is crucial for implementing our disentanglement loss ($L_\text{disen}$) and monotonicity loss ($L_\text{mono}$) (detailed in **Appendix A.7**).
> > > > > > >
> > > > > > > In principle, if interpretability is not a primary concern, other distributions such as normal could be used, as adopted in baseline models like CAVAE, MMCAVAE, and DoubleInfoGAN.
> > > > > > >
> > > > > > >
> > > > > > > 11. Moreover, for many spectrum disorders it is also believed that controls may lie somewhere on the same spectrum as people diagnosed with a neurological or psychiatric disorder. The authors currently set the attribute factors for controls to 0, have the authors thought about using a different prior on the control subject attributes?
> > > > > > >
> > > > > > > Response:
> > > > > > > We thank the reviewer for raising this point. In our approach, we set the attribute factors for control subjects **in the training set** to 0. This is a reasonable and intentional design choice because the control group serves as the reference population for the target group, and the information from the reference group is fully encoded in the context factors. By fixing the attribute factors of controls to a constant value, we ensure that the attribute space captures only the disease-specific variation present in the target group, and a non-dirac prior for controls would violate this “no-information” constraint.
> > > > > > >
> > > > > > > Moreover, setting the attribute factors for controls to 0 also improves interpretability, because the inferred attribute values represent how a patient deviates from the reference group (healthy controls) along a specific dimension (Attr1 or Attr2) at both the level of imaging patterns and severity.
> > > > > > >
> > > > > > > Crucially, the constraint $z_a=0$ is **only** applied to the **training data** (the BG). Once the model is trained, it can be applied to **any asymptomatic individual** (i.e., external control or a subject from a general population cohort). The model will then naturally infer their representation, thereby placing the individual at the appropriate location along the learned spectrum. We demonstrate this capability in **Appendix B.8**, where we apply our trained model to infer the attribute representations of asymptomatic individuals from the general UKBB population, effectively positioning them on the AD spectrum without imposing a zero prior.

---

### Official Review · Reviewer_heto · 2025-11-02

**Soundness:** 3
**Presentation:** 2
**Contribution:** 3
**Rating:** 4
**Confidence:** 3

**Summary:**

The manuscript proposes a contrastive generative adversarial model that disentangles pathological and non-pathological variability in neuroimaging data. The idea is to separate latent factors into shared and disease-specific components. To achieve that, the authors propose additional information-theoretic regularization and latent-space constraints. The proposed models are shown to improve over SurrealGAN and cPCA in extracting disease patterns.

**Strengths:**

- The experiments have been performed on synthetic and real datasets
- The proposed model have been compared with multiple baselines

**Weaknesses:**

- No ablation on hyperparameters for the loss function has been provided, and only mentioned that the parameters have been set to a specific value.
- Authors should support their findings with clinical literature when they say "consistent with characteristic AD pathology"
- Figure 4b is very small and is not readable if the paper is printed.
- No statistical comparison between models in Tables 4, 5, 6, 7, 8

**Questions:**

- Could the authors clarify how the findings derived from InfoSepGAN differ from conventional voxel-wise group difference analyses, and what unique insights the proposed framework provides beyond such standard methods?
- Since InfoSepGAN is a generative model, one would expect to observe progressive or continuous reconstructions as the attribute variable varies. However, the paper does not present examples of such latent traversals or severity-dependent generations. Could the authors provide visual or quantitative evidence that the learned attribute dimensions indeed capture disease severity in a monotonic and interpretable way?

---

> ### Author Response · Authors · 2025-11-24
> **Response to Reviewer heto (Part1)**
>
> It is a pleasure to read the reviewer’s positive summary of our work. We thank the reviewer for the constructive and insightful comments, which will significantly improve the clarity and rigor of our manuscript.
>
> 1. No ablation on hyperparameters for the loss function has been provided, and only mentioned that the parameters have been set to a specific value.
>
> Response:
> Thank you for raising this important point regarding the clarity of our hyperparameter evaluation. We apologize for not highlighting this sufficiently in the main text. We did perform extensive ablation and robustness studies which are detailed in **Appendix B.5**. Specifically: In ablation study: we conducted ablation experiments on both the "basic" and the "middle age gap" cases, demonstrating that all key regularization terms as well as the context encoder $E_c$ contribute to model performance. The effects are especially pronounced in the more challenging “middle age gap” scenario, where distributional shifts are larger (**Fig. 7**). In robustness study: we systematically varied key hyperparameters across the range of $50\%-150\%$ of the preset default values (**Fig.  8**), demonstrating that model performance is robust to these parameter selections. We have revised **Section 4.2 (L330 - L335)** in the main text to explicitly title and highlight the existence and findings of these studies, ensuring the reader is directed to the detailed results in the Appendix.
>
> Regarding the use of specific default values: This choice ensures a fair comparison with baseline models, which also rely on default settings, and provide a stable and reproducible configuration for future researchers. However, for users wishing to refine hyperparameters for a new dataset, we also introduced the Pattern-agr-index (ACI) metric. Since agreement among independently trained models correlates well with representation accuracy, ACI offers a practical and data-driven criterion for selecting hyperparameters (**Appendix A.9** section Model selection using ACI, **Fig. 8**).
>
> 2. Authors should support their findings with clinical literature when they say "consistent with characteristic AD pathology".
>
> Response:
> We thank the reviewer for the insightful comment and agree that grounding our findings in established clinical literature is essential. We have revised **Section 5.3 (L470-L472)** and added citations documenting the cortical and subcortical atrophy distributions commonly reported in AD. The diffuse cortical atrophy observed for Attr1 is consistent with previously reported parietal–occipital–temporal cortical thinning in AD [1]. The focal medial temporal and subcortical atrophy pattern associated with Attr2 aligns with established AD-related neurodegeneration [2, 3]. These references now more clearly support our interpretation.
>
>
> 3. Figure 4b is very small and is not readable if the paper is printed. No statistical comparison between models in Tables 4, 5, 6, 7, 8.
>
> Response:
> We appreciate the feedback on figure quality and result presentation. We have revised **Fig. 4b** to only present the most representative results clearly in the main paper. The full, detailed results and supplementary analyses, including a larger version of the Representational Similarity Analysis (RSA), have shown in **Appendix B.9 (Table 16, 17, 18 and Fig. 12)** for better readability. These extended results also include the ablation variant of InfoSepGAN without the context encoder ($E_c$), further demonstrating that incorporating the context module contributes positively to representation disentanglement.
>
> For the comparisons reported in **Tables 4–8 (now Table 5-10)**, we have now included statistical significance testing between our proposed InfoSepGAN and the best-performing baseline model for each dataset. Each model was independently run 10 times, and we applied a two-sided independent t-test to assess whether the observed performance differences were statistically significant.
>
> [1] Du, An-Tao, et al. "Different regional patterns of cortical thinning in Alzheimer's disease and frontotemporal dementia." Brain 130.4 (2007): 1159-1166.
>
> [2] Scheltens, Philip, et al. "Atrophy of medial temporal lobes on MRI in" probable" Alzheimer's disease and normal ageing: diagnostic value and neuropsychological correlates." Journal of Neurology, Neurosurgery & Psychiatry 55.10 (1992): 967-972.
>
> [3] Yi, Hyon-Ah, et al. "Relation between subcortical grey matter atrophy and conversion from mild cognitive impairment to Alzheimer's disease." Journal of Neurology, Neurosurgery & Psychiatry 87.4 (2016): 425-432.

---

> > ### Author Response · Authors · 2025-11-24
> > **Response to Reviewer heto (Part2)**
> >
> > 4. Could the authors clarify how the findings derived from InfoSepGAN differ from conventional voxel-wise group difference analyses, and what unique insights the proposed framework provides beyond such standard methods?
> >
> > Response:
> > Thank you for this clear and important question. The insights provided by InfoSepGAN fundamentally differ from those obtained through traditional voxel-based morphometry (VBM) or voxel-wise inter-group differential analysis.
> > Traditional VBM identifies group-level mean differences (e.g., AD vs. controls) or tests voxel-wise associations with a single predefined clinical score. In contrast, the core contribution of InfoSepGAN is to decompose disease-related variability into multiple pathological dimensions (attribute factors, $z_a$) while simultaneously separating out non-pathological sources of variation (context factors, $z_c$). This enables InfoSepGAN to (i) quantify AD severity for each participant along two dimensions (Attr1 and Attr2), each reflecting a distinct anatomical atrophy pattern, and (ii) reduce bias caused by non-disease variability, which is not possible with traditional VBM.  Second, InfoSepGAN provides counterfactual generation that enables ROI-level contrast between each patient and their “synthetic twin” (**Section 3.6**). These counterfactual comparisons reveal subject-specific, anatomically interpretable atrophy signatures, operating at the ROI level. VBM results, by contrast, rely on population-level voxel-wise statistical testing. Despite these methodological differences, our results show strong consistency between the two approaches, as illustrated in **Fig. 14b, c**, where regions highlighted by InfoSepGAN align with those identified by voxel-wise VBM associations.
> >
> > VBM in our work is used solely as a visualization and interpretation tool, not as a discovery mechanism. Specifically, we perform voxel-wise associations between tissue density and each learned attribute factor, controlling for age, sex, intracranial volume, and the remaining attribute factor. In other words, the novelty lies in the learned latent attributes themselves, and VBM simply provides an anatomical interpretation of these data-driven dimensions. We have revised the caption of **Fig. 4a** and clarified the description in **Appendix B.6 (VBM Visualization Analysis)** to explicitly state the role of VBM as a visualization tool for interpreting learned representations rather than for primary statistical inference.
> >
> > 5. Could the authors provide visual or quantitative evidence that the learned attribute dimensions indeed capture disease severity in a monotonic and interpretable way?
> >
> > Response:
> > We thank the reviewer for this insightful comment. We fully agree that providing explicit evidence of monotonic progression is crucial to validate the interpretability of the learned attribute factors. We refer the reviewer to **Figure 14 in Appendix B.10** which directly addresses this point.
> >
> > As shown in **Figure 14b** and the newly added voxel-level visualizations in **Figure 14c**, we stratified subjects into subgroups according to monotonically increasing attribute scores. The corresponding ROI change ratios derived from synthetic counterfactuals reveal a clear and progressive increase in the severity and spatial extent of the pathological patterns associated with Attr1 and Attr2. These results confirm that the learned attribute factors capture disease severity in a monotonic and interpretable manner.
> >
> > Furthermore, we have revised the main text in **L494-L500** to explicitly emphasize this monotonicity, highlighting that InfoSepGAN provide progressive, severity-dependent visualizations of pathological variations.

---

### Author Response · Authors · 2025-11-29
**Summary for the Area Chair**

Dear Area Chair,

We thank the reviewers for their constructive feedback. We have supplemented with additional experiments and revised the manuscript. We provide this summary to highlight key evidence and address prior reviewers' concerns to aid your final assessment.

*1. Supplementary experiments*

* **Robustness and Ablation** (Addressing Reviewer heto, 7JoC):
    * We apologize for not emphasizing this earlier. We conducted extensive ablation and robustness studies (**Fig. 7, 8**), including removal of regularization terms, the context encoder $E_c$, and hyperparameter variations (50%-150%). Results show that the full model consistently outperforms ablated versions and is robust to parameter changes. In particular, ablating the context encoder on the ADNI dataset revealed its necessity: removing $E_c$ weakened attribute associations with AD biomarkers and increased leakage of nonclinical variation (age, site, scanner) into the attribute space (**Fig. 12, Table 17**), confirming InfoSepGAN's ability to handle anatomical variability.

* **Generalizability on External Datasets** (Addressing Reviewer 7JoC):
    * To evaluate generalizability, we retrained our model on 1,000 UK Biobank controls as a reference dataset and applied it to over 47,000 asymptomatic subjects. The model was consistent both qualitatively and quantitatively with the model initially trained on ADNI: atrophy maps replicated diffuse cortical (Attr1) and focal medial temporal/subcortical (Attr2) findings (**Fig. 11**), and the learned patterns maintained quantitative consistency (PCI = 0.774).  Moreover, the learned attribute factors in the general population showed significant associations with clinical features (**Table 14**).

* **Comparison with Modern Baselines** (Addressing Reviewer 1Tcw, 7JoC):
    * We compared InfoSepGAN against three recent contrastive analysis methods (CAVAE, DoubleInfoGAN, MMCAVAE) on semi-synthetic and ADNI datasets. InfoSepGAN significantly outperformed these baselines in recovering progressive disease patterns (**Table 10**) and more effectively disentangled AD-specific biomarkers from nonclinical features (**Table 18**).

*2. Clarifying Misunderstandings (Reviewer 7JoC)*

Reviewer 7JoC raised concerns regarding the separation of confounders (e.g., age, scanner), conditional independence, and the risk of shortcut learning. We addressed these with quantitative evidence and architectural justification:

* Our RSA analysis shows that $z_a$ does not preferentially encode age ($\Delta\tau = -0.0049$, $p = 0.1927$), with any residual association reflecting biologically meaningful age-related AD effects. Similarly, while a minimal correlation with scanner type exists in $z_a$, it is significantly weaker than in the context features ($\Delta\tau = -0.0224$, $p < 0.0001$), which overwhelmingly capture non-disease variation (**Table 16, Fig.12**). Importantly, unlike baselines whose attribute spaces show negligible associations with AD biomarkers, InfoSepGAN’s attribute features exhibit the strongest correlations with AD-specific properties (e.g., ADNI-MEM, CSF-Tau), demonstrating superior disentanglement and clinical utility (**Table 18**).

* Attribute and context factors show very low pairwise correlations ($|r| \leq 0.12$), indicating minimal shared information and near-independence in practice (**Fig. 13**). Our design balances independence and model performance via the information-theoretic loss (**Section 3.2**) and independent sampling of $z_a$ and $z_c$ during training to limit information leakage.

* InfoSepGAN is designed to **mitigate** the impact of unknown/nonlinear confounding factors, not provide a theoretical guarantee against all of them. Our approach leverages the contrastive analysis paradigm to encourage the context space to absorb variation shared by the target and background groups (e.g., non-pathological variations). Crucially, the model receives only neuroimaging data as input, forcing it to learn latent factors from the data structure itself, inherently resisting simple metadata shortcuts. Our experiments demonstrate that InfoSepGAN reduces the risk of non-pathological variations leaking into the disease-specific representations compared to prior '1-to-k' baseline methods, which is crucial for improving the accuracy and fairness of subsequent downstream analyses.

*3. Consensus on Novelty and Significance*

* **Reviewer 1Tcw (Score 6)** actively engaged and confirmed that our additional experiments (CA baselines, real-data clarification) "address their main technical questions" and maintained a positive assessment.

*4. Conclusion*

We believe our **25** (semi-)synthetic experiments, external validation on UKBB, and comparisons with modern CA baselines firmly establish InfoSepGAN's robustness and superiority in disentangling heterogeneous disease patterns. We trust these fact-based responses provide a solid basis for your decision.


Sincerely,

The Authors

---

### Meta-Review · Area_Chair_g7if · 2026-01-06

**Summary:**

This paper proposes InfoSepGAN, a contrastive generative framework for disentangling pathological imaging patterns from non pathological variation in neuroimaging data under distribution mismatch between background and target populations.

**Reviewer Concerns:**

While the authors added extensive ablations, robustness analyses, and additional baselines, some reviewers remain unconvinced that the disentanglement of confounders such as scanner effects is sufficiently reliable in real world settings. There is also concern that the framework relies on many interdependent regularization terms and hyperparameters, which may limit reproducibility and practical adoption. Finally, disagreement remains regarding whether the contribution constitutes a clear methodological advance beyond prior contrastive generative approaches, or primarily an incremental refinement tailored to a specific clinical setting.

**Reviewer Scores:**

Overall, the three reviews do not converge toward a clear acceptance consensus. One reviewer rates the paper clearly below threshold, citing concerns about soundness, presentation, and disentanglement validity.

---

### Decision · Program_Chairs · 2026-01-26

Reject